# LLM-Select: Feature Selection with Large Language Models

**Daniel P. Jeong**[1]                                                          *danielje@cs.cmu.edu*

**Zachary C. Lipton**[1,2]                                                       *zlipton@cmu.edu*

**Pradeep Ravikumar**[1]                                                        *pradeepr@cs.cmu.edu*

[1]*Machine Learning Department, Carnegie Mellon University*
[2]*Abridge AI*

**Reviewed on OpenReview:** *https://openreview.net/forum?id=16f7ea1N3p*

## Abstract

In this paper, we demonstrate a surprising capability of large language models (LLMs): given only input feature names and a description of a prediction task, they are capable of selecting the most predictive features, with performance rivaling the standard tools of data science. Remarkably, these models exhibit this capacity across various query mechanisms. For example, we zero-shot prompt an LLM to output a numerical importance score for a feature (e.g., "blood pressure") in predicting an outcome of interest (e.g., "heart failure"), with no additional context. In particular, we find that the latest models, such as GPT-4, can consistently identify the most predictive features regardless of the query mechanism and across various prompting strategies. We illustrate these findings through extensive experiments on real-world data, where we show that LLM-based feature selection consistently achieves strong performance competitive with data-driven methods such as the LASSO, despite never having looked at the downstream training data. Our findings suggest that LLMs may be useful not only for selecting the best features for training *but also for deciding which features to collect in the first place.* This could benefit practitioners in domains like healthcare and the social sciences, where collecting high-quality data comes at a high cost.

## 1 Introduction

Transformer-based large language models (LLMs) pretrained on massive text corpora for next-word prediction exhibit the remarkable capability to generalize to unseen tasks, simply by conditioning on an input prompt that contains task-relevant instructions and a small number of examples (Vaswani et al., 2017; Radford et al., 2019; Brown et al., 2020). With sufficient model scale and an appropriate prompting strategy, these models demonstrate strong performance on various commonsense, symbolic, and arithmetic reasoning tasks (Lewkowycz et al., 2022; Wei et al., 2022b; Kojima et al., 2022; Suzgun et al., 2023; Anil et al., 2023) and complex question-answering and prediction tasks that require real-world knowledge (Petroni et al., 2019; Liévin et al., 2022; Singhal et al., 2023; Manikandan et al., 2023). Such findings suggest that by pretraining on vast amounts of text from various domains, LLMs encode rich knowledge about real-world relationships, which they can leverage for performing various downstream tasks (Choi et al., 2022; Moor et al., 2023).

In this paper, we demonstrate that LLMs are capable of performing feature selection for supervised learning tasks. Given that we are often aware of the real-world semantics associated with the input features (e.g., "blood pressure") and the target outcome (e.g., "heart failure") in a downstream training dataset, we investigate effective ways of prompting an LLM to identify the most informative features for predicting the outcome (Figure 1(a)). For example, we prompt the LLM with "Rank the following features by their importance for predicting the incidence of heart failure: blood pressure, . . ., creatinine."[1], and select the top-ranked features

---

[1]We note that the examples here are simplifications of the exact prompts used in our study. See Appendices C–E for details.

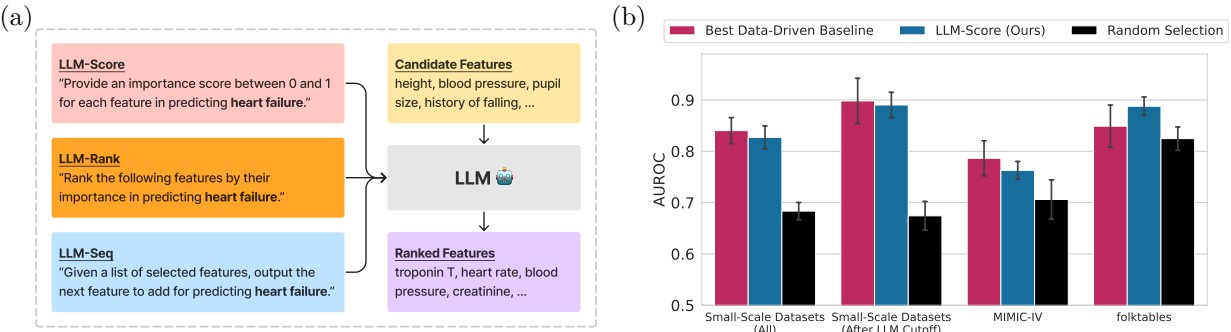

Figure 1: Selecting features by zero-shot prompting an LLM leads to strong downstream predictive performance, competitive with data-driven feature selection methods. (a) Overview of our proposed LLM-Score, LLM-Rank, and LLM-Seq methods (Section 3). (b) Average test AUROC (higher is better) on classification datasets when selecting the top 30% of features according to the best-performing data-driven baseline on each dataset (in red), LLM-Score based on GPT-4 (in blue), and a random feature selection baseline (in black). Error bars indicate standard error across datasets in each group.

to train a downstream prediction model. Alternatively, for each candidate feature, we prompt the LLM with "Provide a feature importance score between 0 and 1 for ⟨candidate feature⟩ for predicting the incidence of heart failure.", and select the features with the highest LLM-generated scores for training.

Surprisingly, we find that *even without looking at the training data*, these models are capable of identifying the most predictive features, with performance rivaling those of data-driven feature selection methods such as the LASSO (Tibshirani, 1996) (Figure 1(b)). Such a result is counterintuitive; given that datasets with identical input feature names can correspond to arbitrarily different data distributions, due to factors such as selection bias and confounding, it is not a priori obvious that selecting features only based on their names will be effective. Remarkably, we find that for the latest models such as GPT-4, even *zero-shot* prompting the LLM to generate a numerical feature importance score one-feature-at-a-time can lead to strong feature selection performance. Moreover, we find that such LLM-generated feature importance scores are correlated with several commonly used feature importance metrics (e.g., Fisher score (Gu et al., 2011), SHAP (Lundberg & Lee, 2017)), suggesting that LLMs are capable of distilling the real-world relationships encoded in their parameters into statistically meaningful scores. We demonstrate these findings by stress-testing our LLM-based feature selection methods against traditional data-driven methods on real-world datasets from diverse domains (e.g., finance, healthcare, criminal justice), using both closed-source (GPT-4 (OpenAI, 2023), GPT-3.5 (Brown et al., 2020)) and open-source LLMs (Llama-2 (Touvron et al., 2023)) of various sizes.

Our main contributions can be summarized as follows:

1. We propose three approaches to prompting an LLM for feature selection: (i) selecting features with the highest LLM-generated feature importance scores (LLM-Score); (ii) selecting features based on an LLM-generated ranking (LLM-Rank); and (iii) sequentially selecting features in a dialogue with an LLM (LLM-Seq) (Section 3).

2. We show that even without access to the downstream training data, LLMs of sufficient scale achieve strong feature selection performance on real-world datasets, often competitive with traditional data-driven feature selection methods such as the LASSO (Result 1–2, Section 4.1; Section 4.2).

3. We comprehensively assess the sensitivity of LLM-based feature selection methods to various prompting strategies and demonstrate that even *zero-shot* prompting the LLM with no additional context about the downstream data can elicit strong feature selection performance (Result 3, Section 4.1).

4. We show that as model scale increases, LLM-generated importance scores generally exhibit higher rank correlation with commonly used feature importance metrics (Result 4, Section 4.1).

Our findings suggest that LLMs may be useful not only for selecting the most predictive features after data collection, **but also for deciding what features to collect in the first place**. This could benefit

practitioners in domains like healthcare and the social sciences, where obtaining high-quality data (e.g., running medical tests for patients, designing survey questions) can be expensive and time-consuming.

## 2 Related Work

### 2.1 Prompting LLMs

Prompting is an effective method for adapting off-the-shelf LLMs to perform new tasks unseen during training, without explicit gradient updates to the model parameters (Radford et al., 2019; Liu et al., 2023). In a standard prompting setup, an output is autoregressively sampled from an LLM conditional on text descriptions of a desired task and optionally a set of in-context input-output examples (Brown et al., 2020) and used as a solution for the given task. Even without task-specific fine-tuning, such a zero-shot or few-shot in-context learning approach can be surprisingly effective for adapting pretrained LLMs towards a wide range of natural-language tasks (Hendrycks et al., 2021; Lin et al., 2022; Patel & Pavlick, 2022; Srivastava et al., 2023), given a language model of sufficient scale (Wei et al., 2022a).

Meanwhile, several works show that LLM outputs can be highly sensitive to the specifics of the input prompt and choice of decoding strategy (Jiang et al., 2020; Zhao et al., 2021; Sclar et al., 2024; Jeong et al., 2024a;b). As such, choosing an appropriate prompting strategy is crucial, especially for challenging reasoning tasks. In our experiments, we mainly consider two prompting techniques—chain-of-thought prompting (CoT; Wei et al., 2022b) and self-consistency decoding (Wang et al., 2023). We focus on these methods as they often dramatically boost performance on tasks that require multi-step reasoning (Kojima et al., 2022; Lewkowycz et al., 2022; Chen et al., 2023), which we hypothesized to be important for feature selection.

**Chain-of-thought prompting (CoT).** CoT prompting (Wei et al., 2022b) is a few-shot prompting method that augments each input-output example with a *chain-of-thought*—a coherent series of natural-language reasoning steps leading to the correct answer. Wang et al. (2023) show that given a large-enough model and an appropriately designed input prompt, CoT prompting can elicit logically consistent step-by-step solutions from the LLM and substantially boost performance on complex tasks such as solving math problems (Lewkowycz et al., 2022; Imani et al., 2023) and answering commonsense (Suzgun et al., 2023; Anil et al., 2023) and knowledge-intensive reasoning questions (Liévin et al., 2022; Singhal et al., 2023).

**Self-consistency decoding.** A common sampling strategy used for LLMs is greedy decoding (Radford et al., 2019; Brown et al., 2020; Chowdhery et al., 2022), where at each token-generation step, the token with the highest probability is taken as the output. In *self-consistency decoding* (Wang et al., 2023) on the other hand, multiple outputs are *randomly* sampled from the LLM (via e.g., temperature sampling (Ackley et al., 1985)) and marginalized to generate the final prediction. Prior works suggest that for CoT prompting, self-consistency decoding can significantly boost performance and that it is especially beneficial when a diverse set of reasoning paths are possible for solving a given task (Wang et al., 2023; Lewkowycz et al., 2022).

**Extensions of CoT prompting.** More recently, tree-of-thoughts (ToT; Yao et al., 2023) and graph-of-thoughts (GoT; Besta et al., 2024) prompting have been proposed as generalizations of CoT prompting (with self-consistency decoding) to further enhance the exploration and evaluation of multiple reasoning paths. ToT prompting formulates reasoning as a traversal over a tree of plausible partial solutions, allowing for explicit branching and backtracking of diverse reasoning paths in a hierarchical manner. GoT prompting further generalizes ToT prompting by formulating reasoning via a graph, additionally allowing for exploration and dependencies across different reasoning paths. In our paper, we focus on standard CoT prompting (Wei et al., 2022b) for an initial investigation on LLM-based feature selection in the simplest setting. We leave an in-depth investigation of whether extensions such as ToT and GoT prompting can improve LLM-based feature selection performance as future work.

### 2.2 Feature Selection

Feature selection is a classical machine learning problem, where given a set of candidate features, the goal is to select the most informative feature subset that is predictive of an outcome of interest (Blum & Langley,

1997; Guyon & Elisseeff, 2003; Chandrashekar & Sahin, 2014; Li et al., 2017). Feature selection methods can generally be grouped into three categories: filter, wrapper, and embedded methods. Filter methods (Lazar et al., 2012) select features by ranking them according to a statistical or information-theoretic criterion— e.g., mutual information (Lewis, 1992; Ding & Peng, 2005; Bennasar et al., 2015), Fisher score (Duda et al., 2001; Gu et al., 2011), maximum mean discrepancy (Song et al., 2012)—and choosing the top ranked features, independent of the downstream learning algorithm. Wrapper methods identify a locally optimal feature subset that maximizes the performance of the downstream prediction model (Kohavi & John, 1997; Yamada et al., 2014; Chen et al., 2017; Climente-González et al., 2019), often by employing a heuristic search strategy (e.g., sequential selection (Ferri et al., 1994; Luo & Chen, 2014; Yasuda et al., 2023), recursive feature elimination (RFE; Guyon et al., 2002)). Embedded methods select features as part of the model learning process, most commonly based on regularization techniques that encourage feature sparsity (Tibshirani, 1996; Yuan & Lin, 2006; Feng & Simon, 2017; Lemhadri et al., 2021), and others based on specialized neural network architectures (Lu et al., 2018; Balın et al., 2019; Yamada et al., 2020). In our experiments, we compare our methods against traditional feature selection baselines from *all three categories*.

**LLMs for feature selection.** In a prior work most similar to ours, Choi et al. (2022) propose the *LMPriors* framework, where they prompt the `davinci-instruct-beta` variant of GPT-3 (Brown et al., 2020) to answer whether each candidate feature should be used to predict the target outcome, and select features whose difference in log-probabilities for generating a "Y" (Yes) or "N" (No) token crosses a predefined threshold. Our work differs from theirs in two key aspects. First, we propose three *different* feature selection methods which all directly use the generated text output and not the associated token probabilities, which are often not directly accessible in closed-source, proprietary LLMs. Second, we provide a more comprehensive evaluation across various model scales and prompting strategies on a larger collection of datasets and derive practical insights. Our proposed feature selection methods perform as strongly as theirs even without the same level of access into the LLM (Appendix A.3.3).

## 3 Selecting Features with LLMs

We address the standard supervised learning setup where, given labeled data $\mathcal{D} = \{(\boldsymbol{x}^{(i)}, y^{(i)})\}_{i=1}^n$ with $\boldsymbol{x}^{(i)} \in \mathbb{R}^d$ and $y^{(i)} \in \mathcal{Y}$, our goal is to learn a prediction model $\hat{f} \in \mathcal{F}$ such that $\hat{f} = \arg\min_{f \in \mathcal{F}} \mathbb{E}_{\mathcal{D}}[\mathcal{L}(f, \mathcal{D})]$ for some model class $\mathcal{F}$ and loss function $\mathcal{L}$. We assume access to *concepts* $\boldsymbol{c} = [c_1, \ldots, c_d]$ for the input features and $c_y$ for the prediction target, which are text descriptions that capture their real-world semantics. For example, when predicting heart failure (1 if positive, 0 otherwise) given a patient's blood pressure and weight measurements, $\boldsymbol{c} = [$"blood pressure", "weight"$]$, $c_y =$ "heart failure", and $\boldsymbol{x} = [x_1, x_2]$ denotes the numerical measurements used to learn $\hat{f}$. Concept annotations are widely available in many practical settings, e.g., as column names in tabular datasets or via datasheets that contain auxiliary metadata. For feature selection, our goal is to find a subset $S \subseteq \{1, \ldots, d\}$ of size $k \ll d$ such that a model $\hat{f}_S$ trained on $\mathcal{D}_S = \{(\boldsymbol{x}_S^{(i)}, y^{(i)})\}_{i=1}^n$, where $\boldsymbol{x}_S^{(i)} = [\boldsymbol{x}_{S_1}^{(i)}, \ldots, \boldsymbol{x}_{S_k}^{(i)}]$, achieves strong performance under a budget on $k$.

### 3.1 Feature Selection with LLMs

To leverage a pretrained LLM $\mathcal{M}$ for feature selection, we prompt $\mathcal{M}$ with the input concepts $\boldsymbol{c}$ and target concept $c_y$, and select features based on the generated output. We consider the following three approaches: (i) selecting features based on LLM-generated feature importance scores; (ii) selecting features based on an LLM-generated ranking; and (iii) sequentially selecting features in a dialogue with an LLM. We design separate prompt templates for each approach and denote them by prompt$_{\text{score}}$, prompt$_{\text{rank}}$, and prompt$_{\text{seq}}$, respectively. Each prompt template can be viewed as a function of the input and target concepts which outputs a set of natural-language instructions tailored to the corresponding selection strategy. While generally, text outputs generated from an LLM need to be processed further to extract the relevant information, we omit such steps in the notation below for simplicity.

**Selection based on LLM-generated feature importance scores (**LLM-SCORE**).** In this approach, we prompt $\mathcal{M}$ for a set of numerical feature importance scores $\boldsymbol{s} = [s_1, \ldots, s_d]$ with $s_j \in [0, 1]$ $\forall j \in \{1, \ldots, d\}$,

where a high $s_j$ indicates that an input concept $c_j$ is closely related to $c_y$. Formally, we can represent this as

$$s_j = \mathcal{M}(\text{prompt}_{\text{score}}(c_j, c_y)), \quad \forall j \in \{1, \ldots, d\}. \tag{1}$$

We then define $S$ to be the set of indices of the top-$k$ concepts with the highest importance scores and use $\mathcal{D}_S$ to learn a downstream prediction model $\hat{f}_S$. Given that the LLM is only given a single input concept $c_j$ and the target concept $c_y$ each time it is prompted, we hypothesize that $s_j$ captures the *marginal* importance of each feature for predicting the target, as informed by the knowledge encoded in $\mathcal{M}$. We note that the feature importance scores $\boldsymbol{s}$ are *directly* parsed from the text output and do not correspond to the token probabilities associated with generating the text output. We also note that the score range of $[0, 1]$ is an arbitrary choice and therefore evaluate the sensitivity of LLM-SCORE to different choices in Appendix A.3.4.

**Selection based on an LLM-generated feature ranking (LLM-RANK).** In this approach, we prompt $\mathcal{M}$ for a ranking $\boldsymbol{r} = [c_{1'}, \ldots, c_{d'}]$ of all input concepts, where the input concepts $\boldsymbol{c}$ are ordered by their conceptual relevance to $c_y$. Formally, we can represent this as

$$\boldsymbol{r} = \mathcal{M}(\text{prompt}_{\text{rank}}(\boldsymbol{c}, c_y)). \tag{2}$$

We define $S$ to be the set of indices of the top-$k$ highest ranked concepts and use $\mathcal{D}_S$ to learn a downstream prediction model $\hat{f}_S$. We hypothesize that the rank of each input concept reflects its *relative* importance for predicting the target, with respect to all of the other input concepts in $\boldsymbol{c}$.

**Sequential selection in a dialogue with an LLM (LLM-SEQ).** In this approach, we consider a selection strategy analogous to sequential selection methods. We start with an empty set of concepts and iteratively add a new concept by prompting the LLM to select a candidate concept that would maximally improve the cross-validation performance of a downstream prediction model. Formally, assuming that our goal is to select $k$ concepts, at each iteration $t = 1, \ldots, k$, we have

$$c^{(t)} = \mathcal{M}(\text{prompt}_{\text{seq}}(\boldsymbol{c}_{S_{t-1}}, c_y)), \tag{3}$$

where $c^{(t)}$ denotes the $t$-th selected input concept, $S_t \subseteq \{1, \ldots, d\}$ denotes the subset of concept indices selected up to the $t$-th iteration, and $S_0 = \emptyset$. We then use $\mathcal{D}_{S_t}$ to train a downstream prediction model $\hat{f}_{S_t}$, where we tune the hyperparameters via 5-fold cross-validation. The cross-validation performance of $\hat{f}_{S_t}$ on $\mathcal{D}_{S_t}$ is then appended to the prompt used for the next iteration (Appendix E). For the initialization of the feature subset, we also consider starting with $S_1 = \{\arg\max_j s_j\}$ containing the concept with the highest score from Equation (1) and iterating over $t = 2, \ldots, k$. However, we focus on the former approach in the main text, as it empirically performs better than the latter (we compare the two approaches in Appendix A.3.5). We hypothesize that this approach encourages the LLM to select a feature that is maximally informative with respect to the feature subset already selected, at each iteration of the algorithm. Meanwhile, we note that since LLM-SEQ is a *greedy* sequential selection approach, the resulting top-$k$ feature subset $S$ is not guaranteed to be globally optimal. This limitation arises as at each iteration $t$, the next feature to add is determined with respect to $S_{t-1}$, which precludes an exploration all possible subsets of $k$.

For instantiating each method, we use LLMs that have been fine-tuned via instruction tuning and reinforcement learning from human feedback (RLHF) (Christiano et al., 2017; Stiennon et al., 2022; Ouyang et al., 2022), which are generally better at following instructions and capable of handling conversational contexts. However, we emphasize that the LLMs are not fine-tuned in any way on the downstream dataset $\mathcal{D}$.

## 4 Experiments

In this section, we demonstrate the effectiveness of the three LLM-based feature selection methods introduced in Section 3 on various real-world prediction tasks. For all of our experiments, we use the following LLMs:

1. GPT-4 (OpenAI, 2023): $\sim$1.7T parameters,
2. GPT-3.5 (Brown et al., 2020): $\sim$175B parameters,

3. Llama-2 (Touvron et al., 2023): 70B parameters,
4. Llama-2 (Touvron et al., 2023): 13B parameters,
5. Llama-2 (Touvron et al., 2023): 7B parameters.

For GPT-4 and GPT-3.5, we use the `gpt-4-0613` and `gpt-3.5-turbo` models available via the OpenAI API. *We clarify that for both models, the official parameter counts have not been disclosed by OpenAI, and that the approximate ($\sim$) number of parameters listed here are rumored estimates*. For Llama-2, we use the HuggingFace checkpoints `llama-2-70b-chat-hf`, `llama-2-13b-chat-hf`, and `llama-2-7b-chat-hf` and use the vLLM framework (Kwon et al., 2023) to increase throughput and speed up output generation.

**Prompt design.** We provide all of the prompt templates used in our experiments in Appendices C–E. The prompt templates were carefully constructed to reliably elicit the desired response in the correct format from each LLM. In the *default* template, we only include (i) the main system prompt (e.g., "Your task is to provide a feature importance score between 0 and 1 for predicting ⟨target concept⟩ and a reasoning behind how the importance score was assigned."), (ii) output format instructions (e.g., "Output your answer in a JSON format."), and (iii) the main user prompt (e.g., "Provide a score and reasoning for ⟨concept⟩."). We emphasize that the default prompts are not "fine-tuned" on each dataset in any way, as they only embed the input and target concepts and no other dataset-specific information. Meanwhile, we examine how the following changes to the input prompt affect feature selection performance:

1. **Adding dataset-specific context:** When prompting the LLM to select features for dataset $\mathcal{D}$, we investigate whether adding auxiliary information about $\mathcal{D}$ (e.g., data collection process, cohort) helps better contextualize the importance of each feature and improve feature selection performance.
2. **Adding few-shot examples:** We investigate whether adding few-shot examples improves LLM-based feature selection performance via in-context learning. For instance, when generating feature importance scores (as in Equation (1)), we include example concepts from $c$ along with their human-annotated feature importance scores (e.g., ["blood pressure", 0.9]) in prompt$_{\text{score}}$.
3. **Adding CoT explanations:** Given the empirical success of CoT prompting (Wei et al., 2022b) in improving the reasoning capabilities of LLMs in few-shot settings, we investigate whether adding CoT reasoning into the few-shot examples (e.g., ["blood pressure", "Blood pressure is important for... Thus, the score is 0.9.", 0.9]) improves the performance of LLM-based feature selection.

For all three changes, we manually construct the relevant inputs via human annotation. For dataset-specific context, we manually summarize the metadata associated with each dataset (e.g., sourced from the "Data Card" of a Kaggle dataset). For selecting the few-shot examples and generating their CoT explanations, we prioritize features whose semantic relevance to the target is unambiguous and least open to subjective interpretation, based on common sense and relevant domain knowledge. For a given dataset, the few-shot examples and CoT explanations provided to the LLMs remain fixed across all queries. See Appendix C for full details on the examples and CoT explanations for each dataset considered in Sections 4.1–4.2.

Concretely, we consider the following six variations of the prompt template in our experiments:

1. Default (No change),
2. Default + Examples,
3. Default + Examples with CoT,
4. Default + Context,
5. Default + Context + Examples,
6. Default + Context + Examples with CoT.

For LLM-RANK and LLM-SEQ, these variations are less straightforward to implement, due to e.g., limited context windows or ambiguity in constructing a valid example. For example, for LLM-RANK, the full list of concepts must be added, along with the main system prompt and output format instructions. We thus focus on LLM-SCORE for exploring how these variations impact feature selection performance.

**Decoding.** By default, we use greedy decoding (i.e., sampling with temperature $T = 0$), given its straight-forward and deterministic behavior. For GPT-4 and GPT-3.5, we account for their inherent non-determinism at the time of writing by sampling 5 responses with $T = 0$ and averaging the feature importance scores parsed from the text generations. Meanwhile, as there is no clear notion of ground truth when assigning feature importance (e.g., what is the ground-truth importance score for "blood pressure" when predicting "heart failure"?) and multiple reasoning paths are possible for determining importance, we also consider self-consistency decoding (Wang et al., 2023). For the latter, we set $T = 0.5$ and average across 5 samples.

**Source code.** To ensure the reproducibility of our results, we open-source the source code used for all of our evaluations detailed below via our GitHub repository[2].

## 4.1 Evaluation on Small-Scale Datasets

We compare LLM-SCORE, LLM-RANK, and LLM-SEQ against several feature selection methods using small-scale, low-dimensional datasets from various domains (e.g., healthcare, criminal justice), each with ∼10–70 features after preprocessing. Here, we focus on the small-scale setting to stress-test the LLM-based feature selection methods in various ways and to allow comparison with methods that are less scalable to high-dimensional settings (e.g., sequential selection baselines). We use seven binary classification datasets (`Credit-G`, `Bank`, `Give Me Some Credit`, `COMPAS Recidivism`, `Pima Indians Diabetes`, `AUS Cars`*, `YouTube`*) and seven regression datasets (`CA Housing`, `Diabetes Progression`, `Wine Quality`, `Miami Housing`, `Used Cars`, `NBA`*, `NYC Rideshare`*), where those marked with an asterisk (*) are datasets **published after the pretraining data cutoff dates of all of the LLMs we evaluate**[3]. We include these datasets to ensure that our findings generalize to datasets that the LLMs have not been trained on. We provide the remaining details on all datasets in Appendix A.1.

We evaluate each feature selection method by measuring how the test performance of a *downstream prediction model* changes as we vary the proportion of features selected from 10% to 100%, in approximately 10% increments. On each dataset and at each proportion, we measure the test performance of an $L_2$-penalized logistic/linear regression model trained using the selected features. For each training run, we perform model selection via grid search with 5-fold cross-validation. We use the area under the ROC curve (AUROC) and mean absolute error (MAE) to measure performance on classification and regression tasks, respectively. For LLM-based feature selection, we select the *concepts $c$* in 10% increments, which may each correspond to more than one feature after preprocessing if the concept is categorical (e.g., one-hot encoding for "ethnicity").

**Baselines.** We compare the LLM-based feature selection methods against the following baselines:

1. LassoNet (Lemhadri et al., 2021),
2. the LASSO (Tibshirani, 1996),
3. forward sequential selection,
4. backward sequential selection,
5. recursive feature elimination (RFE; Guyon et al., 2002),
6. minimum redundancy maximum relevance (MRMR; Ding & Peng, 2005),
7. filtering by mutual information (MI; Lewis, 1992),
8. HSIC-Lasso (Yamada et al., 2014),
9. Concrete Autoencoder (CAE; Balın et al., 2019),
10. Sequential Attention (SA; Yasuda et al., 2023),
11. random feature selection.

---

[2]https://github.com/taekb/llm-select
[3]For GPT-4 and GPT-3.5, the cutoff date is Sep., 2021 at the time of writing (reference). For Llama-2, the cutoff date is Sep., 2022 (reference). As LLMs are regularly updated, the knowledge cutoff dates may be different in more recent versions.

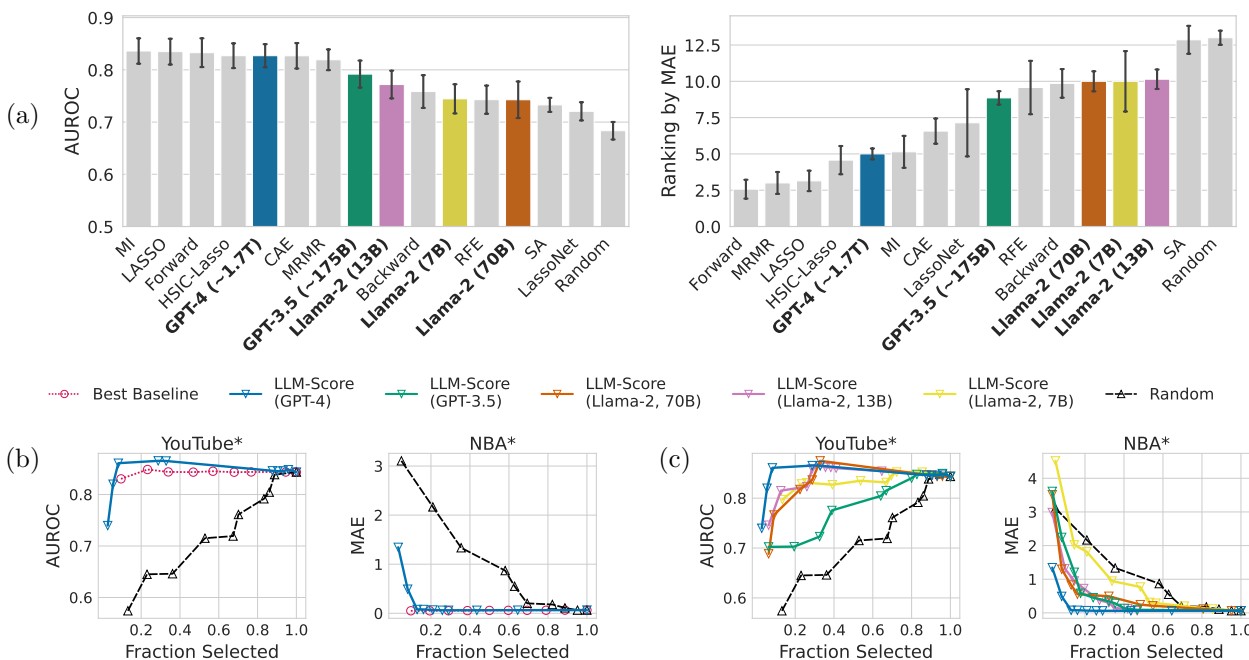

Figure 2: LLM-SCORE shows competitive feature selection performance against data-driven baselines, given an LLM of sufficient scale. (a) Average AUROC (left; higher is better) and ranking by MAE (right; lower is better) across all datasets when selecting the top 30% of features. (b) Feature selection paths for LLM-SCORE (GPT-4), the best-performing baseline, and random selection *on datasets published after the LLM cutoff dates.* (c) Feature selection paths for LLM-SCORE on the same datasets, with varying LLM scale.

For LassoNet and the LASSO, we first compute the regularization paths with warm starts (Friedman et al., 2010) to identify the regularization coefficients corresponding to each feature proportion, and then train a *separate* downstream model as described above. We take this two-step approach to decouple the effects of feature selection and regularization on the downstream test performance. For forward/backward sequential selection, we greedily add/remove a new feature at each iteration based on the 5-fold cross-validation performance resulting from adding/removing each candidate feature. For RFE, we recursively eliminate features with the smallest weights in a logistic/linear regression model selected via a grid search with 5-fold cross-validation using all features. For MI, we select features with the highest marginal mutual information with the target variable based on the training data. When continuous features and/or labels are present, we use the nearest-neighbor approximations (Kraskov et al., 2004; Ross, 2014) available in `scikit-learn` to estimate the empirical mutual information. For HSIC-Lasso, we compute the full kernel matrix if the number of training examples is less than 1000, but otherwise use the block-wise approximation method by Climente-González et al. (2019), using the recommended block size $B = 20$ and number of permutations $M = 3$. For CAE, we use a multi-layer perceptron (MLP) decoder[4] with hidden layers of width 256 and train the model for a maximum of 1000 epochs with a supervised learning objective (as in Appendix F of Balın et al. (2019)). For SA, we use an MLP with 1 hidden layer of width 67 and ReLU activation, following Yasuda et al. (2023). For all experiments, we repeat and average the results over 5 random seeds, which control the train-validation splits used for cross-validation and the behavior of random feature selection, to ensure the robustness of results. We include the remaining details in Appendix A.2.

**Result 1: LLM-based feature selection methods achieve strong performance competitive with data-driven baselines, with sufficient LLM scale (Figure 2).** Figure 2(a) shows the downstream test performance, averaged over all of the classification (left) and regression (right) datasets, when selecting the top 30% of features according to each baseline and LLM-SCORE based on GPT-4, GPT-3.5 and Llama-2.

---

[4]We use the architecture in the official repository for CAE: https://github.com/mfbalin/Concrete-Autoencoders.

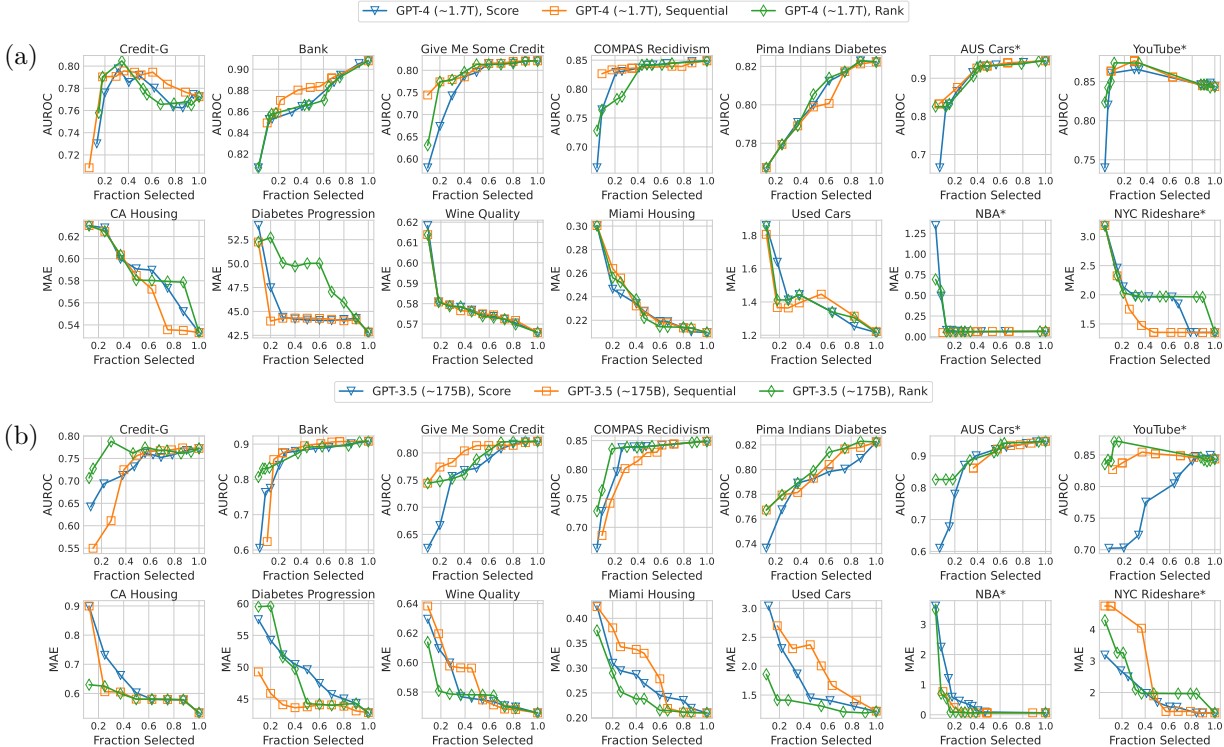

Figure 3: Feature selection paths for LLM-SCORE, LLM-RANK, and LLM-SEQ based on (a) GPT-4 and (b) GPT-3.5 on all classification and regression datasets. Within each panel, the top row shows the results on the classification datasets, and the bottom row shows the results on the regression datasets. GPT-4-based methods all show consistently strong performance across datasets, showing substantial overlap in their corresponding feature selection paths. GPT-3.5-based methods also show similar trends, which are albeit less pronounced. *Datasets marked with an asterisk (*) were published after the LLM cutoff dates.*

For the regression datasets, we report the average *ranking* based on test MAE to account for differences in the scale of MAE values across datasets. For the larger models, notably GPT-4 (in blue) and GPT-3.5 (in green), selecting features based on LLM-generated importance scores leads to strong downstream performance on average, competitive with the data-driven baselines. We observe similar results for LLM-RANK and LLM-SEQ (Figure A1). As an example, in Figure 2(b), we show the *feature selection paths* (i.e., "test performance vs. fraction of features selected" curves) for the best-performing baseline on each dataset, LLM-SCORE based on GPT-4, and random selection on the YouTube* and NBA* datasets published after the LLM cutoff dates. Here, GPT-4-based LLM-SCORE outperforms the best baseline (LASSO) on the YouTube* dataset, achieving higher AUROC overall, and performs as strongly as that (HSIC-Lasso) on the NBA* dataset. Meanwhile, performance is less consistent with smaller LLMs (Figure 2(c)). For example, LLM-SCORE based on Llama-2 (7B) (in yellow) performs well on the YouTube* dataset but close to random on the NBA* dataset. These results demonstrate that using LLMs for feature selection can be effective, but suggest that a sufficiently large model may be required for reliable performance. We include the full results on comparing LLM-SCORE, LLM-RANK, and LLM-SEQ to the data-driven baselines in Appendix A.3.1.

**Result 2: All three LLM-based feature selection methods achieve similarly strong performance (Figure 3).** The feature selection paths for LLM-SCORE, LLM-RANK, and LLM-SEQ based on GPT-4 overlap significantly on all datasets except Diabetes Progression and NYC Rideshare* (Figure 3(a)). Figures 2 and 3(a) together illustrate that GPT-4 achieves consistently strong feature selection performance regardless of the selection mechanism. *Notably, even the simple strategy of querying for an importance score one-feature-at-a-time can be just as effective as those that account for the other features.* We overall observe a similar trend in the feature selection paths for GPT-3.5 (Figure 3(b)), which is albeit less pronounced.

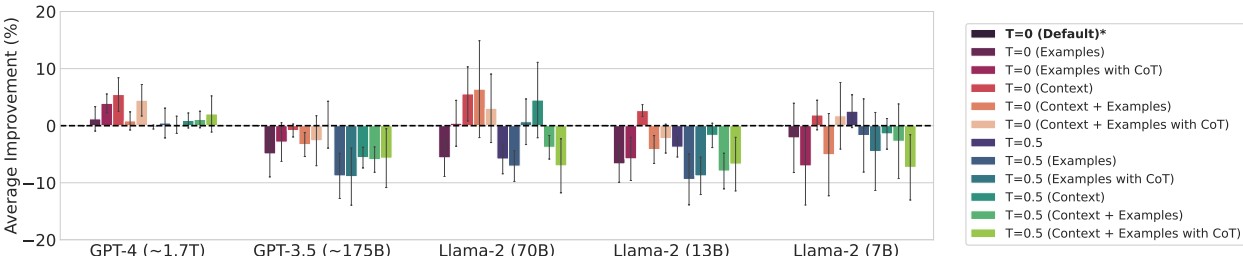

Figure 4: Changes in average improvement (%) in LLM-Score feature selection performance as we vary the decoding strategy ($T = 0$: greedy, $T = 0.5$: self-consistency) and prompt design (in parentheses), compared to the performance achieved under the default prompting setup (in bold; see Section 4). Error bars indicate standard error across datasets. On average, no approach substantially improves over the default setting.

Meanwhile, for the smaller Llama-2 models, the performances of LLM-Score, LLM-Rank, and LLM-Seq are less consistent across datasets (Figures A3–A5), and no single method necessarily outperforms the others. These results suggest that while all three LLM-based feature selection methods can be similarly effective, their effectiveness varies more significantly across datasets as the model size decreases.

**Result 3: Zero-shot prompting with no dataset-specific context and greedy decoding results in strong feature selection performance (Figure 4).** To assess the sensitivity of LLM-Score to prompt design and decoding strategy variations, we evaluate the downstream test performance across six prompt designs ("prompt design" in Section 4) and two decoding strategies (greedy/self-consistency). We use the change in the area under the feature selection paths to measure the impact of each variation. For classification tasks, an increase in the area indicates improved performance, as it suggests that fewer features are needed to achieve high AUROC. Conversely, for regression tasks, a decrease in the area indicates improvement. Thus, we quantify improvement in feature selection performance by computing the % *increase* for the classification tasks and the % *decrease* for the regression tasks. Figure 4 shows the average improvement for each (prompt design, decoding strategy) pair—with respect to that achieved with the default prompt and greedy decoding ($T = 0$). For the settings where we add examples, we only consider a one-shot setting given the relatively small number of features in the datasets. We find that no prompting strategy consistently improves feature selection performance across all LLMs, sometimes even degrading it (e.g., GPT-3.5, Llama-2 (13B)). Meanwhile, we find that the largest and most capable GPT-4 is substantially less sensitive to the prompting strategy and generally benefits from additional context, albeit to a limited extent. These observations suggest that zero-shot prompting without dataset-specific context, combined with greedy decoding, is a strong baseline prompting strategy for LLM-based feature selection.

**Result 4: LLM-Score exhibits higher correlation with widely used feature importance metrics as model scale increases (Figure 5).** To probe the semantics of LLM feature importance metrics, we measure the alignment between LLM-Score and the following feature importance metrics:

1. SHAP (Lundberg & Lee, 2017),
2. Fisher score (Duda et al., 2001),
3. mutual information,
4. Pearson correlation,
5. Spearman correlation,
6. permutation importance (Breiman, 2001).

We compute the Kendall's $\tau$ coefficient $\in [-1, 1]$ (Kendall, 1938) to quantify the agreement in pairwise orderings, where +1/-1 indicates perfect agreement/disagreement. For SHAP, we compute the mean absolute Shapley value for each feature using all test samples, after training an XGBoost model (Chen & Guestrin, 2016). For permutation importance, we measure the average drop in test performance of an $L_2$-penalized

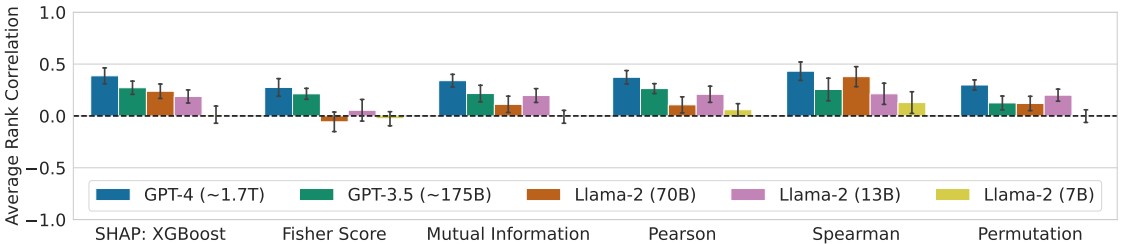

Figure 5: Average rank correlation (Kendall's $\tau$) between each feature importance metric and LLM-Score based on GPT-4, GPT-3.5, and Llama-2. Error bars indicate standard error across datasets. LLM-Score generally exhibits higher rank correlation with standard feature importance metrics as model scale increases, but does not uniquely align to a specific notion of feature importance.

logistic/linear regression model when randomly shuffling the values of each feature 30 times. In Figure 5, we observe that there is no specific notion of importance that LLM-Score consistently aligns to. Meanwhile, as model scale increases, the LLM-generated scores generally exhibit higher correlation with the importance metrics considered (e.g., SHAP, Spearman). An interesting future research direction may be to investigate prompting strategies that increase the alignment of LLM-Score to desired notions of importance. We include additional results on investigating the semantics of LLM-Score in Appendix A.3.7.

## 4.2 Evaluation on Large-Scale Datasets

We show that LLM-based feature selection also achieves strong performance on the more complex large-scale, higher dimensional datasets, each with ~3000 features after preprocessing. Here, we focus on GPT-4-based LLM-Score with the default prompting setup, given its consistently strong performance on small-scale datasets and scalability to a large number of features. We construct supersets of the `Income`, `Employment`, `Public Coverage`, and `Mobility` datasets from `folktables` (Ding et al., 2021) by extracting *all* features available from the 2018 American Community Survey data for California while removing features that lead to label leakage. We also *manually* extract three datasets from the `MIMIC-IV` database (Johnson et al., 2023) for classifying whether an ICU patient was diagnosed with chronic kidney disease (`CKD`), chronic obstructive pulmonary disease (`COPD`), and heart failure (`HF`). Importantly, we note that `MIMIC-IV` is *not publicly available* (access requires special credentials via PhysioNet (Goldberger et al., 2000)) and that the datasets we manually derived are *not* based on existing data preprocessing pipelines (e.g., MIMIC-Extract (Wang et al., 2020), FIDDLE (Tang et al., 2020)). As such, these exact datasets were not part of the LLM pretraining corpora. We provide the remaining dataset details in Appendix B.1.

We evaluate each feature selection method by measuring how the test performance of a downstream prediction model changes when we select the top 10%, 30%, 50%, 70% or 100% of all input *concepts*. We train the following models for downstream prediction: LightGBM (Ke et al., 2017), MLP, and $L_2$-penalized logistic regression. For each feature selection method and dataset, we perform model selection via a random search with 40 hyperparameter samples. We average the test AUROC over 5 random seeds, which control the train-validation splits, the hyperparameter samples, and the initialization of model parameters.

**Baselines.** We compare LLM-Score based on GPT-4 against the following feature selection baselines:

1. LassoNet (Lemhadri et al., 2021),
2. the LASSO (Tibshirani, 1996),
3. minimum redundancy maximum relevance (MRMR; Ding & Peng, 2005),
4. HSIC-Lasso (Yamada et al., 2014),
5. Concrete Autoencoder (CAE; Balın et al., 2019),
6. random feature selection.

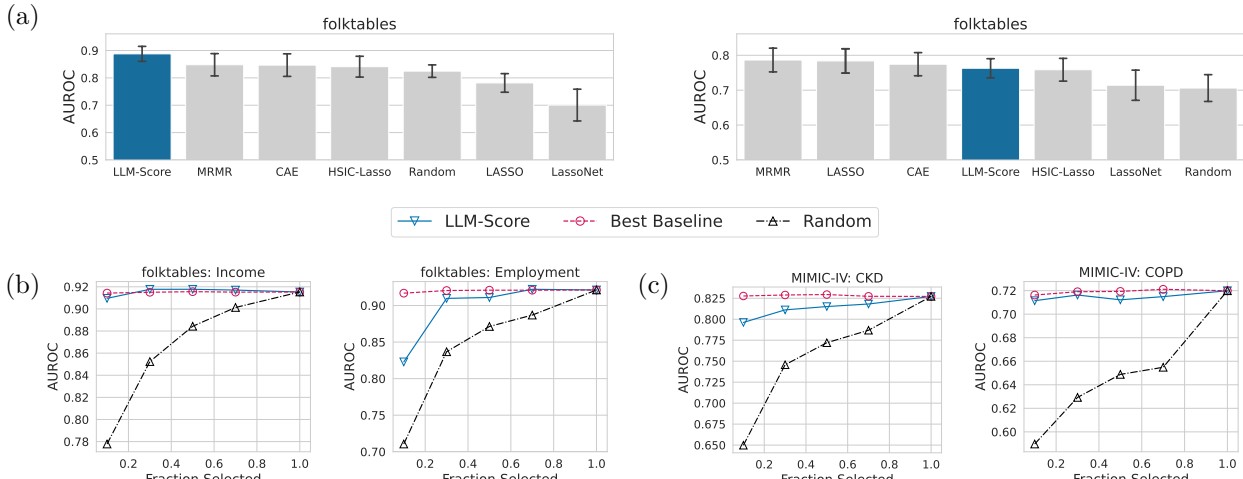

Figure 6: LLM-SCORE based on GPT-4 shows competitive feature selection performance against data-driven baselines on the larger `folktables` and `MIMIC-IV` datasets, each with ∼3000 features. For all plots, we show the results with LightGBM as the prediction model. (a) Average AUROC across all `folktables` (left) and `MIMIC-IV` (right) datasets and downstream prediction models when selecting the top 30% of features. Error bars indicate the standard error across datasets. (b) Feature selection paths for LLM-SCORE, the best-performing baseline, and random selection on the `Income` and `Employment` datasets (`folktables`). (c) Feature selection paths for the same feature selection methods on the `CKD` and `COPD` datasets (`MIMIC-IV`).

Unlike for the small-scale dataset experiments, we exclude forward sequential selection, backward sequential selection, recursive feature elimination (Guyon et al., 2002), and Sequential Attention (Yasuda et al., 2023), given the high computational cost of training $\mathcal{O}(kd)$ separate models when selecting $k$ out of $d$ candidate features. For LassoNet, the LASSO, and the random feature selection baseline, we set up each method to select features at the *concept* level (e.g., "ethnicity" instead of the one-hot encoded features "ethnicity_Asian" or "ethnicity_Hispanic") to match the behavior of LLM-SCORE (e.g., via group-wise feature sparsity (Yuan & Lin, 2006)). For LassoNet and the LASSO, we compute the regularization paths with warm starts (Friedman et al., 2010) (Figure B1) and use the regularization coefficient that selects about the same number of features as LLM-SCORE in each evaluation setting. For MRMR, HSIC-Lasso, and CAE, we select exactly the same number of features as LLM-SCORE in each setting. We use the same setup for HSIC-Lasso as in Section 4.1 and additionally subsample down to 60000 training examples if the training set size exceeds it to avoid out-of-memory errors (`Employment` and `Public Coverage`; Appendix B.1). For CAE, we adopt the setup in Section 4.1 but increase the hidden layer widths for the MLP decoder from 256 to 512. We provide the remaining details in Appendix B.2.

**Result.** Figure 6(a) shows the downstream test AUROC, averaged over all of the `folktables` (left) and `MIMIC-IV` (right) datasets, when (i) selecting the top 30% of features according to each baseline and LLM-SCORE based on GPT-4 and (ii) using LightGBM as the downstream prediction model. Here, we show the performance when selecting the top 30% of features, as the test performance roughly saturates to that of using all features when selecting more than 30% of features for all feature selection methods (Figure B2(a)). We observe that GPT-4-based LLM-SCORE overall performs the best on the `folktables` datasets, and performs on par with HSIC-Lasso and significantly better than LassoNet and the random feature selection baseline on the `MIMIC-IV` datasets. In Figures 6(b–c), we show the full feature selection paths for the best-performing baseline (MRMR), LLM-SCORE, and random feature selection on the `Income` and `Employment` datasets from `folktables` and the `CKD` and `COPD` datasets from `MIMIC-IV`. We observe that for most of these datasets, only a small subset of features are highly predictive of the target outcome, as indicated by the high test AUROC of the best-performing baseline and the low test AUROC of the random feature selection baseline at the 10% and 30% marks. We find that LLM-SCORE is effective at selecting these highly predictive features, with performance rivaling that of the best-performing data-driven baseline and

substantially better than random when selecting the top 10% and 30% of features. Notably, the strong performance on the `MIMIC-IV` datasets suggests that *LLM-SCORE can be effective even in highly specialized domains like healthcare* where, without access to sufficient training data, substantial domain knowledge would be required for feature selection (Section B.1.2). We include the full results for all downstream prediction models on all datasets in Figure B2 in Appendix B.3.

## 5 Discussion and Conclusion

In this work, we demonstrated that LLMs are capable of performing feature selection for supervised learning tasks, *even without access to the downstream training data*. We proposed three different approaches to prompting LLMs for feature selection—LLM-SCORE, LLM-RANK, and LLM-SEQ (Section 3). We found that with sufficient LLM scale, even zero-shot prompting an LLM to select features can result in strong downstream predictive performance, often competitive with data-driven methods such as the LASSO (Tibshirani, 1996) (Results 1 & 3, Section 4.1). For the latest models such as GPT-4, we found that all three LLM-driven methods achieve similarly strong performance and that even the simple strategy of querying for an importance score one-feature-at-a-time (LLM-SCORE) can be as effective as those that account for other available features (Result 2, Section 4.1; Section 4.2). In particular, while the LLM-generated scores do not necessarily align to a specific notion of feature importance (e.g., Fisher score, mutual information), we found that their rank correlations tend to be higher as model scale increases (Result 4, Section 4.1), suggesting that the most capable LLMs are able to translate the real-world relationships encoded in their parameters into statistically meaningful scores. Our findings suggest that LLMs may be useful not only for selecting the best features post data collection, but also for **deciding what features to collect in the first place**. This could potentially benefit practitioners in domains like healthcare and the social sciences, where collecting high-quality data can be expensive and time-consuming.

**Limitations.** First, while we have demonstrated the effectiveness of LLM-based feature selection methods in various domains (e.g., finance, healthcare, criminal justice), it is possible that they show limited performance in extremely specialized or rare domains. Second, as LLM-based feature selection methods rely on textual semantics (e.g., column names in tabular datasets) to identify predictive features, they may not be effective on (i) datasets without sufficient text annotations (Appendix A.3.6) or (ii) datasets where the statistical relationships between features and targets do not align with the semantic associations inferred by the LLM (due to e.g., selection bias). Third, LLMs may exhibit undesirable biases inherited from their pretraining data (Gallegos et al., 2024), which can result in downstream performance disparities across data subpopulations. In the data-driven setting, it is possible to mitigate such issues by selecting features for each subpopulation independently or modifying the training objective to account for group fairness (Zemel et al., 2013; Sagawa et al., 2020; Izmailov et al., 2022). However, it is not immediately obvious how to incorporate similar notions of group fairness into LLM-based feature selection methods, especially for closed-source LLMs which can only be accessed via prompting. As such, combining LLM-driven feature selection with data-driven methods or using it in a human-in-the-loop setup may be a more reliable approach for mitigating bias concerns, especially in safety-critical domains. We leave an in-depth investigation of methods for incorporating group fairness to LLM-based feature selection for future work.

### Acknowledgements

We gratefully acknowledge DARPA (FA8750-23-2-1015), ONR (N00014-23-1-2368), NSF (IIS2211955), UPMC, Highmark Health, Abridge, Ford Research, Mozilla, the PwC Center, Amazon AI, JP Morgan Chase, the Block Center, the Center for Machine Learning and Health, and the CMU Software Engineering Institute (SEI) via Department of Defense contract FA8702-15-D-0002, for their generous support of our research. We also thank Michael Oberst, Goutham Rajendran, Bingbin Liu, and Che-Ping Tsai for helpful feedback and comments on earlier versions of this work.

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

# Appendix for LLM-Select: Feature Selection with Large Language Models

## Table of Contents

# A Additional Details on Small-Scale Dataset Experiments

Here, we provide additional details and results for the small-scale dataset experiments in Section 4.1.

## A.1 Datasets

For all datasets, we report both the total number of features after preprocessing and the total number of concepts (i.e., feature names), which may differ due to the one-hot encoding of categorical features. For example, the `Credit-G` dataset contains 61 features after one-hot encoding the categorical features and only 20 concepts. In this case, when feature selection is performed at the concept level, we select $k$ of the 20 concepts; when feature selection is performed at the feature level, we select $k$ of the 61 preprocessed features.

For each dataset, we randomly shuffle and take a 80–20 train–test split. We then take a 5-fold split of the training set for cross-validation, where the 5-fold splits vary across the random seeds ($=[1,2,3,4,5]$) used throughout the experiments. The test set remains fixed and does not vary with the random seed used. For classification datasets, we always take a stratified split to preserve the label proportions across the train, validation, and test sets. We standardize all of the numerical features to have zero mean and unit variance, and one-hot encode all categorical features. Below, we provide the remaining details for each dataset.

**Classification Datasets:**

- `Credit-G` (Hofmann, 1994) is a UCI dataset[5], where the goal is to predict whether a client at a bank carries high credit risk, given a set of attributes about the client (e.g., credit history, savings account status). The dataset contains 1000 samples and 61 features after preprocessing (20 concepts), with 700 positive samples and 300 negative samples.

- `Bank` (Moro et al., 2012) is a UCI dataset[6], where the goal is to predict whether a client at a bank will subscribe to a term deposit, given data collected from a telemarketing campaign at a Portuguese banking institution from 2008 to 2013. The dataset contains 45211 samples and 51 features after preprocessing (16 concepts), with 5289 positive samples and 39922 negative samples.

- `Give Me Some Credit` is a Kaggle dataset[7], where the goal is to predict whether an individual is likely to experience significant financial distress/delinquency within the next two years, given information about the individual's financial status (e.g., debt ratio, monthly income). The dataset contains 120269 samples and 10 features after preprocessing (10 concepts), with 8357 positive samples and 111912 negative samples.

- `COMPAS Recidivism` is a dataset collected from a 2016 study[8] on the racial biases present in the Correctional Offender Management Profiling for Alternative Sanctions (COMPAS) algorithm (Larson et al., 2016). The goal is to predict whether a criminal defendant carries high risk of recidivism, given their criminal history and demographic attributes. The dataset contains 6172 samples and 23 features after preprocessing (14 concepts), with 2751 positive samples and 3421 negative samples.

- `Pima Indians Diabetes` (Smith et al., 1988) is a Kaggle dataset[9], where the goal is to predict whether a female adult patient of Pima Indian heritage has diabetes, given a set of clinical measurements and demographics. The dataset contains 768 samples and 8 features after preprocessing (8 concepts), with 268 positive samples and 500 negative samples.

- `AUS Cars`* is a Kaggle dataset[10] published in 2023, where the goal is to predict whether the price of a car in Australia is above \$30000, given a set of attributes about the car (e.g., year of manufacture, number of doors in the car, fuel consumption rate). The dataset contains 14283 samples and 33 features after preprocessing (11 concepts), with 6396 positive samples and 7887 negative samples.

---

[5] https://archive.ics.uci.edu/dataset/144/statlog+german+credit+data
[6] https://archive.ics.uci.edu/dataset/222/bank+marketing
[7] https://www.kaggle.com/c/GiveMeSomeCredit
[8] https://www.propublica.org/article/how-we-analyzed-the-compas-recidivism-algorithm
[9] https://www.kaggle.com/datasets/uciml/pima-indians-diabetes-database
[10] https://www.kaggle.com/datasets/nelgiriyewithana/australian-vehicle-prices

- `YouTube`* is a Kaggle dataset[11] published in 2023, where the goal is to predict whether YouTube channel has more than 20 million subscribers, given a set of attributes about the channel (e.g., total number of videos uploaded on the channel, date when the channel was created). The dataset contains 588 samples and 72 features after preprocessing (22 concepts), with 268 positive samples and 320 negative samples.

**Regression Datasets:**

- `CA Housing` (Pace & Barry, 1997) is a StatLib dataset[12], where the goal is to predict the median housing price of a US Census block group in California, given data collected from the 1990 US Census. A block group, typically a population of 600 to 3000 people, is the smallest geographical unit for which the US Census Bureau publishes sample data. The dataset contains 20640 samples and 8 features after preprocessing (8 concepts).
- `Diabetes Progression` is a dataset[13] used in a study by Efron et al. (2004), where the goal is to predict the disease progression level in diabetic patients, given their baseline blood serum measurements from the previous year and demographic information. The dataset contains 442 samples and 10 features after preprocessing (10 concepts).
- `Wine Quality` (Cortez et al., 2009) is a UCI dataset[14] collected from red and white vinho verde wine samples from northern Portugal, where the goal is to predict whether a wine is high or low quality, given its various physicochemical measurements (e.g., acidity, density). The dataset contains 6497 samples and 11 features after preprocessing (11 concepts).
- `Miami Housing` (Grinsztajn et al., 2022) is an OpenML dataset[15], where the goal is to predict the selling price of a house in Miami in 2016, given structural (e.g., area, structure quality) and geographic (e.g., longitude, latitude) information about each house. The dataset contains 13932 samples and 15 features after preprocessing (15 concepts).
- `Used Cars` is a Kaggle dataset[16], where the goal is to predict the selling price of a used car, given a set of attributes about each car (e.g., age, fuel type, number of previous owners). The dataset contains 301 samples and 11 features after preprocessing (7 concepts).
- `NBA`* is a Kaggle dataset[17] published in 2023, where the goal is to predict the number of points per game for an NBA basketball player, given a set of attributes and statistics (from the 2023–2024 NBA season) about the player (e.g., blocks per game, position, team). The dataset contains 388 samples and 61 features after preprocessing (27 concepts).
- `NYC Rideshare`* is a Kaggle dataset[18] published in 2023, where the goal is to predict the total pay given to a rideshare-app driver in NYC after a trip, given information about the trip (e.g., time elapsed from ride request to dropoff, total distance of the trip). The original dataset on Kaggle contains some data from 2022, so we only extract the data rideshare data from 2023 for preprocessing and evaluation. The dataset contains 5000 samples and 19 features after preprocessing (15 concepts).

## A.2 Feature Selection, Model Training, and Hyperparameter Optimization

As described in Section 4.1 of the main text, we evaluate the effectiveness of each feature selection method by measuring how the test performance of a *downstream* $L_2$-penalized logistic regression (for classification tasks) or linear regression (for regression tasks) model changes as we vary the proportion of features selected from 10% to 100%, in approximately 10% increments. For logistic regression, we minimize the negative log-likelihood using the L-BFGS optimizer (Zhu et al., 1997) and use importance weighting to balance the

---

[11]https://www.kaggle.com/datasets/nelgiriyewithana/global-youtube-statistics-2023
[12]https://www.dcc.fc.up.pt/~ltorgo/Regression/cal_housing.html
[13]https://www4.stat.ncsu.edu/~boos/var.select/diabetes.html
[14]https://archive.ics.uci.edu/dataset/186/wine+quality
[15]https://www.openml.org/search?type=data&sort=runs&id=43093&status=active
[16]https://www.kaggle.com/datasets/vijayaadithyanvg/car-price-predictionused-cars
[17]https://www.kaggle.com/datasets/bryanchungweather/nba-player-stats-dataset-for-the-2023-2024
[18]https://www.kaggle.com/datasets/aaronweymouth/nyc-rideshare-raw-data

weights of the positive and negative samples, as some of the datasets exhibit label imbalance. For linear regression, we minimize the mean squared error (MSE) using the L-BFGS optimizer. For all experiments, we aggregate the results over 5 random seeds ($=[1,2,3,4,5]$), which control the train-validation splits and the behavior of random feature selection, in order to ensure the robustness of results.

### A.2.1    Additional Details on Feature Selection Methods

**LassoNet.** For feature selection with LassoNet (Lemhadri et al., 2021), we use a multi-layer perceptron (MLP) with 1 hidden layer and 100 hidden units for all datasets and fix the hierarchy coefficient $M$ to the recommended value of 10. For a given dataset, we take a 80-20 split of the training set (split controlled by the random seed $= [1, 2, 3, 4, 5]$), using the latter 20% split for validation. For classification tasks, we minimize the binary cross-entropy loss using the Adam optimizer (Kingma & Ba, 2015) with the default learning rate of $10^{-3}$ and the default momentum hyperparameters of $\beta_1 = 0.9, \beta_2 = 0.999$, and $\epsilon = 10^{-8}$. For regression tasks, we minimize the MSE using the Adam optimizer with the same configuration used for classification tasks. Following Lemhadri et al. (2021), we compute the *dense-to-sparse* regularization paths with warm starts (Friedman et al., 2010). We first train an unregularized MLP for 10 epochs, using the validation split for early stopping with a patience of 3 epochs. We then increase the regularization coefficient starting from $\lambda = 10^{-6}$, iteratively multiplying powers of 1.02 until all of features become inactive (i.e., $\lambda = 10^{-6}, 1.02 \cdot 10^{-6}, 1.02^2 \cdot 10^{-6}, \ldots$), where for each value of $\lambda$, we further train the model for 3 epochs. After computing the full regularization path, we identify the regularization strengths that select 10% to 100% of all features in approximately 10% increments.

**LASSO.** For feature selection with the LASSO (Tibshirani, 1996) on each dataset, we first train an $L_1$-penalized logistic regression model with inverse regularization coefficient $C = 10^{-4}$ (for classification tasks) or an $L_1$-penalized linear regression model with regularization coefficient $\lambda = 10^4$ (for regression tasks)[19]. To compute the *sparse-to-dense* regularization paths with warm starts (Friedman et al., 2010), we then gradually increase $C$ by iteratively multiplying powers of 1.02 (i.e., $C = 10^{-4}, 1.02 \cdot 10^{-4}, 1.02^2 \cdot 10^{-4}, \ldots$) and gradually decrease $\lambda$ by iteratively multiplying powers of $\frac{1}{1.02}$ (i.e., $\lambda = 10^4, 1.02^{-1} \cdot 10^4, 1.02^{-2} \cdot 10^4, \ldots$) until all features become active. For logistic regression, we train each model by minimizing the negative log-likelihood using the SAGA optimizer (Defazio et al., 2014). For linear regression, we train each model by minimizing the MSE using the SAGA optimizer. After computing the full regularization path, we identify the regularization strengths that select 10% to 100% of all features in approximately 10% increments.

**Forward/backward sequential selection.** For forward or backward sequential selection, we first run a grid search with 5-fold cross-validation to select the best hyperparameter to use for training an $L_2$-penalized logistic/linear regression model for sequential selection. For classification tasks, we sweep through the inverse regularization coefficients $C = [0.1, 0.5, 1, 5, 10, 50, 100]$ and select the value of $C$ that leads to the highest cross-validation area under the ROC curve (AUROC). For regression tasks, we sweep through the regularization coefficients $\lambda = [0.001, 0.005, 0.01, 0.05, 0.1, 0.5, 1, 5, 10]$ and select the value of $\lambda$ that leads to the lowest cross-validation mean absolute error (MAE). We use the selected hyperparameter value for training a model at each iteration of forward or backward sequential selection. For forward sequential selection, we start with an empty feature subset and iteratively add a new feature which maximizes the cross-validation performance when included in the feature subset. For backward sequential selection, we start with all features and iteratively remove a feature such that the cross-validation performance degrades minimally when excluding it from the feature subset.

**Random feature selection.** For random feature selection, we randomly select the input *concepts* in approximately 10% increments, as in LLM-based feature selection. For example, on the `Credit-G` dataset, which contains a total of 20 concepts (61 features after preprocessing), selecting 10% via random feature selection is equivalent to selecting 2 out of the 20 concepts, which may correspond to more than $6 \approx 61 \cdot 0.1$ features. The randomness in the feature subset is controlled by the random seeds ($=[1,2,3,4,5]$) used throughout the experiments.

---

[19]For logistic regression, we configure the *inverse* regularization coefficient $C$ instead of the regularization coefficient $\lambda$, as the `scikit-learn` implementation of logistic regression uses the former.

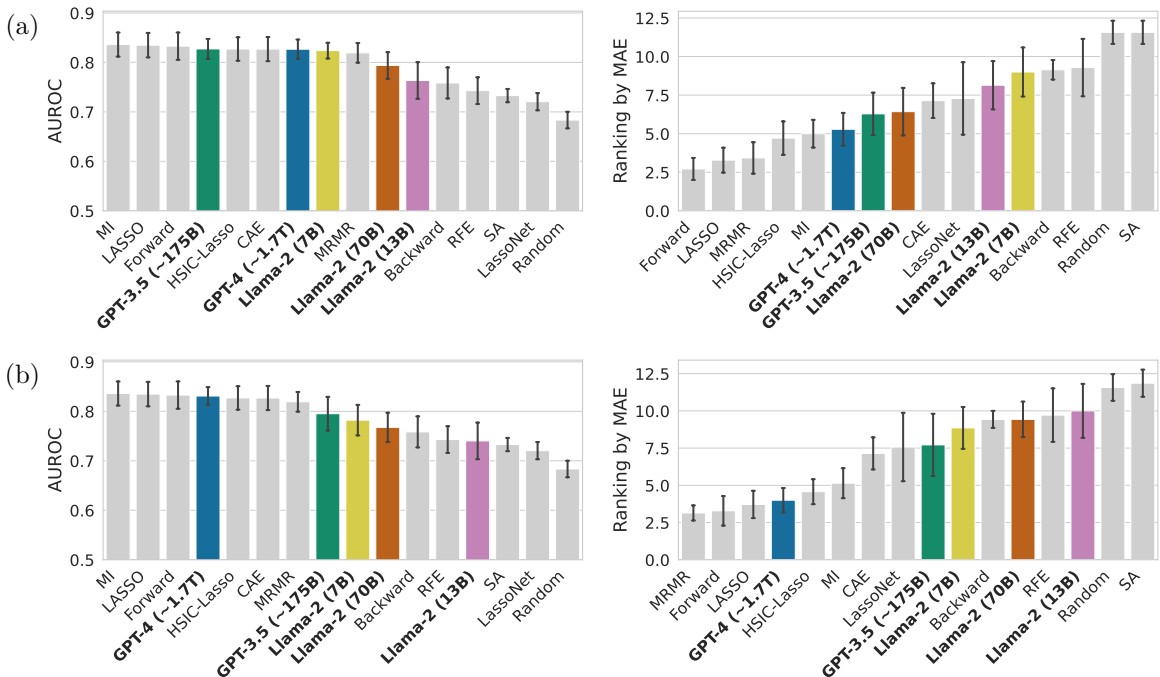

Figure A1: LLM-RANK and LLM-SEQ achieve strong feature selection performance competitive with data-driven baselines, given an LLM of sufficient scale. (a) Average AUROC (left; higher is better) and ranking by MAE (right; lower is better) across all datasets when selecting the top 30% of features with LLM-RANK. (b) Average AUROC (left; higher is better) and ranking by MAE (right; lower is better) across all datasets when selecting the top 30% of features with LLM-SEQ.

### A.2.2 Downstream Model Training and Hyperparameter Optimization

For each dataset and at each proportion of features selected by each method, we run a grid search with 5-fold cross-validation to select the best hyperparameter to use for training and evaluation of the downstream prediction model. For classification tasks, we sweep through the inverse regularization coefficients $C = [0.1, 0.5, 1, 5, 10, 50, 100]$ and select the hyperparameter value that leads to the highest cross-validation AUROC. For regression tasks, we sweep through the regularization coefficients $\lambda = [0.001, 0.005, 0.01, 0.05, 0.1, 0.5, 1, 5, 10]$ and select the hyperparameter value that leads to the lowest cross-validation MAE. We then train a final logistic/linear regression model with the selected regularization strength using the full training set, and measure the performance of the final model on the test set. We repeat this process 5 times with different random seeds (=[1,2,3,4,5]) and average the results.

### A.3 Additional Experimental Results

### A.3.1 Comparison of LLM-SCORE, LLM-RANK, and LLM-SEQ to Baselines

In this section, we include additional experimental results that compare the performances of LLM-SCORE, LLM-RANK, and LLM-SEQ against those of the baselines described in Section 4.1 of the main text. We note that all of the results shown in this section are based on the LLM outputs generated from the *default prompt*, which does not include any dataset-specific context or few-shot examples, and with *greedy decoding*. For LLM-SEQ, we present the results for the setup where we start with an empty feature subset. As described in Section 4.1 of the main text, we evaluate the effectiveness of each feature selection method by measuring how the test performance of a *downstream* model changes as we vary the proportion of features selected from 10% to 100%, in approximately 10% increments. Intuitively, an effective feature selection method should be able to identify a highly informative subset of features and enable strong downstream predictive performance,

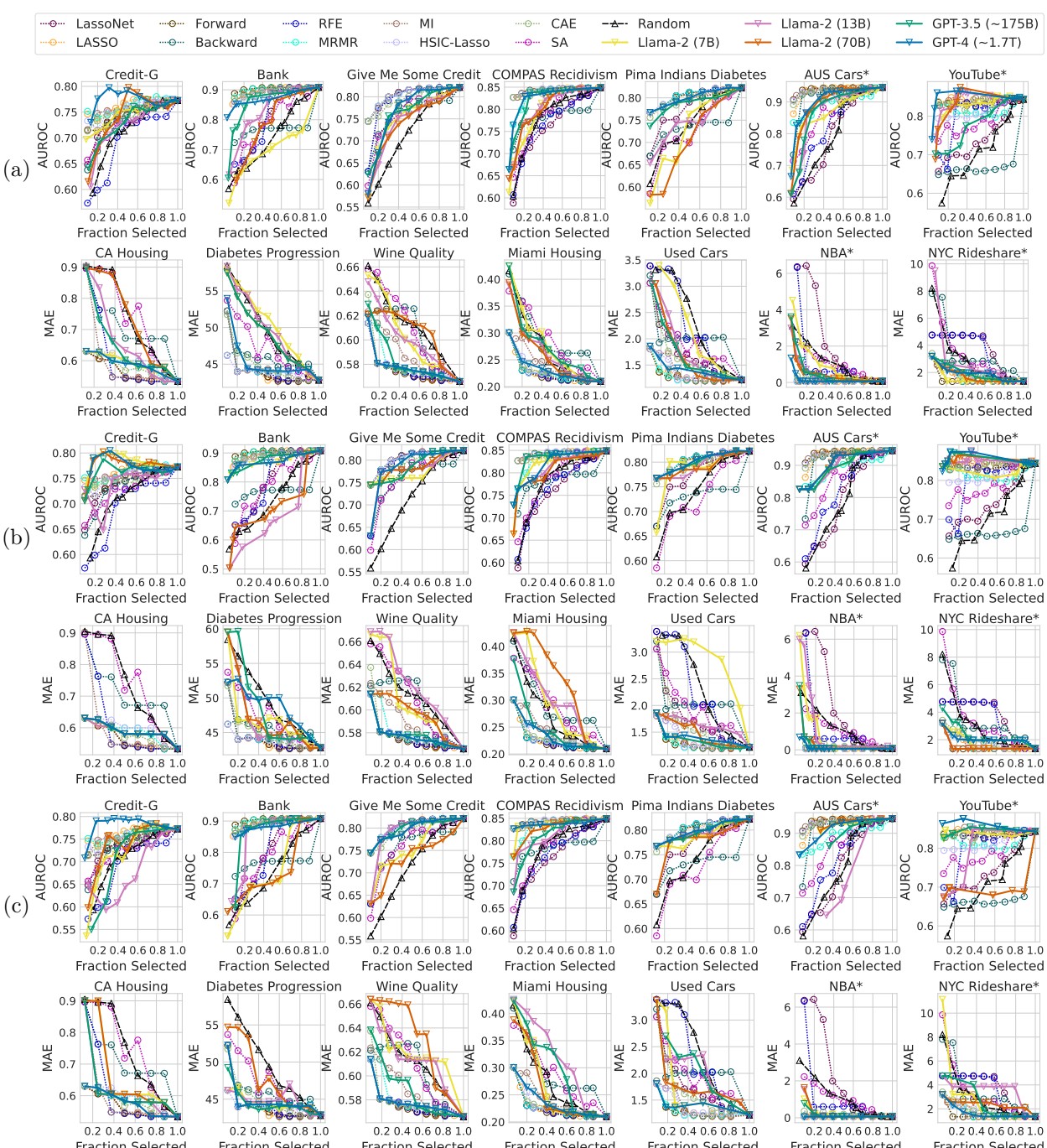

Figure A2: Feature selection paths for all baselines and the (a) LLM-Score, (b) LLM-Rank, and (c) LLM-Seq methods (ours, in solid lines) based on GPT-4, GPT-3.5, and the Llama-2 models. In each panel, the top row shows the test AUROC (higher is better) on the classification datasets and the bottom row shows the test MAE (lower is better) on the regression datasets. *Datasets marked with an asterisk (\*) were published after the LLM cutoff dates.* On most datasets, LLM-based feature selection methods show strong performance comparable to the baselines. Larger models, especially GPT-4, show consistently strong performance across all datasets.

even when the number of selected features is relatively small. For all experiments, the results are averaged over 5 random seeds (=[1,2,3,4,5]), which control the train-validation splits and random feature selection.

Figure A1 shows the downstream test performance, averaged over all of the classification (left) and regression (right) datasets, when selecting the top 30% of features according to each baseline, and (a) LLM-RANK and (b) LLM-SEQ, based on GPT-4, GPT-3.5, and Llama-2. For the regression datasets, we report the average *ranking* based on test MAE to account for differences in the scale of MAE values across datasets. We observe that LLM-RANK and LLM-SEQ both show strong feature selection performance on average, competitive with the data-driven baselines.

Figure A2 shows the feature selection paths for the baselines and for LLM-SCORE, LLM-RANK, and LLM-SEQ based on GPT-4, GPT-3.5, and the Llama-2 models. Table A1 shows the area under the feature selection paths in Figure A2 for the baselines and for LLM-SCORE, LLM-RANK, and LLM-SEQ based on GPT-4, GPT-3.5, and the Llama-2 models. As discussed in Result 3, Section 4.1 of the main text, the area under each feature selection path serves as a quantitative summary of feature selection performance, where a higher area is desirable for classification tasks and a lower area is desirable for regression tasks. For each dataset, we highlight in bold the area of the best-performing baseline and the area of the best-performing LLM-based feature selection method for each LLM.

In Figure A2, we observe that LLM-SCORE, LLM-RANK, and LLM-SEQ all achieve strong feature selection performance competitive with the data-driven baselines. For the larger models, notably GPT-4 (in blue) and GPT-3.5 (in green), we consistently observe such strong performance across many datasets. For the smaller models, notably the Llama-2 models (in orange, pink, and yellow), we observe that feature selection performance is generally more sensitive to the choice of selection strategy (i.e., LLM-RANK or LLM-SEQ), but that performance can be strong when an adequate selection strategy is used. For example, on the `Credit-G` and `COMPAS Recidivism` datasets, we see that using LLM-RANK significantly improves the performances of GPT-3.5 and the Llama-2 models over that achieved with LLM-SCORE (shown in Figure 2)—from being close to the average performance of random feature selection (in black) to being on par or better than those of the best-performing baselines. For LLM-SEQ, we observe a similar improvement on the `Pima Indians Diabetes` and `COMPAS Recidivism` datasets.

Table A1 further illustrates these findings from Figure A2. We see that for the larger GPT-4 and GPT-3.5 models, the area of the best-performing LLM-based feature selection method is often on par with the area of the best-performing baseline, across most datasets. For example, on the `Credit-G` dataset, we see that the best-performing baseline (MRMR) achieves an area of 0.7496, while the best-performing selection strategy for GPT-4 (LLM-SEQ) outperforms it with an area of 0.7710, and the best-performing selection strategy for GPT-3.5 (LLM-RANK) performs as well with an area of 0.7495. We observe that these models achieve strong performance on the other datasets as well (e.g., `COMPAS Recidivism`, `Pima Indians Diabetes`, `Wine Quality`). Meanwhile, for the smaller Llama-2 models, we see that the area of the best-performing selection strategy can be on par with that of the best-performing baseline but that feature selection performance more sensitive to the choice of selection strategy. For example, on the `Pima Indians Diabetes` dataset, we see that the areas of the best-performing selection strategies for the three Llama-2 models are 0.7728 (70B), 0.7744 (13B), and 0.7709 (7B), respectively, which are on par with the area of 0.7855 for the best-performing baseline (RFE). However, as noted in Result 1, Section 4.1 of the main text, the performance of the smaller models are less consistent across datasets and the three LLM-based feature selection methods.

The above results demonstrate that all three LLM-based feature selection methods can be competitive with traditional baselines but that a sufficiently large model may be required to ensure reliable performance.

### A.3.2 Comparison of LLM-SCORE, LLM-RANK, and LLM-SEQ Across LLMs of Varying Scale

In this section, we include additional results that compare the feature selection behaviors and performances of LLM-SCORE, LLM-RANK, and LLM-SEQ across LLMs of varying scale. We note that all of the results shown in this section are based on the LLM outputs generated from the *default prompt*, which does not include any dataset-specific context or few-shot examples, and with *greedy decoding*. As done for GPT-4 and GPT-3.5 in Result 2, Section 4.1 of the main text (Figure 3), we show the changes in average test performance for LLM-SCORE, LLM-SEQ, and LLM-RANK for GPT-3.5 and the Llama-2 models in Figures A3–A5. In Figure A6, we also show the average rank correlation (Kendall's $\tau$) between (i) LLM-SCORE and LLM-RANK (left) and that between (ii) LLM-SCORE and LLM-SEQ (right) to show how well the orderings

Table A1: Area under the feature selection paths for the baselines and our LLM-Score, LLM-Rank, and LLM-Seq methods based on GPT-4, GPT-3.5, and the Llama-2 models. For classification datasets, higher is better (↑). For regression datasets, lower is better (↓). `Give Me Credit` and `Pima Diabetes` are shorthands for the `Give Me Some Credit` and `Pima Indians Diabetes` datasets, respectively. For each dataset, we boldface the area of the best-performing baseline and that of the best-performing LLM-based method for each LLM. *Datasets marked with an asterisk (\*) were published after the LLM cutoff dates.*

| | Classification Datasets ↑ | | | | | | | Regression Datasets ↓ | | | | | | |
| | Credit-G | Bank | Give Me Credit | COMPAS Recidivism | Pima Diabetes | AUS Cars* | YouTube* | CA Housing | Diabetes Progression | Wine Quality | Miami Housing | Used Cars | NBA* | NYC Rideshare* |
|---|---|---|---|---|---|---|---|---|---|---|---|---|---|---|
| LassoNet | 0.7253 | 0.7915 | **0.7889** | 0.7504 | 0.7564 | 0.7818 | 0.6905 | 0.5721 | 44.6832 | 0.5775 | 0.2288 | 1.8574 | 2.5073 | 3.7539 |
| LASSO | 0.7304 | **0.8819** | 0.7783 | 0.8069 | 0.7829 | 0.9144 | **0.8264** | 0.5726 | 45.0356 | **0.5727** | 0.2308 | **0.8533** | 0.0625 | 1.6211 |
| Forward | 0.7355 | 0.8847 | 0.7884 | 0.8287 | 0.7842 | **0.9204** | 0.8181 | **0.5711** | 44.5965 | 0.5771 | **0.2273** | 1.3148 | 0.0610 | **1.5367** |
| Backward | 0.7201 | 0.7592 | 0.7463 | 0.7722 | 0.7188 | 0.8939 | 0.6634 | 0.7274 | 45.4677 | 0.6053 | 0.2965 | 2.0984 | 0.1074 | 3.6240 |
| MRMR | **0.7496** | 0.8815 | 0.7883 | 0.8199 | 0.7850 | 0.8751 | 0.7997 | 0.5935 | 45.2207 | 0.5821 | 0.2297 | 1.3287 | 0.0645 | 1.7508 |
| MI | 0.7421 | 0.8767 | 0.7882 | 0.8207 | 0.7798 | 0.9119 | 0.8230 | 0.6221 | 45.0259 | 0.5899 | 0.2429 | 1.3927 | 0.0654 | 1.6746 |
| RFE | 0.6817 | 0.7986 | 0.7762 | 0.7568 | **0.7855** | 0.8111 | 0.7884 | 0.6342 | 44.7423 | 0.5780 | 0.2282 | 2.3120 | 1.2952 | 3.4855 |
| HSIC-Lasso | 0.7436 | 0.8712 | 0.7882 | **0.8288** | 0.7826 | 0.9169 | 0.7974 | 0.6005 | **44.1273** | 0.5800 | 0.2348 | 1.3956 | **0.0571** | 1.9681 |
| CAE | 0.7357 | 0.8739 | 0.7790 | 0.8241 | 0.7766 | 0.9150 | 0.8215 | 0.5854 | 45.2773 | 0.5825 | 0.2337 | 1.7302 | 0.2000 | 1.9896 |
| SA | 0.7152 | 0.7821 | 0.7465 | 0.7759 | 0.7120 | 0.8480 | 0.7358 | 0.7590 | 47.5920 | 0.6129 | 0.2905 | 2.0027 | 1.0580 | 3.5182 |
| Random | 0.6965 | 0.7408 | 0.7038 | 0.7686 | 0.7265 | 0.7861 | 0.6948 | 0.7555 | 50.2139 | 0.6152 | 0.2780 | 2.4022 | 1.1800 | 3.1562 |
| GPT-4 (LLM-Score) | 0.7556 | **0.8686** | 0.7504 | 0.8192 | 0.7813 | 0.8874 | 0.8474 | **0.5930** | 45.9081 | 0.5804 | **0.2353** | 1.4363 | 0.1576 | 1.9757 |
| GPT-4 (LLM-Rank) | 0.7570 | **0.8687** | **0.7717** | 0.8132 | **0.7821** | 0.9001 | **0.8547** | 0.5948 | 48.9612 | **0.5795** | 0.2365 | 1.4251 | 0.1209 | 2.0982 |
| GPT-4 (LLM-Seq) | **0.7710** | 0.8641 | 0.7417 | **0.8231** | 0.7794 | **0.9022** | 0.8452 | 0.5834 | **45.3283** | 0.5800 | 0.2382 | **1.4234** | **0.0600** | **1.7038** |
| GPT-3.5 (LLM-Score) | 0.7158 | 0.8562 | 0.7481 | 0.8133 | 0.7693 | 0.8562 | 0.7709 | 0.6682 | 49.6221 | 0.5859 | 0.2806 | 1.8029 | 0.5328 | **1.9832** |
| GPT-3.5 (LLM-Rank) | **0.7495** | **0.8730** | 0.7719 | **0.8243** | **0.7813** | **0.8924** | **0.8508** | **0.5943** | 49.2408 | **0.5802** | **0.2510** | **1.3739** | 0.3101 | 2.3767 |
| GPT-3.5 (LLM-Seq) | 0.6952 | 0.8543 | **0.7838** | 0.7970 | 0.7776 | 0.8334 | 0.8290 | 0.6432 | **44.7747** | 0.5909 | 0.3014 | 2.0378 | **0.1828** | 2.7214 |
| Llama-2-70B (LLM-Score) | 0.7258 | **0.7797** | 0.7259 | 0.7946 | 0.6934 | 0.8760 | 0.8268 | 0.7512 | 49.6221 | 0.6074 | **0.2634** | 1.7945 | 0.5208 | 1.7974 |
| Llama-2-70B (LLM-Rank) | **0.7618** | 0.7158 | **0.7706** | 0.8004 | **0.7728** | **0.9016** | **0.8506** | **0.5941** | **48.1246** | **0.5972** | 0.3377 | **1.4991** | 0.3164 | **1.5433** |
| Llama-2-70B (LLM-Seq) | 0.7263 | 0.7436 | 0.7217 | **0.8138** | 0.7587 | 0.8742 | 0.6865 | 0.6858 | 48.3056 | 0.6312 | 0.2649 | 1.8600 | **0.2952** | 2.3938 |
| Llama-2-13B (LLM-Score) | **0.7092** | 0.8279 | 0.7284 | 0.7953 | 0.7300 | 0.8557 | 0.8297 | 0.6823 | 50.3093 | **0.6036** | **0.2676** | 1.8727 | 0.4452 | 3.0830 |
| Llama-2-13B (LLM-Rank) | 0.6631 | 0.6636 | 0.7603 | 0.7695 | **0.7744** | **0.8936** | **0.8444** | **0.5925** | 47.5913 | 0.6248 | 0.3038 | **1.6483** | 1.0380 | **1.5407** |
| Llama-2-13B (LLM-Seq) | 0.6688 | **0.8470** | **0.7746** | **0.7974** | **0.7744** | 0.7425 | 0.7840 | 0.6374 | **44.9351** | 0.6167 | 0.3207 | 1.9734 | **0.2974** | 3.5624 |
| Llama-2-7B (LLM-Score) | 0.7331 | 0.7033 | 0.7358 | 0.7917 | 0.7132 | 0.8527 | 0.8107 | 0.5952 | 50.9002 | **0.6061** | 0.2833 | 2.2908 | 1.0696 | **1.6089** |
| Llama-2-7B (LLM-Rank) | **0.7671** | **0.8676** | **0.7615** | 0.8115 | 0.7661 | **0.8924** | 0.8185 | **0.5925** | 47.6220 | 0.6117 | 0.3006 | 2.8661 | 0.7897 | 1.7905 |
| Llama-2-7B (LLM-Seq) | 0.7102 | 0.7484 | 0.7434 | **0.8205** | **0.7709** | 0.8691 | **0.8216** | 0.6888 | **44.5974** | 0.6219 | **0.2726** | **1.8047** | **0.2413** | 2.9630 |

of the features by their relevance to the prediction target agree between each pair selection strategies being compared. As described in Result 4, Section 4.1 of the main text, the Kendall's $\tau$ coefficient $\in [-1, 1]$ quantifies the agreement in pairwise orderings, where $+1/\text{-}1$ indicates perfect agreement/disagreement.

Together with Figure 3 discussed in Result 2, Section 4.1 of the main text, Figures A3–A5 show that the feature selection paths of LLM-Score, LLM-Rank, and LLM-Seq are generally more consistent with one another (i.e., exhibits less variability) with increasing model scale, with GPT-4 showing almost perfect alignment on all datasets except `Diabetes Progression`. Figure A6 also shows that the average rank correlation between LLM-Score and LLM-Rank and that between LLM-Score and LLM-Seq both tend to increase with increasing model scale. Table A1 further illustrates this finding, where we see that with increasing model scale, the areas under the feature selection paths for LLM-Score, LLM-Rank, and LLM-Seq tend to be more similar.

The above results suggest that the assignment of feature importance tends to be more consistent across LLM-Score, LLM-Rank, and LLM-Seq for the larger models. Meanwhile, for the smaller models, we see that choosing a different selection strategy leads to qualitatively distinct results, translating to increased variability in the average test performance of the downstream prediction model.

### A.3.3   Comparison of LLM-Score, LLM-Rank, and LLM-Seq to LMPriors (Choi et al., 2022)

In this section, we include additional results that compare the average feature selection performance of LLM-Score, LLM-Rank, and LLM-Seq to that of the LMPriors framework (Choi et al., 2022). As discussed in Section 2.2, *our methods are more broadly applicable than the LMPriors framework*, which requires access to the token probabilities often not provided in state-of-the-art proprietary LLMs such as GPT-4 and GPT-3.5. Meanwhile, the original implementation of LMPriors uses the `davinci-instruct-beta` variant of GPT-3 (Brown et al., 2020), which is no longer serviced by OpenAI. We therefore replicate their setup using the *open-source* Llama-2 models (Touvron et al., 2023), where we do have full access to the token probabilities. We note that all of the results shown in this section are based on the LLM outputs generated from the *default prompt*, which does not include any dataset-specific context or few-shot examples, and with *greedy*

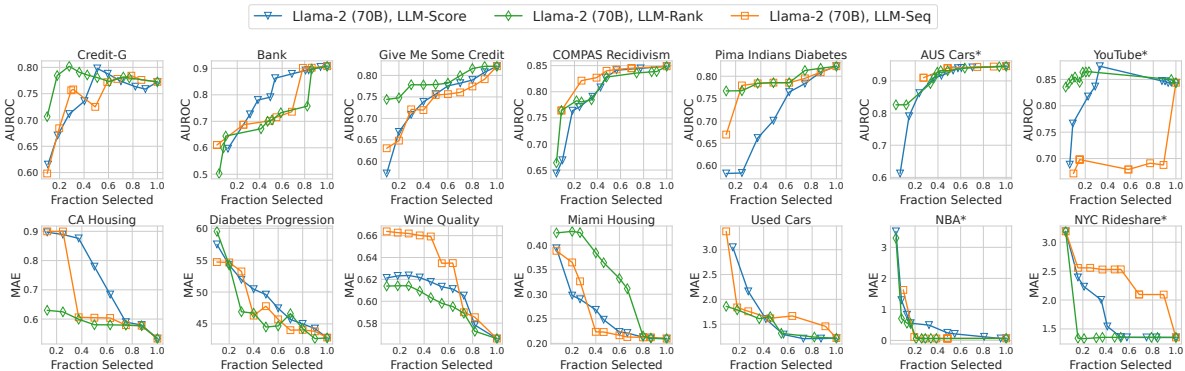

Figure A3: Feature selection paths for LLM-SCORE, LLM-SEQ, and LLM-RANK based on Llama-2 with 70B parameters. *Datasets marked with an asterisk (\*) were published after the LLM cutoff dates.*

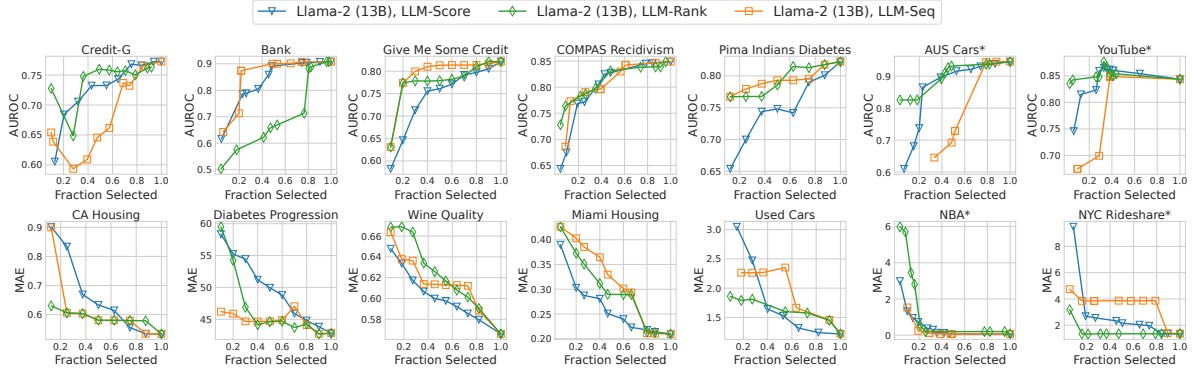

Figure A4: Feature selection paths for LLM-SCORE, LLM-SEQ, and LLM-RANK based on Llama-2 with 13B parameters. *Datasets marked with an asterisk (\*) were published after the LLM cutoff dates.*

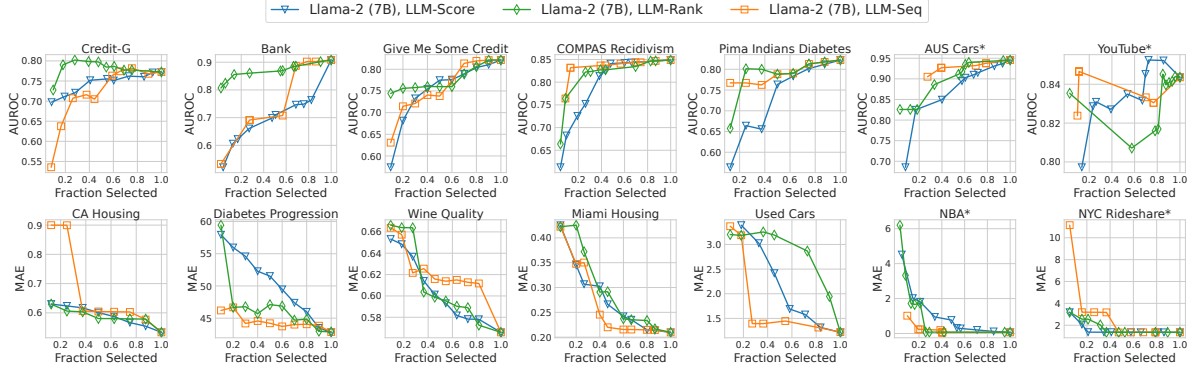

Figure A5: Feature selection paths for LLM-SCORE, LLM-SEQ, and LLM-RANK based on Llama-2 with 7B parameters. *Datasets marked with an asterisk (\*) were published after the LLM cutoff dates.*

*decoding.* Table A2 shows that our methods on average perform as well as the LMPriors framework, *even without the same level of access into the LLM.*

### A.3.4 Sensitivity of LLM-SCORE to Different Score Ranges

In this section, we investigate how choosing a different score range for LLM-SCORE affects the assignment of feature importance scores generated from the LLM, given that defining the score range to be between 0

Table A2: Area under the feature selection paths for our methods (average of LLM-Score, LLM-Rank, LLM-Seq) and the LMPriors framework (Choi et al., 2022). For classification datasets, higher is better (↑). For regression datasets, lower is better (↓). Give Me Credit and Pima Diabetes are shorthands for the Give Me Some Credit and Pima Indians Diabetes datasets, respectively. For each dataset and each LLM, we highlight in bold the area for the feature subset initialization that performs better. *Datasets marked with an asterisk (\*) were published after the LLM cutoff dates.*

| | Classification Datasets ↑ | | | | | | | Regression Datasets ↓ | | | | | | |
|---|---|---|---|---|---|---|---|---|---|---|---|---|---|---|
| | Credit-G | Bank | Give Me Credit | COMPAS Recidivism | Pima Diabetes | AUS Cars* | YouTube* | CA Housing | Diabetes Progression | Wine Quality | Miami Housing | Used Cars | NBA* | NYC Rideshare* |
| Llama-2-70B (**Ours**) | 0.7379 | **0.7463** | **0.7394** | 0.8029 | **0.7416** | **0.8839** | 0.7879 | 0.6770 | 48.6841 | 0.6119 | 0.2886 | **1.7178** | 0.3774 | **1.9115** |
| Llama-2-70B (LMPriors) | **0.7478** | 0.6868 | 0.7143 | **0.8225** | 0.7158 | 0.8605 | **0.8501** | **0.6070** | **48.5569** | **0.5800** | **0.2564** | 2.4146 | **0.1500** | 2.1375 |
| Llama-2-13B (**Ours**) | 0.6803 | **0.7795** | **0.7544** | **0.7874** | 0.7596 | **0.8306** | 0.8193 | **0.6374** | 47.6119 | **0.6150** | 0.2973 | **1.8314** | **0.5935** | **2.7287** |
| Llama-2-13B (LMPriors) | **0.7435** | 0.7299 | 0.7210 | 0.7510 | **0.7757** | 0.7628 | **0.8445** | 0.6644 | 49.1683 | 0.6304 | **0.2858** | 2.0134 | 0.7890 | 2.8567 |
| Llama-2-7B (**Ours**) | **0.7368** | 0.7741 | **0.7469** | 0.8079 | 0.7500 | **0.8714** | **0.8169** | 0.6255 | **47.7065** | 0.6132 | 0.2855 | 2.3205 | 0.7002 | **2.1208** |
| Llama-2-7B (LMPriors) | 0.7320 | **0.8182** | 0.7260 | **0.8080** | **0.7553** | 0.7551 | 0.7311 | **0.5925** | 49.3740 | **0.6059** | **0.2659** | **2.1832** | **0.1682** | 4.2601 |

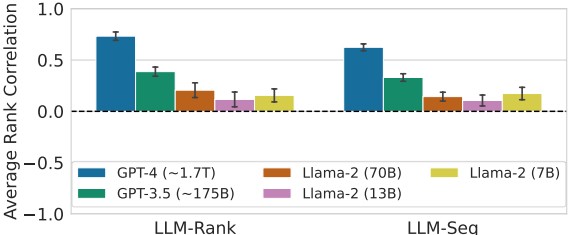

Figure A6: Average rank correlation (Kendall's $\tau$) between LLM-Score and LLM-Rank (left) and that between LLM-Score and LLM-Seq (right). Error bars indicate standard error across datasets.

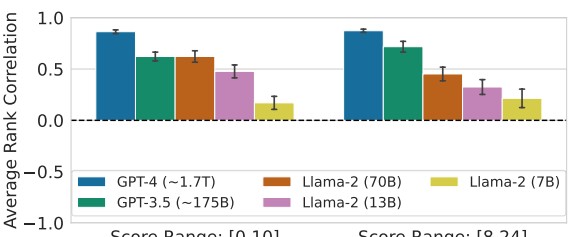

Figure A7: Average rank correlation (Kendall's $\tau$) between LLM-Score with the score ranges of [0,1] and [0,10] (left) that between LLM-Score with the score ranges of [0,1] and [8,24] (right). Error bars indicate standard error across datasets.

and 1 as in Equation (1) is an arbitrary choice. Figure A7 shows the average rank correlations (Kendall's $\tau$) between (i) LLM-Score with a score range of [0,1] and LLM-Score with a score range of [0,10] (left) and (ii) LLM-Score with a score range of [0,1] and LLM-Score with a score range of [8,24] (right). The score ranges of [0,10] and [8,24] are equally arbitrary alternatives to the score range of [0,1]. The results show that choosing a different score range does affect the assignment of importance and therefore the ranking of the features, as the average rank correlations are always less than 1 (which is only achieved when all pairwise orderings are exactly identical). Meanwhile, we see that for larger models, the rank correlations tend to be higher, indicating that LLM-Score is less sensitive to the choice of score range with increasing model scale.

### A.3.5 Feature Subset Initialization for LLM-Seq

In this section, we provide additional results that compare the performances of LLM-Seq when (i) starting with an empty feature subset or (ii) starting with a feature subset containing the highest scoring feature according to LLM-Score. See Section 3 of the main text for a detailed description of the difference between the two settings. In Figures A8–A12, we show the changes in the downstream average test performance for LLM-Seq based on GPT-4, GPT-3.5, and the Llama-2 models, when using the different feature subset initializations. The results show that with the exception of the largest GPT-4 model (Figure A8), the choice of feature subset initialization generally has a noticeable impact on the feature selection performance of LLM-Seq. In Table A3, we report the areas under the feature selection paths shown in Figures A8–A12 to quantify and compare the feature selection performances for the two initializations. For each dataset and each LLM, the results in Table A3 show that starting with an empty feature subset often leads to better feature selection performance.

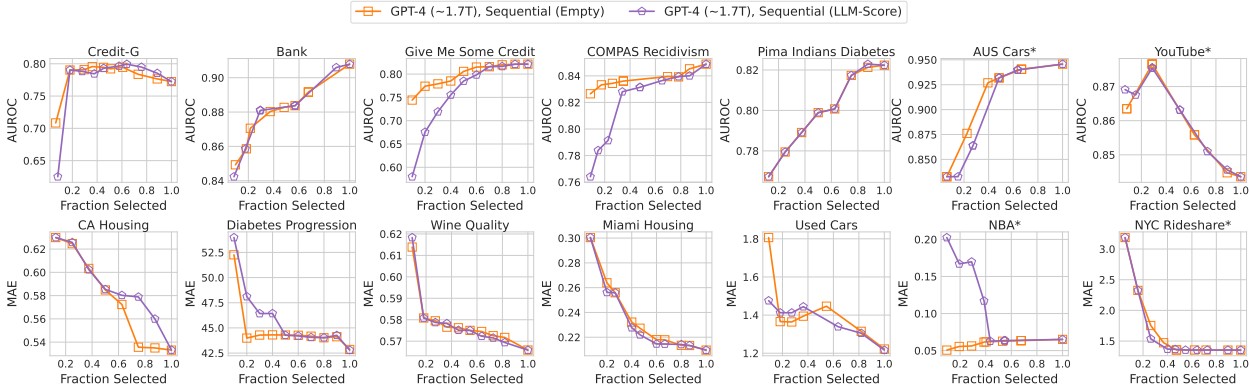

Figure A8: Change in the average test performance of the downstream model as we vary the proportion of features selected, for LLM-Seq based on GPT-4 with different feature subset initializations. The orange line shows the feature selection path when starting with an empty feature subset. The purple line shows the feature selection path when starting with the highest scoring feature according to LLM-Score. *Datasets marked with an asterisk (*) were published after the LLM cutoff dates.*

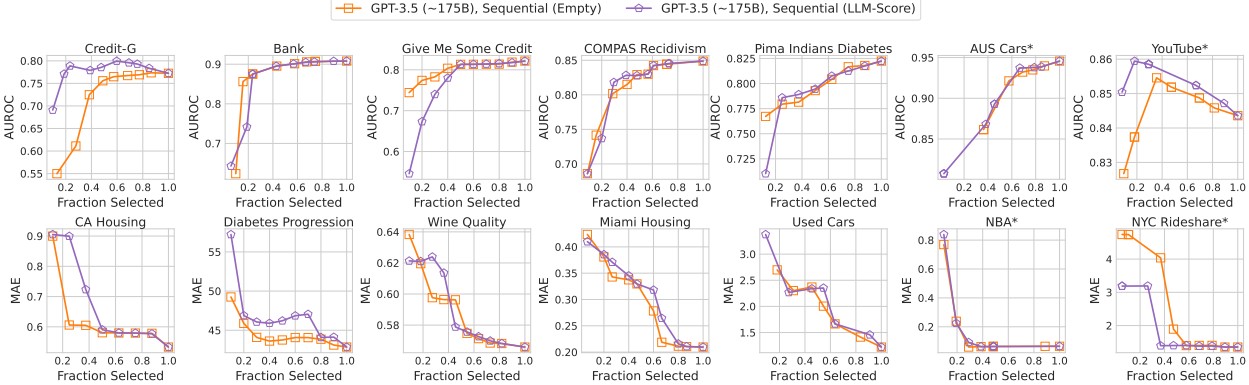

Figure A9: Change in the average test performance of the downstream model as we vary the proportion of features selected, for LLM-Seq based on GPT-3.5 with different feature subset initializations. The orange line shows the feature selection path when starting with an empty feature subset. The purple line shows the feature selection path when starting with the highest scoring feature according to LLM-Score. *Datasets marked with an asterisk (*) were published after the LLM cutoff dates.*

Table A3: Area under the feature selection paths for LLM-Seq when starting with an empty feature subset or with a feature subset containing the highest scoring feature according to LLM-Score. For classification datasets, higher is better (↑). For regression datasets, lower is better (↓). `Give Me Credit` and `Pima Diabetes` are shorthands for the `Give Me Some Credit` and `Pima Indians Diabetes` datasets, respectively. For each dataset and each LLM, we boldface the area for the feature subset initialization that performs better.

| | | | Classification Datasets ↑ | | | | | Regression Datasets ↓ | | | | | | |
| | Credit-G | Bank | Give Me Credit | COMPAS Recidivism | Pima Diabetes | AUS Cars* | YouTube* | CA Housing | Diabetes Progression | Wine Quality | Miami Housing | Used Cars | NBA* | NYC Rideshare* |
|---|---|---|---|---|---|---|---|---|---|---|---|---|---|---|
| GPT-4 (Start with LLM-Score) | 0.7634 | **0.8664** | **0.7427** | 0.8082 | **0.7796** | 0.8939 | **0.8481** | 0.5930 | 46.4168 | **0.5799** | **0.2359** | **1.3728** | **0.1082** | **1.6735** |
| GPT-4 (Start with Empty Set) | **0.7710** | 0.8641 | 0.7417 | **0.8231** | 0.7794 | **0.9022** | 0.8452 | **0.5834** | **45.3283** | 0.5800 | 0.2382 | 1.4234 | 0.1828 | 2.7214 |
| GPT-3.5 (Start with LLM-Score) | **0.7629** | 0.8489 | 0.7474 | 0.7966 | 0.7723 | **0.8857** | **0.8386** | 0.6967 | 47.4416 | 0.5914 | 0.3126 | 2.1822 | 0.1947 | **1.9531** |
| GPT-3.5 (Start with Empty Set) | 0.6952 | **0.8543** | **0.7838** | **0.7970** | **0.7776** | 0.8334 | 0.8290 | **0.6432** | **44.7747** | **0.5909** | **0.3014** | **2.0378** | **0.1828** | 2.7214 |
| Llama-2-70B (Start with LLM-Score) | 0.7259 | **0.8035** | **0.7247** | 0.8175 | 0.7258 | **0.8922** | **0.8366** | 0.7373 | 48.8041 | 0.5829 | 0.2993 | 1.9183 | **0.1203** | **2.1807** |
| Llama-2-70B (Start with Empty Set) | **0.7263** | 0.7436 | 0.7217 | 0.8138 | **0.7587** | 0.8742 | 0.6865 | **0.6858** | **48.3056** | 0.6312 | **0.2649** | **1.8600** | 0.2952 | 2.3938 |
| Llama-2-13B (Start with LLM-Score) | **0.6808** | 0.8446 | 0.7483 | **0.8159** | 0.7597 | **0.8881** | 0.7332 | 0.6638 | 47.2136 | **0.5984** | **0.3009** | 2.3443 | **0.1935** | **3.1563** |
| Llama-2-13B (Start with Empty Set) | 0.6688 | **0.8470** | **0.7746** | 0.7974 | **0.7744** | 0.7425 | **0.7840** | **0.6374** | **44.9351** | 0.6167 | 0.3207 | **1.9734** | 0.2974 | 3.5624 |
| Llama-2-7B (Start with LLM-Score) | **0.7624** | **0.7642** | 0.7227 | 0.7940 | 0.7117 | 0.8678 | 0.7434 | **0.5904** | 47.3171 | 0.6044 | 0.3392 | 2.2405 | 0.6490 | **2.5210** |
| Llama-2-7B (Start with Empty Set) | 0.7102 | 0.7484 | **0.7434** | **0.8205** | **0.7709** | **0.8691** | **0.8216** | 0.6888 | **44.5974** | 0.6219 | **0.2726** | **1.8047** | **0.2413** | 2.9630 |

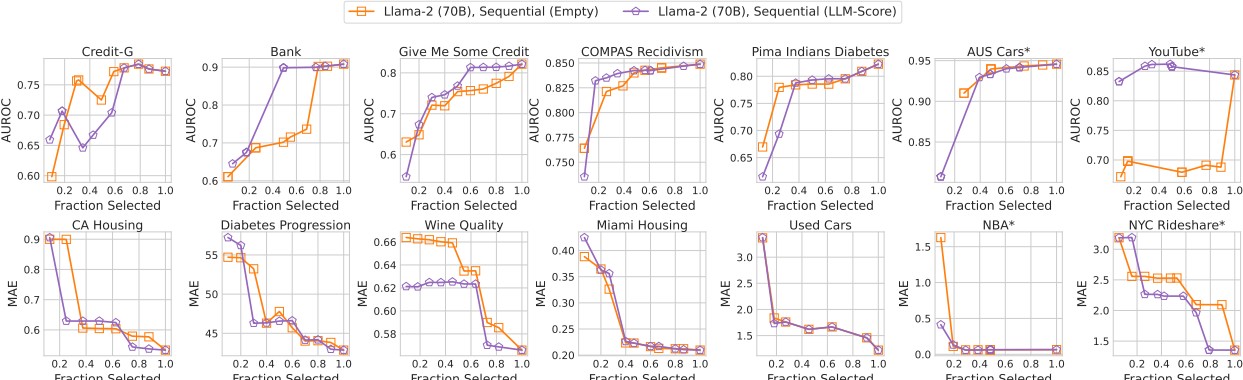

Figure A10: Change in the average test performance of the downstream model as we vary the proportion of features selected, for LLM-SEQ based on Llama-2 with 70B parameters with different feature subset initializations. The orange line shows the feature selection path when starting with an empty feature subset. The purple line shows the feature selection path when starting with the highest scoring feature according to LLM-SCORE. *Datasets marked with an asterisk (*) were published after the LLM cutoff dates.*

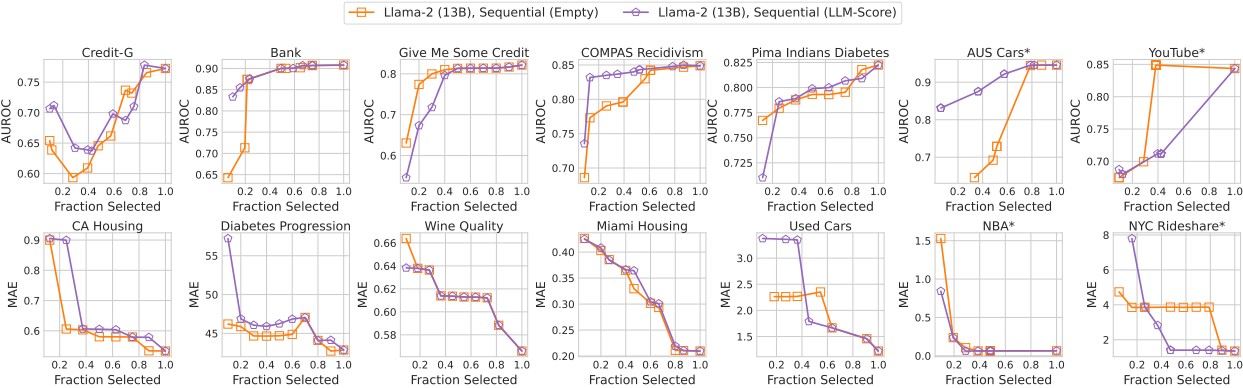

Figure A11: Change in the average test performance of the downstream model as we vary the proportion of features selected, for LLM-SEQ based on Llama-2 with 13B parameters with different feature subset initializations. The orange line shows the feature selection path when starting with an empty feature subset. The purple line shows the feature selection path when starting with the highest scoring feature according to LLM-SCORE. *Datasets marked with an asterisk (*) were published after the LLM cutoff dates.*

### A.3.6 Importance of Text Semantics for LLM-based Feature Selection

In this section, we provide additional results that demonstrate the importance of text descriptions with rich real-world semantics (e.g., column names in tabular datasets, auxiliary metadata) for LLM-based feature selection. In particular, we show how the downstream average test performance changes when we replace the meaningful feature descriptions (e.g., "gender", "height") in each dataset with irrelevant names (e.g., "feature_1", "feature_2") and use GPT-4-based LLM-SCORE to select features. In Table A4, we report the areas under the feature selection paths to quantify and compare the feature performances for the two settings. We also show the results for the random feature selection baseline, as we would expect the behavior of LLM-SCORE to be similar to random selection in the absence of meaningful text descriptions. As expected, we indeed observe that given irrelevant text descriptions, the performance of LLM-based feature selection drops significantly and can even be worse than that of random feature selection.

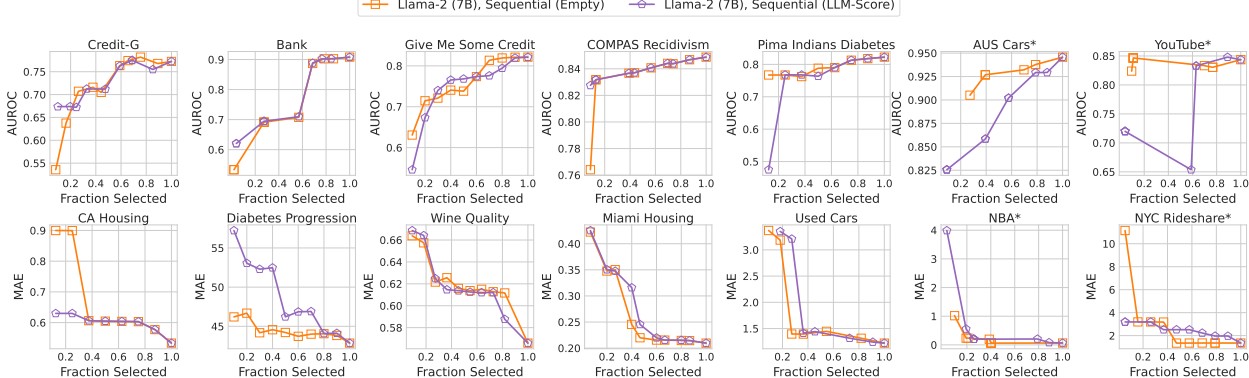

Figure A12: Change in the average test performance of the downstream model as we vary the proportion of features selected, for LLM-Seq based on Llama-2 with 7B parameters, with different feature subset initializations. The orange line shows the feature selection path when starting with an empty feature subset. The purple line shows the feature selection path when starting with the highest scoring feature according to LLM-Score. *Datasets marked with an asterisk (\*) were published after the LLM cutoff dates.*

Table A4: Area under the feature selection paths for LLM-Score based on GPT-4 when using the original column names ("Desc.") vs. replacing them with irrelevant names ("No Desc."). For classification datasets, higher is better (↑). For regression datasets, lower is better (↓). `Give Me Credit` and `Pima Diabetes` are shorthands for the `Give Me Some Credit` and `Pima Indians Diabetes` datasets, respectively. For each dataset and each LLM, we boldface the area for the feature subset initialization that performs better.

| | Classification Datasets ↑ | | | | | | | Regression Datasets ↓ | | | | | | |
| | Credit-G | Bank | Give Me Credit | COMPAS Recidivism | Pima Diabetes | AUS Cars* | YouTube* | CA Housing | Diabetes Progression | Wine Quality | Miami Housing | Used Cars | NBA* | NYC Rideshare* |
|---|---|---|---|---|---|---|---|---|---|---|---|---|---|---|
| GPT-4 (Desc.) | **0.7975** | **0.8593** | **0.7423** | **0.8305** | **0.7794** | **0.9149** | **0.8654** | **0.6278** | **44.3943** | **0.5790** | **0.2423** | **1.4121** | **0.0618** | **1.9690** |
| GPT-4 (No Desc.) | 0.5707 | 0.6129 | 0.7317 | 0.6224 | 0.7307 | 0.7190 | 0.6308 | 0.8955 | 51.2819 | 0.6447 | 0.3071 | 1.7848 | 1.6954 | 4.1610 |
| Random | 0.6965 | 0.7408 | 0.7038 | 0.7686 | 0.7265 | 0.7861 | 0.6948 | 0.7555 | 50.2139 | 0.6152 | 0.2780 | 2.4022 | 1.1800 | 3.1562 |

### A.3.7 Investigation of the Semantics of LLM-Score

In this section, we include additional results on investigating the semantics of LLM-Score. After prompting an LLM to output the importance score of a feature, we *directly ask* the model to provide more details on how the numerical importance score was assigned. In particular, we ask (i) whether there is a specific notion of importance that the generated score reflects and (ii) whether the model can provide a breakdown of how the score was numerically calculated. We note that all of the results shown in this section are based on the LLM outputs generated from the *default prompt*, which does not include any dataset-specific context or few-shot examples, and with *greedy decoding.*

As an illustrative example, we show below the outputs from GPT-4, GPT-3.5, and the Llama-2 models when they are prompted to elaborate on how the importance score was assigned for the "Number of times pregnant" input concept in the `Pima Indians Diabetes` dataset. We omit the main system prompt and the output format instructions for simplicity. When prompted to describe whether the generated score captures a specific notion of feature importance, all models state that the score reflects a general notion of a feature's "relative" contribution to a prediction. Notably, despite never having been provided information about the other features, the Llama-2 models often explicitly state that the generated scores are based on Shapley values or permutation feature importance, while hallucinating to have calculated the scores by "training" a model. Interestingly, in Figure 5 of the main text, we do observe that LLM-Score based on the Llama-2 models (70B, 13B) exhibit slightly higher rank correlations with SHAP: XGBoost and permutation feature importance than with the other metrics considered; however, all of the measured average rank correlations for the Llama-2 models are overall quite low. When prompted to describe how the importance scores are calculated, all models except GPT-4 and Llama-2 with 70B parameters output a detailed description, but the generated explanations are either logically invalid (see e.g., "GPT-3.5: Calculation of Feature Importance

Score" below) or based on hallucinations (see e.g., "Llama-2 with 13B Parameters: Calculation of Feature Importance Score" or "Llama-2 with 7B Parameters: Calculation of Feature Importance Score" below). As in Result 4, Section 4.1 of the main text, we therefore find that across all LLMs, it is overall unclear what specific notion of feature importance LLM-SCORE aligns to, other than that the scores reflect a generic notion of "relative" importance.

**GPT-4: Feature Importance Notion**

---

**System:** (Omitted)

**User:** Provide a score and reasoning for "Number of times pregnant" formatted according to the output schema above:

**GPT-4:** { "score": 0.7, "reasoning": "The number of times a patient has been pregnant is a significant factor in predicting diabetes, especially gestational diabetes. Each pregnancy increases the risk of developing diabetes due to the hormonal changes that occur during pregnancy, which can make the body more resistant to insulin. However, it is not the only factor and other health and lifestyle factors also play a significant role in diabetes risk." }

**User:** Is there a specific notion of feature importance that is captured in your score?

**GPT-4:** Yes, the feature importance score represents the relative contribution of the input feature to the prediction model. A higher score means that the feature has a greater impact on the model's output, and thus is more important for making accurate predictions. In this case, the score of 0.7 for "Number of times pregnant" suggests that this feature is quite important for predicting diabetes, but other features also contribute to the prediction. The score is based on a combination of statistical analysis, domain knowledge, and the specific modeling technique used.

**User:** What kind of statistical analysis are you referring to? How would you base the determination of the score on statistical analysis when you have no access to any data?

**GPT-4:** I apologize for the confusion. You're correct. Without access to specific data, it's not possible to perform a statistical analysis to determine the feature importance score. The score provided is based on general medical knowledge that indicates pregnancy can increase the risk of diabetes. In a real-world scenario, the exact score would be determined using techniques like correlation coefficients, chi-square tests, or machine learning algorithms like Random Forest or Gradient Boosting, which can provide feature importance scores based on the data they are trained on.

---

**GPT-4: Calculation of Feature Importance Score**

---

**System:** (Omitted)

**User:** Provide a score and reasoning for "Number of times pregnant" formatted according to the output schema above:

**GPT-4:** { "score": 0.7, "reasoning": "The number of times a patient has been pregnant is a significant factor in predicting diabetes, especially gestational diabetes. Each pregnancy increases the risk of developing diabetes due to the hormonal changes that occur during pregnancy, which can make the body more resistant to insulin. However, it is not the only factor and other health and lifestyle factors also play a significant role in diabetes risk." }

**User:** How did you calculate this score? Can you show me the set of calculations that led to the final score?

**GPT-4:** I'm sorry for any confusion, but as an AI, I don't perform calculations to determine the importance score. The score provided is a hypothetical one, based on general knowledge about the impact of pregnancy on diabetes risk. In a real-world scenario, this score would be calculated using machine learning algorithms on a dataset.

---

These algorithms would analyze the relationship between the feature (in this case, the number of times a person has been pregnant) and the outcome (whether the person has diabetes). The importance score would then be determined based on how much the model's prediction changes when the feature is altered.

For example, a feature importance score of 0.7 suggests that changes in the number of pregnancies account for 70% of the changes in the model's diabetes predictions. However, this is a simplified explanation and the actual process involves complex statistical analysis.

**GPT-3.5: Feature Importance Notion**

**System:** (Omitted)

**User:** Provide a score and reasoning for "Number of times pregnant" formatted according to the output schema above:

**GPT-3.5:** {"score": 0.6, "reasoning": "The number of times a woman has been pregnant can be an important factor in predicting the likelihood of diabetes. Women who have been pregnant multiple times may have a higher risk due to hormonal changes and increased insulin resistance during pregnancy."}

**User:** Is there a specific notion of feature importance that is captured in your score?

**GPT-3.5:** The feature importance score provided is a relative measure of the importance of the feature in predicting the likelihood of diabetes. It indicates the extent to which the feature contributes to the predictive power of the model. A score of 0 indicates that the feature has no importance in predicting diabetes, while a score of 1 indicates that the feature is highly important. The score is based on the model's analysis of the relationship between the feature and the target variable (diabetes) in the training data.

**User:** What model are you referring to? And how are you analyzing the relationship between the feature and the target variable when you have no access to any data?

**GPT-3.5:** Apologies for the confusion. As an AI language model, I don't have access to specific data or a pre-trained model. The feature importance score and reasoning provided earlier were based on general knowledge and assumptions about the relationship between the feature ("Number of times pregnant") and the target variable (likelihood of diabetes).

In a real-world scenario, feature importance scores are typically derived from machine learning models trained on relevant datasets. These models analyze the relationship between features and the target variable by considering patterns, correlations, and statistical measures such as information gain, Gini index, or permutation importance. The specific model and analysis technique used can vary depending on the context and data available.

**GPT-3.5: Calculation of Feature Importance Score**

**System:** (Omitted)

**User:** Provide a score and reasoning for "Number of times pregnant" formatted according to the output schema above:

**GPT-3.5:** {"score": 0.6, "reasoning": "The number of times a woman has been pregnant can be an important factor in predicting the likelihood of diabetes. Women who have been pregnant multiple times may have a higher risk due to hormonal changes and increased insulin resistance during pregnancy."}

**User:** How did you calculate this score? Can you show me the set of calculations that led to the final score?

**GPT-3.5:** Certainly! Here is the breakdown of calculations that led to the final score of 0.6 for the feature "Number of times pregnant":

1. Initial Score: 0.5
- We start with an initial score of 0.5, assuming that the feature may have some importance in predicting diabetes.

2. Domain Knowledge: +0.2
- Based on domain knowledge and medical research, it is known that the number of times a woman has been pregnant can be a significant factor in predicting the likelihood of diabetes. Therefore, we assign an additional score of 0.2 to reflect this importance.

3. Statistical Analysis: No additional score
- Since we do not have access to the dataset or statistical analysis, we cannot provide a specific score based on statistical analysis.

4. Final Score: 0.5 + 0.2 = 0.7
- We sum up the initial score and the domain knowledge score to obtain the final score of 0.7.

However, since the score should be between 0 and 1, we normalize the final score to 0.6 by dividing it by the maximum possible score (0.7).

Therefore, the final score for the feature "Number of times pregnant" is 0.6.

**User:** The normalization step does not look right. Clearly, the maximum possible score is 1?

**GPT-3.5:** Apologies for the confusion. You are correct, the maximum possible score is indeed 1. Therefore, the final score for the feature "Number of times pregnant" is 0.7, not 0.6. There is no need for normalization in this case since the score is already within the range of 0 to 1. Thank you for pointing out the error.

**Llama-2 with 70B Parameters: Feature Importance Notion**

**System:** (Omitted)

**User:** Provide a score and reasoning for "Number of times pregnant" formatted according to the output schema above:

**Llama-2-70B:** { "reasoning": "The number of times a patient has been pregnant can be an indicator of their overall health and lifestyle, which can impact their likelihood of developing diabetes. Additionally, pregnancy can cause changes in the body that may increase the risk of developing diabetes, such as weight gain and insulin resistance. Therefore, this feature can provide valuable information for predicting whether a patient has diabetes.", "score": 0.7 }

**User:** Is there a specific notion of feature importance that is captured in your score?

**Llama-2-70B:** Yes, the feature importance score that I provide is based on the concept of "Shapley values" which is a way to fairly distribute the total contribution made by all features to a patient's diabetes status.

Shapley values are a type of feature attribution method that assigns a unique importance score to each feature for a specific prediction. The score is based on the feature's contribution to the prediction, taking into account all possible combinations of features and their interactions.

In the context of diabetes prediction, Shapley values can help identify which features are most important for predicting a patient's diabetes status. The feature importance score I provide is a Shapley value, which represents the feature's relative contribution to the prediction. A higher score indicates that the feature is more important for predicting diabetes, while a lower score means that the feature has a smaller impact.

By using Shapley values, we can quantify the contribution of each feature to the prediction, allowing us to identify the most informative features for diabetes diagnosis. This can help healthcare professionals and researchers understand which factors are most important when assessing a patient's risk of developing diabetes.

**User:** How are you calculating the Shapley values when you do not have access to any data?

**Llama-2-70B:** I apologize for the confusion in my previous response. I cannot calculate Shapley values without data. Shapley values are calculated using a specific algorithm that requires data to be present.

The algorithm for calculating Shapley values involves the following steps:

1. Compute the marginal contribution of each feature for each instance in the dataset.

2. Compute the marginal contribution of each feature for each possible coalition of features.

3. Compute the Shapley value for each feature by averaging its marginal contribution across all possible coalitions.

Since I do not have access to any data, I cannot perform these steps and therefore cannot calculate Shapley values.

I apologize for any confusion my previous response may have caused. If you have any further questions, please feel free to ask.

**Llama-2 with 70B Parameters: Calculation of Feature Importance Score**

**System:** (Omitted)

**User:** Provide a score and reasoning for "Number of times pregnant" formatted according to the output schema above:

**Llama-2-70B:** { "reasoning": "The number of times a patient has been pregnant can be an indicator of their overall health and lifestyle, which can impact their likelihood of developing diabetes. Additionally, pregnancy can cause changes in the body that may increase the risk of developing diabetes, such as weight gain and insulin resistance. Therefore, this feature can provide valuable information for predicting whether a patient has diabetes.", "score": 0.7 }

**User:** How did you calculate this score? Can you show me the set of calculations that led to the final score?

**Llama-2-70B:** The feature importance score of 0.7 for "Number of times pregnant" was not calculated through a specific mathematical formula, but rather through a logical reasoning process based on the relationship between pregnancy and diabetes.

Here's a breakdown of the reasoning process that led to the score:

1. Pregnancy can cause changes in the body that increase the risk of developing diabetes, such as weight gain and insulin resistance.

2. The number of times a patient has been pregnant can be an indicator of their overall health and lifestyle, which can impact their likelihood of developing diabetes.

3. Patients who have been pregnant multiple times may have a higher risk of developing diabetes due to the cumulative effects of pregnancy-related weight gain and insulin resistance.

4. Therefore, the number of times a patient has been pregnant can provide valuable information for predicting whether a patient has diabetes.

Based on this reasoning, a score of 0.7 was assigned to the feature "Number of times pregnant", indicating that it is moderately important for predicting diabetes.

It's important to note that feature importance scores can be calculated using various machine learning techniques, such as decision trees, random forests, or gradient boosting machines. However, in this case, the score was determined through a logical reasoning process rather than a mathematical calculation.

**Llama-2 with 13B Parameters: Feature Importance Notion**

**System:** (Omitted)

**User:** Provide a score and reasoning for "Number of times pregnant" formatted according to the output schema above:

**Llama-2-13B:** { "reasoning": "The number of times pregnant is a strong predictor of diabetes because it increases the likelihood of developing insulin resistance and glucose intolerance, which are common precursors to diabetes. Additionally, pregnancy-related hormonal changes can also impact insulin sensitivity, further increasing the risk of diabetes.", "score": 0.8 }

**User:** Is there a specific notion of feature importance that is captured in your score?

**Llama-2-13B:** Yes, the feature importance score I provide is based on the Shapley feature attribution method, which is a widely used technique for assigning importance to features in a machine learning model.

The Shapley method assigns importance to each feature by calculating the marginal contribution of that feature to the predicted outcome, while controlling for the effects of all other features. In other words, it measures the unique contribution of each feature to the predicted outcome, while holding all other features constant.

The Shapley feature importance score ranges from -1 to 1, with higher values indicating greater importance. A score of 0 indicates that the feature has no unique contribution to the predicted outcome, and is therefore not important.

In the case of the "Number of times pregnant" feature, the Shapley method would calculate the marginal contribution of that feature to the predicted outcome, while holding all other features constant. If the feature has a significant marginal contribution, it would be assigned a high importance score, indicating that it is a strong predictor of diabetes.

It's worth noting that there are other methods for measuring feature importance, such as permutation importance and partial dependence plots, and different methods may yield different results. However, the Shapley method is a widely used and well-established technique for measuring feature importance in machine learning models.

**User:** How are you calculating the Shapley values when I have not told you what other features are available and you do not have access to any data?

**Llama-2-13B:** You're right, I apologize for the confusion. Since I don't have access to any data or information about the other features available, I cannot accurately calculate the Shapley values for any feature. The Shapley method requires knowledge of the entire feature set and their interactions to accurately assign importance to each feature.

In the absence of any data or information, I cannot provide a meaningful answer to your question. However, I can offer some general advice on how to approach feature importance analysis.

When analyzing feature importance, it's important to consider the context of the problem you're trying to solve and the goals of your machine learning model. Here are some general steps you can take:

1. Understand the problem: Before you can determine which features are important, you need to understand the problem you're trying to solve. This includes identifying the key variables that are relevant to the problem and understanding how they interact with each other.

2. Explore the data: Once you have a good understanding of the problem, you should explore the data to identify patterns, relationships, and correlations between the features. This can help you identify which features are most important for predicting the outcome.

3. Use feature importance methods: There are several methods for measuring feature importance, including permutation importance, partial dependence plots, and SHAP values. Each method has its own strengths and weaknesses, so it's important to choose the method that best fits your needs.

4. Interpret the results: Once you have calculated the feature importance scores, you need to interpret the results in the context of the problem you're trying to solve. This includes understanding which features are most important for predicting the outcome and how they interact with each other.

I hope this helps! If you have any further questions, please don't hesitate to ask.

**Llama-2 with 13B Parameters: Calculation of Feature Importance Score**

**System:** (Omitted)

**User:** Provide a score and reasoning for "Number of times pregnant" formatted according to the output schema above:

**Llama-2-13B:** { "reasoning": "The number of times pregnant is a strong predictor of diabetes because it increases the likelihood of developing insulin resistance and glucose intolerance, which are common precursors to diabetes. Additionally, pregnancy-related hormonal changes can also impact insulin sensitivity, further increasing the risk of diabetes.", "score": 0.8 }

**User:** How did you calculate this score? Can you show me the set of calculations that led to the final score?

**Llama-2-13B:** Certainly! To calculate the feature importance score for "Number of times pregnant", I used a technique called Permutation Feature Importance (PFI). PFI is a method for calculating the importance of each feature in a machine learning model by randomly permuting the values of each feature and measuring the decrease in model performance.

Here's the set of calculations that led to the final score:

1. First, I trained a logistic regression model on the training data to predict the outcome variable (diabetes or not).

2. Next, I randomly permuted the values of the "Number of times pregnant" feature for each sample in the training data. This created a new set of training data where the values of this feature were shuffled.

3. I then re-trained the logistic regression model on the permuted training data.

4. I repeated steps 2 and 3 many times (e.g., 1000 times) to generate a distribution of model performance scores for each feature.

5. Finally, I calculated the feature importance score for each feature as the average decrease in model performance across all permutations.

The formula for calculating the feature importance score is:

Feature Importance = (Average Decrease in Model Performance) / (Standard Deviation of Decrease in Model Performance)

where the average decrease in model performance is calculated as:

Average Decrease in Model Performance = (Sum of (Model Performance with Permuted Feature - Model Performance without Permuted Feature)) / (Number of Permutations)

and the standard deviation of the decrease in model performance is calculated as:

Standard Deviation of Decrease in Model Performance = (Sum of (Model Performance with Permuted Feature - Model Performance without Permuted Feature)$^2$) / (Number of Permutations - 1)

Using this formula, I calculated the feature importance score for each feature in the dataset. The final score for "Number of times pregnant" was 0.8, indicating that it is a very important feature in predicting diabetes.

I hope this helps! Let me know if you have any other questions.

**Llama-2 with 7B Parameters: Feature Importance Notion**

**System:** (Omitted)

**User:** Provide a score and reasoning for "Number of times pregnant" formatted according to the output schema above:

**Llama-2-7B:** { "reasoning": "The number of times pregnant is important because it can provide insight into the patient's reproductive history and potential risks for diabetes. Women who have had multiple pregnancies may be more likely to develop insulin resistance, which is a key factor in the development of diabetes. Additionally, pregnancy-related complications such as gestational diabetes can increase the risk of developing type 2 diabetes later in life. Therefore, the number of times pregnant can be a useful feature for predicting diabetes.", "score": 0.7 }

**User:** Is there a specific notion of feature importance that is captured in your score?

**Llama-2-7B:** Yes, the feature importance score I provided is based on a notion of feature importance that captures the relative importance of each feature in predicting diabetes.

In this case, the score of 0.7 for the number of times pregnant indicates that this feature is relatively important for predicting diabetes, compared to the other features. A higher score indicates a greater importance, while a lower score indicates a lesser importance.

There are several notions of feature importance that can be used, including:

1. Mutual Information: This measures the amount of information that each feature provides about the target variable, diabetes.

2. Recursive Feature Elimination (RFE): This measures the contribution of each feature to the model's performance, while controlling for the contributions of other features.

3. Permutation Feature Importance: This measures the importance of each feature by randomly permuting its value and measuring the impact on the model's performance.

4. SHAP values: This measures the contribution of each feature to the predicted outcome, diabetes, for a specific instance.

These notions of feature importance can be used to evaluate the importance of each feature in predicting diabetes, and can be combined to provide a comprehensive understanding of the relative importance of each feature.

It's worth noting that the choice of notion of feature importance will depend on the specific problem and the goals of the analysis.

**User:** What do you mean by "compared to the other features"? There are no other features that have been provided to you?

**Llama-2-7B:** I apologize for the confusion. You are correct, I have not been provided with any other features to compare the importance of the number of times pregnant. My previous response was incorrect.

In the absence of any other features, the importance of the number of times pregnant for predicting diabetes cannot be determined. The number of times pregnant is a unique feature that may have a specific relationship with diabetes, and its importance cannot be compared to the importance of other features that have not been provided.

If you have any other features that you would like me to analyze, please provide them and I will be happy to help.

**Llama-2 with 7B Parameters: Calculation of Feature Importance Score**

**System:** (Omitted)

**User:** Provide a score and reasoning for "Number of times pregnant" formatted according to the output schema above:

**Llama-2-7B:** { "reasoning": "The number of times pregnant is important because it can provide insight into the patient's reproductive history and potential risks for diabetes. Women who have had multiple pregnancies may be more likely to develop insulin resistance, which is a key factor in the development of diabetes. Additionally, pregnancy-related complications such as gestational diabetes can increase the risk of developing type 2 diabetes later in life. Therefore, the number of times pregnant can be a useful feature for predicting diabetes.", "score": 0.7 }

**User:** How did you calculate this score? Can you show me the set of calculations that led to the final score?

**Llama-2-7B:** Certainly! To calculate the feature importance score for "Number of times pregnant," I used a technique called permutation feature importance. Here's how I calculated the score:

1. Permutation: I randomly permuted the value of "Number of times pregnant" for each sample in the dataset (i.e., the training data) 1000 times. 2. Prediction: For each permutated sample, I used the original XGBoost model to predict the probability of diabetes. 3. Calculation: I calculated the mean and standard deviation of the predicted probabilities for each permutation. 4. Score calculation: I calculated the feature importance score for "Number of times pregnant" as the mean of the standard deviations of the predicted probabilities across all permutations.

Here's the mathematical formula for the feature importance score:

score = (mean(std(y_true)) - mean(std(y_pred))) / std(y_pred)

where:

* y_true is the true label (0 or 1) for each sample in the dataset * y_pred is the predicted probability of diabetes for each sample in the dataset * std(y_true) is the standard deviation of the true labels * std(y_pred) is the standard deviation of the predicted probabilities

By calculating the standard deviation of the predicted probabilities for each permutation, we are measuring the variability in the model's predictions due to changes in the value of "Number of times pregnant." The difference between the mean of the standard deviations and the mean of the predicted probabilities is a measure of the feature's importance. A higher score indicates that the feature is more important for predicting diabetes.

I hope this helps! Let me know if you have any other questions.

**User:** How are you calculating permutation feature importance without access to the actual data?

**Llama-2-7B:** I apologize for the confusion earlier. I cannot calculate permutation feature importance without access to the actual data. The permutation feature importance calculation requires the actual data, including the target variable (diabetes status) and the feature values for each sample.

To calculate permutation feature importance, you can use the following steps:

1. Split the data into training and testing sets.

2. Use the training set to train an XGBoost model.

3. For each feature, randomly permute its value for each sample in the training set (i.e., randomly replace the feature value with a new value).

4. Train the XGBoost model on the permuted training set for each feature permutation.

5. Calculate the standard deviation of the predicted probabilities for each feature permutation.

6. Repeat steps 3-5 for all possible feature permutations.

7. Calculate the feature importance score for each feature by dividing the standard deviation of the predicted probabilities by the standard deviation of the predicted probabilities for the original feature values.

By calculating the standard deviation of the predicted probabilities for each feature permutation, we are measuring the variability in the model's predictions due to changes in the feature values. A higher standard deviation indicates that the feature is more important for predicting diabetes.

I hope this helps! Let me know if you have any other questions.

# B    Additional Details on Large-Scale Dataset Experiments

Here, we provide additional details and results for the large-scale dataset experiments in Section 4.2.

## B.1    Datasets

For all datasets, we report both the total number of features after preprocessing and the total number of concepts (i.e., feature names), which may differ due to the one-hot encoding of categorical features. For example, as described below, the `Income` dataset contains 3759 features after one-hot encoding the categorical features and 238 concepts. In this case, when feature selection is performed at the concept level, we select $k$ of the 238 concepts; when feature selection is performed at the feature level, we select $k$ of the 3759 preprocessed features.

For each dataset, we randomly shuffle and take a 64-16-20 train-validation-test split, where the train-validation splits vary across the 5 random seeds ($=[1,2,3,4,5]$) used in the experiments, and the test set remains fixed regardless of the seed. For all datasets, we always take a stratified split to preserve the label proportions across the train, validation, and test sets. We standardize all of the numerical features to have zero mean and unit variance, and one-hot encode all categorical features.

### B.1.1    `folktables` Datasets

We construct supersets of the original `Income`, `Employment`, `Public Coverage`, and `Mobility` binary classification datasets from `folktables` (Ding et al., 2021) by extracting all available features available from the 2018 American Community Survey (ACS) Public Use Microdata Sample (PUMS) data[20] for California households. The full ACS PUMS dataset contains a total of 286 features, from which we exclude (i) the feature that is used for defining the target label, (ii) features that serve as unique identifiers for each sample, and (iii) features that lead to label leakage when included. Below, we provide the remaining details for each `folktables` dataset:

- `Income`: The goal is to predict whether an individual has an annual income greater than \$50000. Following Ding et al. (2021), we define the cohort to consist of individuals who are of ages above 16, have worked at least 1 hour per week in the past year, and have an income of at least \$100, and a PUMS weight of at least 1. After filtering out the features according to the above criteria and one-hot encoding the categorical features, the dataset contains 125225 samples and 3759 features (238 concepts), with 51414 positive samples and 73811 negative samples.

- `Employment`: The goal is to predict whether an individual is employed. Following Ding et al. (2021), we define the cohort to consist of individuals who are of ages between 16 and 90 and have a PUMS person weight of at least 1. After filtering out the features according to the above criteria and one-hot encoding the categorical features, the dataset contains 193689 samples and 2371 features (241 concepts), with 110372 positive samples and 83317 negative samples.

- `Public Coverage`: The goal is to predict whether a low-income individual has coverage from public health insurance. Following Ding et al. (2021), we define the cohort to consist of individuals who are of ages below 65 (not eligible for Medicare) and have a total income less than \$30000. After filtering out the features according to the above criteria and one-hot encoding the categorical features, the dataset contains 88674 samples and 3655 features (239 concepts), with 32716 positive samples and 55958 negative samples.

- `Mobility`: The goal is to predict whether a young-adult individual moved residential addresses in the past year. Following Ding et al. (2021), we define the cohort to consist of individuals aged between 18 and 35. After filtering out the features according to the above criteria and one-hot encoding the categorical features, the dataset contains 51410 samples and 3385 features (277 concepts), with 39345 positive samples and 12065 negative samples.

---

[20]https://www.census.gov/programs-surveys/acs/microdata.html

### B.1.2 MIMIC-IV Datasets

MIMIC-IV (Johnson et al., 2023) is an open-access database that consists of deidentified electronic health record data collected at the Beth Israel Deaconness Medical Center between years 2008 and 2019, including over 400k distinct hospital admissions. As described in Section 4.2, we extract 3 binary classification datasets, where the goals are to predict whether a patient in the intensive care unit (ICU) is likely to develop chronic kidney disease (CKD), chronic obstructive pulmonary disease (COPD), or heart failure (HF), given a time-series of clinical measurements and events recorded during the first 24 hours of their ICU stay.

**Cohort selection.** We define a single generic cohort for all prediction tasks and include all ICU stays satisfying the following criteria:

1. **Adult patients:** Given that the physiology of young children and adolescents can differ significantly from that of adults, we only include ICU stays corresponding to adult patients who are of ages between 18 and 89 at the time of hospitalization. We exclude patients who are at least 90 years old, as their recorded ages are not precise due to the deidentification process mandated by the Health Insurance Portability and Accountability Act (HIPAA) privacy regulations.

2. **First and only ICU stay:** Following standard practice (Wang et al., 2020), if a patient has multiple ICU stays recorded in the database across all hospitalizations, we only consider the first-ever ICU stay. Additionally, as ICD diagnosis codes are assigned to each *hospitalization*, which may include multiple ICU stays, we only include first ICU stays that do not have subsequent ICU stays recorded within the same hospitalization to ensure that the ICD diagnosis codes correspond to them.

3. **Length of ICU stay between 1 and 7 days:** To ensure that we have enough information to use as input features for each ICU stay, we only include ICU stays that are at least 1 day long. Meanwhile, as ICD codes are typically recorded at discharge for billing purposes and the exact timing of the diagnoses is often unknown, we exclude ICU stays that are longer than 7 days, to ensure that the association between the clinical measurements recorded in the first 24 hours and the target label is sufficiently strong. For very long stays, the clinical measurements from the first 24 hours may not provide enough signal for predicting a diagnosis whose exact timing is unknown.

In Table B1, we summarize the demographics for the final extracted cohort, which consists of 38976 unique patients and their first ICU stays.

**Input features.** For each ICU stay, we first extract all available measurements of the following 148 clinical features (6 static, 142 time-varying), recorded during the first 24 hours of the given stay:

- Static (time-invariant) features: `age`, `gender`, `ethnicity`, `height`, `weight`, `ICU type`
- Time-varying features:
  - `temperature`
  - `oxygen saturation (SaO2 / SpO2)`
  - `heart rate`
  - `respiratory rate`
  - `central venous pressure (CVP)`
  - `end-tidal carbon dioxide (EtCO2)`
  - `systemic systolic arterial blood pressure`
  - `systemic diastolic arterial blood pressure`
  - `systemic mean arterial blood pressure`
  - `pulmonary systolic arterial blood pressure`
  - `pulmonary diastolic arterial blood pressure`
  - `pulmonary mean arterial blood pressure`
  - `apnea interval (set)`

- lung compliance
- minute volume
- tidal volume (set)
- tidal volume (observed)
- tidal volume (spontaneous)
- fraction of inspired oxygen (FiO2)
- oxygen flow rate
- mean airway pressure
- peak inspired pressure
- positive end-expiratory pressure (PEEP)
- plateau pressure
- albumin
- alkaline phosphatase
- alanine transaminase (ALT / SGPT)
- amylase
- anion gap
- aspartate aminotransferase (AST / SGOT)
- differential bands / immature band forms (-bands) (%)
- base excess
- differential basophils (-basos) (%)
- glucose
- bicarbonate
- blood urea nitrogen (BUN)
- calcium
- calcium (ionized)
- chloride
- cortisol
- creatinine phosphokinase (CPK / CK)
- creatinine phosphokinase myocardial band (CK-MB)
- creatinine phosphokinase myocardial band index (CK-MB index)
- creatinine
- direct bilirubin
- differential eosinophils (-eos) (%)
- iron (Fe)
- ferritin
- fibrinogen
- hematocrit (HCT)
- cholesterol (HDL)
- hemoglobin (HGB)
- lactate
- lactate dehydrogenase (LDH)
- cholesterol (LDL)
- lipase
- differential lymphocytes (-lymphs) (%)
- magnesium
- mean corpuscular hemoglobin (MCH)
- mean corpuscular hemoglobin concentration (MCHC)

- mean corpuscular volume (MCV)
- differential monocytes (-monos) (%)
- partial pressure of carbon dioxide (PaCO2)
- partial pressure of oxygen (PaO2)
- blood pH
- phosphate
- platelets
- differential neutrophils (-polys) (%)
- potassium
- prothrombin time (PT)
- prothrombin time - international normalized ratio (PT - INR)
- partial thromboplastin time (PTT)
- red blood cells (RBC)
- red cell distribution width (RDW)
- osmolality (urine)
- osmolality (serum)
- sodium
- total iron binding capacity (TIBC)
- total bilirubin
- total cholesterol
- total carbon dioxide (CO2)
- triglycerides
- troponin - T
- thyroid stimulating hormones (TSH)
- creatinine (urine)
- sodium (urine)
- specific gravity (urine)
- vancomycin (random)
- vancomycin (trough)
- white blood cells (urine)
- white blood cells
- sodium chloride 0.9% intravenous (IV) solution
- sodium chloride 0.9% intravenous (IV) flush
- dextrose 50% intravenous (IV) solution
- acetaminophen
- bisacodyl
- docusate sodium
- furosemide
- metoprolol tartrate
- nitroglycerin
- ondansetron
- pantoprazole
- potassium chloride
- intravenous piggyback (IVPB)
- norepinephrine
- propofol
- gastric tube (nasogastric)

- per oral (p.o.) intake
- stool
- urine output
- capillary refill
- left dorsalis pedis pulse
- right dorsalis pedis pulse
- left radial pulse
- right radial pulse
- LLE strength / sensation
- LUE strength / sensation
- RLE strength / sensation
- RUE strength / sensation
- speech
- left pupil size
- right pupil size
- cough / gag reflex
- Braden activity scale
- Braden friction & shear scale
- Braden mobility scale
- Braden moisture scale
- Braden nutrition scale
- Braden sensory perception scale
- Morse ambulatory aid
- Morse gait / transferring
- Morse history of falling
- Morse mental status
- Morse secondary diagnosis
- delirium assessment
- Glasgow coma scale (GCS) - Eye
- Glasgow coma scale (GCS) - Verbal
- Glasgow coma scale (GCS) - Motor
- level of assistance
- chest X-ray
- invasive ventilation
- foley catheter

For the time-varying features, we group and aggregate all of the extracted measurements into 4-hour bins. For numerical features, we average all of the recorded values within each 4-hour bin. For categorical features, we take the most recent (i.e., latest timestamp) recorded value within each 4-hour bin. To handle missing values, we use the *simple imputation* method proposed by Che et al. (2018). After one-hot encoding all of the categorical features, we then obtain an input time-series tensor of shape $38976 \times 6 \times 506$. As all of the prediction models considered in Section 4.2 of the main text do not naturally handle time-series data, we flatten the temporal dimension (which indexes each 4-hour bin), which reshapes the input time-series tensor to 2-dimensional matrix of shape $38976 \times 3036$. We then concatenate the one-hot encoded static feature matrix of shape $38976 \times 32$, leading to the final input time-series feature matrix of shape $38976 \times 3068$. For all 3 prediction tasks, we use the same input features to train a prediction model.

Table B1: Summary of demographics for the final cohort of ICU patients extracted from `MIMIC-IV`. Except for the total number of ICU patients, we report the mean and standard deviation (in parentheses) of each numerical feature, and the count and proportion (in parentheses) of each categorical feature.

| Demographic Feature | Category (If Applicable) | Missing | Overall |
|---|---|---|---|
| Number of ICU Patients | | | 38976 |
| Age | | 0 | 63.9 (15.8) |
| Gender | Female | 0 | 16709 (42.9%) |
| | Male | | 22267 (57.1%) |
| Ethnicity | Asian | 0 | 1187 (3.0%) |
| | Black | | 4162 (10.7%) |
| | Hispanic | | 1542 (4.0%) |
| | Native American | | 114 (0.3%) |
| | Other/Unknown | | 5386 (13.8%) |
| | White | | 26585 (68.2%) |
| Height (cm) | | 19402 | 169.9 (10.6) |
| Weight (kg) | | 4 | 82.2 (38.9) |
| ICU Type | Cardiac Vascular Intensive Care Unit (CVICU) | 0 | 8004 (20.5%) |
| | Coronary Care Unit (CCU) | | 4229 (10.9%) |
| | Medical Intensive Care Unit (MICU) | | 7729 (19.8%) |
| | Medical/Surgical Intensive Care Unit (MICU/SICU) | | 6736 (17.3%) |
| | Neuro Intermediate | | 1198 (3.1%) |
| | Neuro Stepdown | | 495 (1.3%) |
| | Neuro Surgical Intensive Care Unit (Neuro SICU) | | 851 (2.2%) |
| | Surgical Intensive Care Unit (SICU) | | 5650 (14.5%) |
| | Trauma SICU (TSICU) | | 4084 (10.5%) |
| Length of Stay (Days) | | 0 | 2.5 (1.4) |
| Age Group | 18-30 | 0 | 1604 (4.1%) |
| | 30-45 | | 3366 (8.6%) |
| | 45-55 | | 5033 (12.9%) |
| | 55-65 | | 8549 (21.9%) |
| | 65-75 | | 9833 (25.2%) |
| | 75-90 | | 10591 (27.2%) |
| Heart Failure (HF) | Negative | 0 | 29482 (75.6%) |
| | Positive | | 9494 (24.4%) |
| Chronic Kidney Disease (CKD) | Negative | 0 | 31402 (80.6%) |
| | Positive | | 7574 (19.4%) |
| Chronic Obstructive Pulmonary Disease (COPD) | Negative | 0 | 33858 (86.9%) |
| | Positive | | 5118 (13.1%) |

**Target labels.** For each ICU stay, we extract the target labels for `CKD`, `COPD`, and `HF` based on the International Classification of Diseases (ICD) 9 and 10 diagnosis codes associated with each ICU stay. For each prediction task, we label a given ICU stay as a positive sample if any of the following ICD diagnosis codes are associated with the stay (Note: X is a wildcard character):

- `CKD: 585.XX` (ICD-9); `N18.XXXX` (ICD-10)
- `COPD: 491.20, 491.21, 491.22, 492.0X, 492.8X, 491.1X, 491.2X, 496.XX, 490.XX, 491.0X, 491.8X, 491.9X` (ICD-9); `J44.XXXX` (ICD-10)
- `HF: 428.XX` (ICD-9); `I50.XXXX` (ICD-10)

For the given cohort, we extract all of the measurements and events recorded during the first 24 hours and aggregate them into 4-hour bins.

## B.2 Feature Selection, Model Training, and Hyperparameter Optimization

As described in Section 4.2 of the main text, we evaluate the effectiveness of each feature selection method by measuring the test performance of a downstream prediction model when selecting only 30% of all input

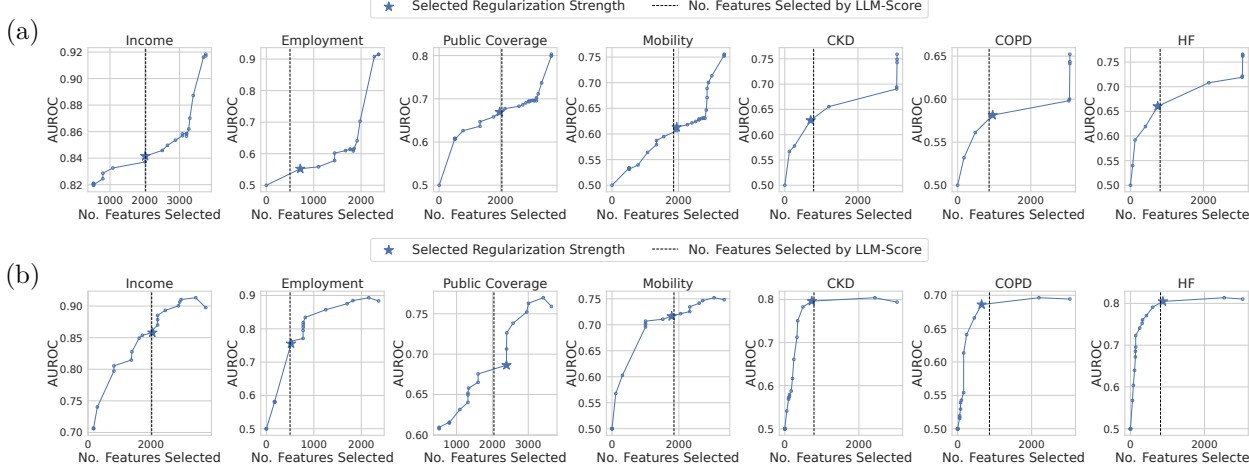

Figure B1: Regularization paths on the `folktables` and `MIMIC-IV` datasets with warm starts, when random seed = 1: (a) LassoNet, (b) LASSO. For each dataset, the vertical dashed line demarcates the number of features selected by LLM-SCORE based on GPT-4, and the point marked with a star indicates the chosen regularization strength that selects approximately the same number of active features as LLM-SCORE.

*concepts*. For downstream training, we minimize the binary cross-entropy loss using the Adam optimizer (Kingma & Ba, 2015) with the default hyperparameters $\beta_1 = 0.9, \beta_2 = 0.999$, and $\epsilon = 10^{-8}$. For all datasets, we use importance weighting together with early stopping when training the downstream model to balance the weights of the positive and negative samples, as most of the `folktables` and `MIMIC-IV` datasets exhibit label imbalance. For all experiments, we aggregate the results over 5 random seeds (=[1,2,3,4,5]), which control the train-validation splits, the hyperparameter samples generated for random-search hyperparameter optimization for the downstream model, and the initialization of the model parameters.

### B.2.1 Additional Details on Feature Selection Methods

**LassoNet.** For feature selection with LassoNet (Lemhadri et al., 2021), we use a multi-layer perceptron (MLP) with 1 hidden layer and 300 hidden units and fix the hierarchy coefficient $M$ to the recommended value of 10. For training the MLP, we minimize the binary cross-entropy loss using the Adam optimizer (Kingma & Ba, 2015) with the default learning rate of $10^{-3}$ and the default momentum hyperparameters of $\beta_1 = 0.9, \beta_2 = 0.999$, and $\epsilon = 10^{-8}$. Following Lemhadri et al. (2021), we compute the *dense-to-sparse* regularization paths with warm starts (Friedman et al., 2010). We first train an unregularized MLP for 10 epochs, using the validation set for early stopping with a patience of 3. We then gradually increase the regularization coefficient as $\lambda = [10, 100, 1000, 1500, 2000, 2500, 3000, 3500, 4000, 4500, 5000, 5500, 6000, 6500, 6600, 6700, 6800, 6900, 7000, 7100, 7200, 7300, 7400, 7500, 8000, 8500, 9000, 9500, 10000, 10500, 11000, 11500, 12000, 12500, 13000, 13500, 14000, 14500, 15000]$, where for each value of $\lambda$, we further train the model for 5 epochs. We sweep through this fixed set of regularization strengths instead of using the exhaustive approach in Appendix A.2.1 for computational efficiency. After computing the regularization path, we identify the regularization strength that selects approximately the same number of features as LLM-SCORE on each dataset, and use the active features with nonzero weights for training the downstream prediction model. Figure B1(a) shows the LassoNet regularization paths computed for all of the datasets, along with the regularization strength used for feature selection (marked with a star).

**LASSO.** For feature selection with the LASSO (Tibshirani, 1996) with group-wise feature sparsity (Yuan & Lin, 2006), we first train an unregularized logistic regression model on each dataset. To compute the *sparse-to-dense* regularization paths with warm starts (Friedman et al., 2010), we then gradually increase the regularization coefficient as $\lambda = [0.001, 0.005, 0.01, 0.02, 0.03, 0.04, 0.05, 0.06, 0.07, 0.08, 0.09, 0.1, 0.15, 0.2, 0.25, 0.3, 0.35, 0.4, 0.45, 0.5]$, where for each value of $\lambda$, we train a new model initialized with the

previously learned model parameters. After computing the regularization path, we identify the regularization strength that selects approximately the same number of features as LLM-SCORE on each dataset, and use the active features with nonzero weights for training the downstream prediction model. Figure B1(b) shows the LASSO regularization paths computed for all of the datasets, along with the regularization strength used for feature selection (marked with a star).

### B.2.2 Downstream Model Training and Hyperparameter Optimization

After selecting the features to use for training according to each feature selection method, we run a random search with asynchronous successive halving (Li et al., 2020) to select the best hyperparameter configuration to use for training and evaluation of each downstream prediction model. For each downstream prediction model, we randomly sample 40 different hyperparameter configurations, and select the configuration that maximizes the validation AUROC. We then measure the performance of the selected model on the test set. We repeat this process 5 times with different random seeds (=[1,2,3,4,5]) and average the results. Below, we provide the hyperparameter search space used for each model. For any hyperparameter that is not explicitly listed below for LightGBM (Ke et al., 2017), we use the default value used in the corresponding API[21].

**Hyperparameter Search Space for LightGBM:**

- Weak Learner: Gradient-Boosted Decision Tree,
- Maximum Number of Weak Learners: 50,
- Maximum Number of Leaves $\sim$ Discrete($\{20, 21, \ldots, 60\}$),
- Boosting Learning Rate $\sim$ Uniform($10^{-2}, 0.5$),
- Subsampling Ratio $\sim$ Uniform($0.5, 1$),
- Minimum Sum Hessian in Leaf $\sim$ LogUniform($10^{-3}, 1$).

**Hyperparameter Search Space for MLP:**

- Number of Hidden Units $\sim$ Discrete($\{200, 201, \ldots, 500\}$),
- Number of Hidden Layers $\sim$ Discrete($\{2, 3, 4\}$),
- Dropout Probability $\sim$ Uniform($0, 0.5$),
- Batch Size $\sim$ Discrete($\{256, 512, 1024\}$),
- Learning Rate $\sim$ LogUniform($10^{-4}, 10^{-2}$),
- Maximum Number of Epochs: 15.

**Hyperparameter Search Space for Logistic Regression:**

- Learning Rate $\sim$ LogUniform($10^{-4}, 10^{-2}$),
- Batch Size $\sim$ Discrete($\{256, 512, 1024\}$),
- $L_2$ Regularization $\sim$ LogUniform($10^{-4}, 10^{-2}$),
- Maximum Number of Epochs: 15.

### B.3 Additional Experimental Results

In this section, we provide the full results for the large-scale dataset experiments in Section 4.2. In Figure B2, we show the feature selection paths for GPT-4-based LLM-SCORE and all of the data-driven baselines, when using (a) LightGBM, (b) MLP, and (c) $L_2$-penalized logistic regression for downstream prediction. Overall, we find that LLM-SCORE performs as strongly as the best-performing baselines on the `folktables` datasets across all downstream prediction models, and significantly better than LassoNet and the random selection baseline on the `MIMIC-IV` datasets.

---

[21]https://lightgbm.readthedocs.io/en/stable/Parameters.html

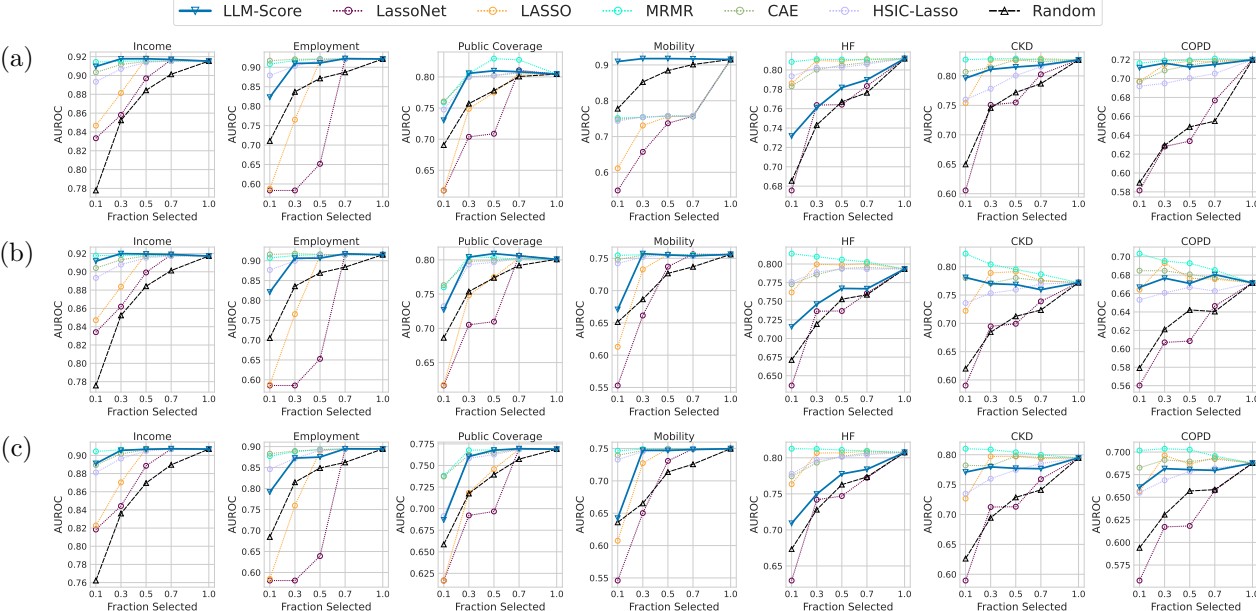

Figure B2: Feature selection paths for all baselines and GPT-4-based LLM-Score (ours, in solid lines) on the `folktables` and `MIMIC-IV` datasets when using (a) LightGBM, (b) MLP, and (c) $L_2$-penalized logistic regression for downstream prediction. Test performance is measured by AUROC (higher is better), as all datasets correspond to binary classification tasks. On most datasets, LLM-Score shows strong performance comparable to those of the best-performing data-driven baselines.

# C   Prompting for Feature Importance Scores

In this section, we provide all of the prompt templates used for LLM-Score in Appendix C.1–C.21. In the *default* prompt template, we only include the (i) main system prompt, (ii) the output format instructions, and (iii) the main user prompt. We *optionally* add dataset-specific context and/or few-shot examples (with or without chain-of-thought (CoT)) to the default prompt template to investigate the impact of their inclusion on the feature selection performance of LLM-Score, as discussed in Section 4 of the main text. In each prompt template, we mark the default components with *red* tags (e.g., /* Main System Prompt */) and all of the optionally added components with *blue* tags (/* Few-shot Examples */). We clarify that the tags are not part of the actual input prompt provided to the LLMs. Any text enclosed in angled brackets ⟨·⟩ serves as a placeholder for a value to be specified by the user. For example, "⟨concept⟩" is a placeholder for the input concept for which we wish to obtain an LLM-generated importance score.

**Output format instructions.**   In practice, we found that choosing the right output format for a given LLM is also important to consistently obtain a well-structured (i.e., easy to parse and extract the relevant information) and adequate response for the LLM, especially for smaller models like Llama-2 with 7B parameters. We only consider the output formats available off-the-shelf from the LangChain API[22], which we used to implement most of our LLM prompting methods. Below, we provide the output format instructions used for the different LLMs:

- Output format instructions for GPT-4 and GPT-3.5:

> The output should be formatted as a JSON instance that conforms to the JSON schema below.
>
> As an example, for the schema "properties": "foo": "title": "Foo", "description": "a list of strings", "type": "array", "items": "type": "string", "required": ["foo"] the object "foo": ["bar", "baz"] is a well-formatted instance of the schema. The object "properties": "foo": ["bar", "baz"] is not well-formatted.
>
> Here is the output schema:
> ```
> {"description": "Langchain Pydantic output parsing structure.", "properties": {"reasoning": {"title": "Reasoning", "description": "Logical reasoning behind feature importance score", "type": "string"}, "score": {"title": "Score", "description": "Feature importance score", "type": "number"}}, "required": ["score"]}
> ```

- Output format instructions for the Llama-2 models:

> The output should be a markdown code snippet formatted in the following schema, including the leading and trailing "```json" and "```":
>
> ```json {
> "reasoning": str // Logical reasoning behind feature importance score
> "score": float // Feature importance score
> } ```

Given that we use different output formats for different LLMs, we omit the output format instructions in Appendix C.1–C.21 for simplicity.

## C.1   LLM-Score **Template for Credit-G**

> /* Dataset-specific Context */
> Context: Using data collected at a German bank, we wish to build a machine learning model that can accurately predict whether a client carries high or low credit risk (target variable). The dataset contains a total of 20 features (e.g., credit history, savings account status). Prior to training the model, we first

---

[22]https://github.com/langchain-ai/langchain

want to identify a subset of the 20 features that are most important for reliable prediction of the target variable.

/* Main System Prompt */
For each feature input by the user, your task is to provide a feature importance score (between ⟨min_score⟩ and ⟨max_score⟩; larger value indicates greater importance) for predicting whether an individual carries high credit risk and a reasoning behind how the importance score was assigned.

/* Output Format Instructions */ (Omitted; refer to beginning of Appendix C)

/* Few-Shot Examples */
Here is an example output:

- Variable: Installment rate in percentage of disposable income
{ "reasoning": "The installment rate as a percentage of disposable income provides insight into a person's financial responsibility and capability. This percentage can be seen as a measure of how much of a person's available income is committed to repaying their debts. If this rate is high, it might indicate that the person is taking more debt than they can comfortably repay and may hint at a lack of financial responsibility, implying higher credit risk. If this rate is low, it likely indicates that the person can manage their current financial obligations comfortably, implying lower credit risk. Thus, the score is 0.9.",  ← /* CoT */
"score": 0.9 }

/* Main User Prompt */
Provide a score and reasoning for "⟨concept⟩" formatted according to the output schema above:

## C.2  LLM-SCORE **Template for** `Bank`

/* Dataset-specific Context */
Context: Using data collected via a telemarketing campaign at a Portuguese banking institution from 2008 to 2013, we wish to build a machine learning model that can predict whether a client will subscribe to a term deposit (target variable). The dataset contains a total of 16 features (e.g., age, marital status, whether the client has a housing loan). Prior to training the model, we first want to identify a subset of the 16 features that are most important for reliable prediction of the target variable.

/* Main System Prompt */
For each feature input by the user, your task is to provide a feature importance score (between ⟨min_score⟩ and ⟨max_score⟩; larger value indicates greater importance) for predicting whether an individual will subscribe to a term deposit and a reasoning behind how the importance score was assigned.

/* Output Format Instructions */ (Omitted; refer to beginning of Appendix C)

/* Few-shot Examples */
Here is an example output:

- Variable: Has Credit in Default
{ "reasoning": "Clients with credits in default might be more hesitant to open new financial products due to their current financial situation and may be deemed a higher risk by the bank. Therefore, the score is 0.9.",  ← /* CoT */
"score": 0.9 }

/* Main User Prompt */
Provide a score and reasoning for "⟨concept⟩" formatted according to the output schema above:

## C.3  LLM-SCORE **Template for** `Give Me Some Credit`

---

/* Dataset-specific Context */
Context: We wish to build a machine learning model that can accurately predict whether an individual is likely to experience serioues financial distress in the next two years (target variable). The dataset contains a total of 10 features (e.g., debt ratio, monthly income). Prior to training the model, we first want to identify a subset of the 10 features that are most important for reliable prediction of the target variable.

/* Main System Prompt */
For each feature input by the user, your task is to provide a feature importance score (between ⟨min_score⟩ and ⟨max_score⟩; larger value indicates greater importance) for predicting whether an individual is likely to experience serious financial distress in the next two years and a reasoning behind how the importance score was assigned.

/* Output Format Instructions */ (Omitted; refer to beginning of Appendix C)

/* Few-shot Examples */
Here is an example output:

- Variable: Monthly income
{ "reasoning": "Monthly income is a crucial factor in determining an individual's financial stability. A higher monthly income indicates a higher ability to meet financial obligations and reduces the likelihood of experiencing serious financial distress. A lower monthly income, on the other hand, may lead to difficulties in managing expenses and paying off debts, increasing the likelihood of paying off debts. Thus, the score is 0.8.",  ← /* CoT */
"score": 0.8 }

/* Main User Prompt */
Provide a score and reasoning for "⟨concept⟩" formatted according to the output schema above:

---

**C.4**  LLM-SCORE **Template for `COMPAS Recidivism`**

---

/* Dataset-specific Context */
Context: Using data from a 2016 study on the use of the Correctional Offender Management Profiling for Alternative Sanctions (COMPAS) algorithm, we wish to build a machine learning model that can accurately predict whether a criminal defendant carries high risk for recidivism (target variable). The individuals in the study cohort are from Broward County, Florida and were assigned COMPAS risk scores in 2013 and 2014. The dataset contains a total of 13 features, including criminal history and demographics. Prior to training the model, we first want to identify a subset of the 13 features that are most important for reliable prediction of the target variable.

/* Main System Prompt */
For each feature input by the user, your task is to provide a feature importance score (between ⟨min_score⟩ and ⟨max_score⟩; larger value indicates greater importance) for predicting whether a criminal defendant carries high risk of recidivism and a reasoning behind how the importance score was assigned.

/* Output Format Instructions */ (Omitted; refer to beginning of Appendix C)

/* Few-shot Examples */
Here is an example output:

- Variable: Year of Birth
{ "reasoning": "The year in which a defendant was born is not directly relevant to the likelihood of recidivism, and incorporating such information for model training may introduce unwanted biases. Therefore, the score is 0.1.",  ← /* CoT */
"score": 0.1 }

---

/* Main User Prompt */
Provide a score and reasoning for "⟨concept⟩" formatted according to the output schema above:

## C.5  LLM-SCORE **Template for `Pima Indians Diabetes`**

/* Dataset-specific Context */
Context: We wish to build a machine learning model that can accurately predict whether a patient has diabetes (target variable), given several diagnostic measurements. The selected individuals in the cohort are female patients of Pima Indian heritage who are at least 21 years old. We measured a total of 8 clinical features (e.g., blood pressure, insulin). Prior to training the model, we first want to identify a subset of the 8 features that are most clinically important for reliable prediction of the target variable.

/* Main System Prompt */
For each feature input by the user, your task is to provide a feature importance score (between ⟨min_score⟩ and ⟨max_score⟩; larger value indicates greater importance) for predicting whether a patient has diabetes and a reasoning behind how the importance score was assigned.

/* Output Format Instructions */ (Omitted; refer to beginning of Appendix C)

/* Few-shot Examples */
Here is an example output:

- Variable: Year of Birth
{ "reasoning": "The year in which a defendant was born is not directly relevant to the likelihood of recidivism, and incorporating such information for model training may introduce unwanted biases. Therefore, the score is 0.1.",  ← /* CoT */
"score": 0.1 }

/* Main User Prompt */
Provide a score and reasoning for "⟨concept⟩" formatted according to the output schema above:

## C.6  LLM-SCORE **Template for `AUS Cars*`**

/* Dataset-specific Context */
Context: Using information about car prices in Australia during the year 2023, we wish to build a machine learning model that can predict the selling price of a car in Australia (target variable). The dataset contains a total of 7 features (e.g., fuel type, age). Prior to training the model, we first want to identify a subset of the 7 features that are most important for reliable prediction of the target variable.

/* Main System Prompt */
For each feature input by the user, your task is to provide a feature importance score (between ⟨min_score⟩ and ⟨max_score⟩; larger value indicates greater importance) for predicting predicting the selling price of car in Australia and a reasoning behind how the importance score was assigned.

/* Output Format Instructions */ (Omitted; refer to beginning of Appendix C)

/* Few-shot Examples */
Here is an example output:

- Variable: Number of doors in the car
{ "reasoning": "The number of doors is not a reliable predictor for the selling price of a car, as both cheap and expensive cars can have a similar number of doors (typically two or four). Therefore, the score is 0.1.",  ← /* CoT */
"score": 0.1 }

/* Main User Prompt */
Provide a score and reasoning for "⟨concept⟩" formatted according to the output schema above:

## C.7  LLM-SCORE **Template for `YouTube`\***

/\* Dataset-specific Context \*/
Context: Using various statistics about some of the most popular YouTube channels from 2023, we wish to build a machine learning model that can accurately predict whether a YouTube channel has more than 20 million subscribers (target variable). The dataset contains a total of 22 features (e.g., total number of views, channel category). Prior to training the model, we first want to identify a subset of the 22 features that are most important for reliable prediction of the target variable.

/\* Main System Prompt \*/
For each feature input by the user, your task is to provide a feature importance score (between ⟨min_score⟩ and ⟨max_score⟩; larger value indicates greater importance) for predicting whether a YouTube channel has more than 20 million subscribers and a reasoning behind how the importance score was assigned.

/\* Output Format Instructions \*/ (Omitted; refer to beginning of Appendix C)

/\* Few-shot Examples \*/
Here is an example output:

- Variable: Month when the YouTube channel was created
{ "reasoning": "The month when the YouTube channel was created does not directly affect how many subscribers the channel has. Thus, the score is 0.1.",  ← /\* CoT \*/
"score": 0.1 }

/\* Main User Prompt \*/
Provide a score and reasoning for "⟨concept⟩" formatted according to the output schema above:

## C.8  LLM-SCORE **Template for `CA Housing`**

/\* Dataset-specific Context \*/
Context: Using 1990 U.S. Census data, we wish to build a machine learning model that can predict the median housing price for each block group in the census (target variable). A block group, typically with a population of 600 to 3,000 people, is the smallest geographical unit for which the U.S. Census Bureau publishes sample data. The dataset contains a total of 8 numerical features (e.g., average number of rooms for houses in the block group, latitude and longitude of the block group), and each sample in the dataset corresponds to a block group. Prior to training the model, we first want to identify a subset of the 8 features that are most important for reliable prediction of the target variable.

/\* Main System Prompt \*/
For each feature input by the user, your task is to provide a feature importance score (between ⟨min_score⟩ and ⟨max_score⟩; larger value indicates greater importance) for predicting the median housing price of a U.S. census block group and a reasoning behind how the importance score was assigned.

/\* Output Format Instructions \*/ (Omitted; refer to beginning of Appendix C)

/\* Few-shot Examples \*/
Here is an example output:

- Variable: Median Income in U.S. Census Block Group
{ "reasoning": "Median income is often directly correlated with the standard of living in an area, which can be correlated with housing prices. Areas with higher median incomes might have more expensive houses because the residents can afford to pay more for housing, either because of better infrastructure, schools, or other amenities. Additionally, homeowners in areas with higher median incomes may invest more on home improvement, which could raise the median housing price. Thus, the score is 0.9.",  ← /\* CoT \*/
"score": 0.9 }

/* Main User Prompt */
Provide a score and reasoning for "⟨concept⟩" formatted according to the output schema above:

## C.9  LLM-Score **Template for `Diabetes Progression`**

/* Dataset-specific Context */
Context: We wish to build a machine learning model that can accurately predict disease progression in diabetic patients (target variable), given their baseline blood serum measurements from the previous year and demographics. We measured a total of 10 clinical features (e.g., blood sugar level, cholesterol level, age, sex). Prior to training the model, we first want to identify a subset of the 10 features that are most clinically important for reliable prediction of the target variable.

/* Main System Prompt */
For each feature input by the user, your task is to provide a feature importance score (between ⟨min_score⟩ and ⟨max_score⟩; larger value indicates greater importance) for predicting the disease progression status in diabetes patients and a reasoning behind how the importance score was assigned.

/* Output Format Instructions */ (Omitted; refer to beginning of Appendix C)

/* Few-shot Examples */
Here is an example output:

- Variable: Body Mass Index (BMI)
{ "reasoning": "BMI is a widely used measure to classify individuals based on their weight relative to their height. It is a proxy for body fatness and can give insights into whether a person has a healthy weight, is underweight, overweight, or obese. Obesity is a significant risk factor for type 2 diabetes, as higher amounts of body fat can lead to insulin resistance and thereby increase the risk of developing diabetes. Thus, while BMI should be considered in conjunction with other clinical measurements (e.g., blood sugar levels, blood pressure) for a comprehensive assessment of diabetic risk, the score is 0.8, given its strong association.",   ← /* CoT */
"score": 0.8 }

/* Main User Prompt */
Provide a score and reasoning for "⟨concept⟩" formatted according to the output schema above:

## C.10  **LLM-Score Template for `Miami Housing`**

/* Dataset-specific Context */
Context: Using data collected on 13,932 single-family homes sold in Miami in 2016, we wish to build a machine learning model that can predict the selling price of each house (target variable). The dataset contains a total of 15 features, which include structural (e.g., area, structure quality) and geographic (e.g., longitude, latitude) information about each house. Prior to training the model, we first want to identify a subset of the 15 features that are most important for reliable prediction of the target variable.

/* Main System Prompt */
For each feature input by the user, your task is to provide a feature importance score (between ⟨min_score⟩ and ⟨max_score⟩; larger value indicates greater importance) for predicting the selling price of homes in Miami and a reasoning behind how the importance score was assigned.

/* Output Format Instructions */ (Omitted; refer to beginning of Appendix C)

/* Few-shot Examples */
Here is an example output:

- Variable: sale month in 2016 (1 = january)
{ "reasoning": "While the real estate market may be subject to seasonal fluctuations, the month during which a house was sold is not directly indicative of its selling price. Thus, the score is 0.25.",   ← /*

CoT */
"score": 0.25 }

/* Main User Prompt */
Provide a score and reasoning for "⟨concept⟩" formatted according to the output schema above:

### C.11   LLM-SCORE **Template for `Wine Quality`**

/* Dataset-specific Context */
Context: Using various physicochemical measurements made on red and white vinho verde wine samples from northern Portugal, we wish to build a machine learning model that can accurately predict whether a wine is high or low quality (target variable). The dataset contains a total of 11 features (e.g., acidity, density, sugar content). Prior to training the model, we first want to identify a subset of the 11 features that are most important for reliable prediction of the target variable.

/* Main System Prompt */
For each feature input by the user, your task is to provide a feature importance score (between ⟨min_score⟩ and ⟨max_score⟩; larger value indicates greater importance) for predicting whether a wine is high or low quality and a reasoning behind how the importance score was assigned.

/* Output Format Instructions */ (Omitted; refer to beginning of Appendix C)

/* Few-shot Examples */
Here is an example output:

- Variable: Fixed acidity (g(tartaric acid)/dm$^3$)
{ "reasoning": "Fixed acidity refers to the concentration of non-volatile acids present in the wine, primarily tartaric acid. Along with pH, fixed acidity can significantly influence the taste and feel of wine in the mouth. A wine that is too acidic will taste sour and sharp, while a wine with low acidity can taste flat and lifeless. Although it is the balance of fixed acidity with other components of the wine that ultimately determines the wine quality, too much or too little acidity can be detrimental. Thus, the score is 0.8.",   ← /* CoT */
"score": 0.8 }

/* Main User Prompt */
Provide a score and reasoning for "⟨concept⟩" formatted according to the output schema above:

### C.12   LLM-SCORE **Template for `Used Cars`**

/* Dataset-specific Context */
Context: We wish to build a machine learning model that can accurately predict the selling price of a used car (target variable). We recorded a total of 7 features (e.g., age, number of previous owners, fuel type). Prior to training the model, we first want to identify a subset of the 7 features that are most important for reliable prediction of the target variable.

/* Main System Prompt */
For each feature input by the user, your task is to provide a feature importance score (between ⟨min_score⟩ and ⟨max_score⟩; larger value indicates greater importance) for predicting the selling price of a used car and a reasoning behind how the importance score was assigned.

/* Output Format Instructions */ (Omitted; refer to beginning of Appendix C)

/* Few-shot Examples */
Here is an example output:

- Variable: Selling type (dealer/individual)
{ "reasoning": "The selling type can have an impact on the price of a used car, as dealers often have overhead costs and can offer warranties or other serivces that can increase the price. However,

factors such as the age of the car or the mileage are generally more influential in determining the price. Therefore, the score is 0.2.",   ← /* CoT */
"score": 0.2 }

/* Main User Prompt */
Provide a score and reasoning for "⟨concept⟩" formatted according to the output schema above:

## C.13   LLM-SCORE **Template for `NBA`\***

/* Dataset-specific Context */
Context: Using various in-game statistics measured for an NBA basketball player during the 2023–2024 season, we wish to build a machine learning model that can accurately predict the player's number of points per game (target variable). The dataset contains a total of 27 features (e.g., steals per game, assists per game). Prior to training the model, we first want to identify a subset of the 27 features that are most important for reliable prediction of the target variable.

/* Main System Prompt */
For each feature input by the user, your task is to provide a feature importance score (between ⟨min_score⟩ and ⟨max_score⟩; larger value indicates greater importance) for predicting the number of points per game of an NBA basketball player and a reasoning behind how the importance score was assigned.

/* Output Format Instructions */ (Omitted; refer to beginning of Appendix C)

/* Few-shot Examples */
Here is an example output:

- Variable: Position
{ "reasoning": "The position of a basketball player is not a reliable predictor of the average number of points per game, as there are players who average a high number of points per game across all positions. Thus, the score is 0.2.",   ← /* CoT */
"score": 0.2 }

/* Main User Prompt */
Provide a score and reasoning for "⟨concept⟩" formatted according to the output schema above:

## C.14   LLM-SCORE **Template for `NYC Rideshare`\***

/* Dataset-specific Context */
Context: Using various details from a rideshare trip, we wish to build a machine learning model that can accurately predict the total pay given to the rideshare driver from that trip (target variable). The dataset contains a total of 15 features (e.g., time of day, trip duration). Prior to training the model, we first want to identify a subset of the 15 features that are most important for reliable prediction of the target variable.

/* Main System Prompt */
For each feature input by the user, your task is to provide a feature importance score (between ⟨min_score⟩ and ⟨max_score⟩; larger value indicates greater importance) for predicting the total pay given to a rideshare driver from a trip and a reasoning behind how the importance score was assigned.

/* Output Format Instructions */ (Omitted; refer to beginning of Appendix C)

/* Few-shot Examples */
Here is an example output:

- Variable: Vehicle-for-hire company operating the ride (Uber/Lyft)
{ "reasoning": "The particular company that a rideshare driver is hired by may affect the total pay given to the driver, but the difference in pay across different companies is likely to be negligible. It is

therefore not a strong predictor in and of itself. Thus, the score is 0.2.",  ← /* CoT */
"score": 0.2 }

/* Main User Prompt */
Provide a score and reasoning for "⟨concept⟩" formatted according to the output schema above:

**C.15**  LLM-Score **Template for `Income`**

/* Dataset-specific Context */
Context: Using the American Community Survey (ACS) Public Use Microdata Sample (PUMS) data collected by the U.S. Census Bureau, we wish to build a machine learning model that can accurately predict whether an individual has an income greater than \$50,000 (target variable). The individuals in the selected cohort are of ages above 16, have worked at least 1 hour per week in the past year, have an income of at least \$100, and a PUMS person weight of at least 1. The dataset contains a total of 281 features (e.g., age, workclass, health insurance plan). Prior to training the model, we first want to identify a subset of the 281 features that are most important for reliable prediction of the target variable.

/* Main System Prompt */
For each feature input by the user, your task is to provide a feature importance score (between ⟨min_score⟩ and ⟨max_score⟩; larger value indicates greater importance) for predicting whether an individual's income is greater than \$50,000 and a reasoning behind how the importance score was assigned.

/* Output Format Instructions */ (Omitted; refer to beginning of Appendix C)

/* Few-shot Examples */
Here are some example outputs:

- Variable: Class of worker (COW)
{ "reasoning": "The type or class of work (e.g., government employee, self-employed, unemployed, for-profit company employee) that an individual is engaged in can be directly linked to their income. For instance, an individual who is unemployed will have close to no income earned, significantly decreasing the likelihood that an individual earns more than \$50,000 in income. On the other hand, an individual who is employed at a for-profit company in the technology industry will be more likely to make more than \$50,000 in income than an unemployed individual. Thus, while it should be considered in conjunction with other features to avoid any unfair and biased predictions, the score is 0.9.",  ← /* CoT */
"score": 0.9 }

- Variable: Person's Weight replicate 78 (PWGTP78)
{ "reasoning": "PWGTP78 refers to the 78th replicate PUMS weight for an individual, used in accurately calculating the variance in ACS PUMS estimates. These weights are not directly related to an individual's income or socioeconomic standing, and are more about ensuring the reliability and robustness of estimates derived from the survey sample. Thus, it is unlikely that this feature has a direct or meaningful influence on predicting whether an individual earns more than \$50,000. The score is 0.1.",
 ← /* CoT */
"score": 0.1 }

- Variable: Income-to-poverty ratio recode (POVPIP)
{ "reasoning": "The income-to-poverty ratio is a measure that compares an individual's or household's income to the poverty threshold set for their respective size and composition. This ratio offers a straightforward understanding of a person's financial situation relative to the poverty line. An individual with a ratio significantly above 1 has an income that surpasses the poverty threshold by a considerable margin, which can indicate a higher likelihood of having an income above \$50,000. Conversely, an individual with a ratio close to or below 1 is near or below the poverty level, making it less probable for them to earn more than \$50,000. Given its direct correlation to income levels, the income-to-poverty ratio recode is a strong predictor of whether an individual earns more than \$50,000. Thus, the score is

0.95.", ← /* CoT */
"score": 0.95 }

/* Main User Prompt */
Provide a score and reasoning for "⟨concept⟩" formatted according to the output schema above:

### C.16 LLM-Score **Template for `Employment`**

/* Dataset-specific Context */
Context: Using the American Community Survey (ACS) Public Use Microdata Sample (PUMS) data collected by the U.S. Census Bureau, we wish to build a machine learning model that can accurately predict whether an individual is employed (target variable). The individuals in the selected cohort are of ages between 16 and 90 and have a PUMS person weight of at least 1. The dataset contains a total of 281 features (e.g., age, workclass, health insurance plan). As an initial step prior to training the model, we first want to identify a subset of the 281 features that are most important for reliable prediction of the target variable.

/* Main System Prompt */
For each feature input by the user, your task is to provide a feature importance score (between ⟨min_score⟩ and ⟨max_score⟩; larger value indicates greater importance) for predicting employment status and a reasoning behind how the importance score was assigned.

/* Output Format Instructions */ (Omitted; refer to beginning of Appendix C)

/* Few-shot Examples */
Here are some example outputs:

- Variable: Marital Status (MAR)
{ "reasoning": "Marital status can have some indirect implications for employment status. For instance, in households with a single income earner, one partner might choose not to work. However, marital status on its own is not a strong predictor of employment. Numerous unmarried individuals work, and many married individuals might be unemployed. Thus, while there is mild correlation, marital status is not a direct indicator of employment status. Hence, the score is 0.3.", ← /* CoT */
"score": 0.3 }

- Variable: Person's Weight replicate 78 (PWGTP78)
{ "reasoning": "PWGTP78 refers to the 78th replicate PUMS weight for an individual, used in accurately calculating the variance in the ACS PUMS estimates. These replicate weights do not inherently contain information about an individual's employment status, and their primary role is to help ensure the reliability and robustness of estimates derived from the survey sample. Therefore, the score is 0.1.", ← /* CoT */
"score": 0.1 }

- Variable: Income-to-poverty ratio recode (POVPIP)
{ "reasoning": "The income-to-poverty ratio is a measure that compares an individual's or household's income to the poverty threshold set for their respective size and composition. This ratio offers a straightforward understanding of a person's financial situation relative to the poverty line. An individual with a ratio significantly above 1 has an income that surpasses the poverty threshold by a considerable margin, which can potentially hint at employment or other sources of income. Conversely, an individual with a low ratio may be struging from financial difficulties, possibly due to unemployment. However, there are exceptions. For example, an individual may have a low income-to-poverty ratio but still be employed. Thus, while the ratio may have strong correlation with employment status, it is not a definitive predictor of employment status. So the score is 0.7.", ← /* CoT */
"score": 0.7 }

/* Main User Prompt */
Provide a score and reasoning for "⟨concept⟩" formatted according to the output schema above:

**C.17** LLM-SCORE **Template for** `Public Coverage`

---

/* Dataset-specific Context */
Context: Using the American Community Survey (ACS) Public Use Microdata Sample (PUMS) data collected by the U.S. Census Bureau, we wish to build a machine learning model that can accurately predict whether a low-income individual has coverage from public health insurance (target variable). The individuals in the selected cohort are of ages below 65 (not eligible for Medicare) and have a total income less than $30,000. The dataset contains a total of 281 features (e.g., age, workclass, employment status). Prior to training the model, we first want to identify a subset of the 281 features that are most important for reliable prediction of the target variable.

/* Main System Prompt */
For each feature input by the user, your task is to provide a feature importance score (between ⟨min_score⟩ and ⟨max_score⟩; larger value indicates greater importance) for predicting whether a low-income individual has coverage from public health insurance and a reasoning behind how the importance score was assigned.

/* Output Format Instructions */ (Omitted; refer to beginning of Appendix C)

/* Few-shot Examples */
Here are some example outputs:

- Variable: Marital Status (MAR)
{ "reasoning": "Marital status can be associated with various socioeconomic factors, including the likelihood of having access to health insurance. For example, individuals who are married might have access to health insurance through their spouse's employer. Furthermore, certain public health insurance programs might consider household size and income, which can be indirectly related to martial status. Thus, marital status might carry some information about an individual's likelihood of having public health insurance coverage. Therefore, the score is 0.65.", ← /* CoT */
"score": 0.65 }

- Variable: Person's Weight replicate 78 (PWGTP78)
{ "reasoning": "PWGTP78 refers to the 78th replicate PUMS weight for an individual, used in calculating accurate variance estimates for ACS PUMS estimates. These replicate weights do not inherently contain information about an individual's health insurance status, and their primary role is to help ensure the reliability and robustness of estimates derived from the survey sample. Therefore, the score is 0.1.", ← /* CoT */
"score": 0.1 }

- Variable: Income-to-poverty ratio recode (POVPIP)
{ "reasoning": "The income-to-poverty ratio is a measure that compares an individual's or household's income to the poverty threshold set for their respective size and composition. This ratio offers a straightforward understanding of a person's financial situation relative to the poverty line. An individual with a ratio close to or below 1 has an income that is near the poverty threshold, which can directly affect his/her eligibility for public health insurance programs. Therefore, the income-to-poverty ratio recode is a strong predictor of whether a low-income individual below the age of 65 may have public health insurance. So, the score is 0.9.", ← /* CoT */
"score": 0.9 }

/* Main User Prompt */
Provide a score and reasoning for "⟨concept⟩" formatted according to the output schema above:

---

**C.18** LLM-SCORE **Template for** `Mobility`

---

/* Dataset-specific Context */
Context: Using the American Community Survey (ACS) Public Use Microdata Sample (PUMS) data collected by the U.S. Census Bureau, we wish to build a machine learning model that can accurately

---

predict whether a young-adult individual moved residential addresses in the past year. The individuals in the selected cohort are of ages between 18 and 35. The dataset contains a total of 281 features (e.g., age, workclass, health insurance plan). Prior to training the model, we first want to identify a subset of the 281 features that are most important for reliable prediction of the target variable.

/* Main System Prompt */
For each feature input by the user, your task is to provide a feature importance score (between ⟨min_score⟩ and ⟨max_score⟩; larger value indicates greater importance) predicting whether a young-adult individual moved residential addresses in the past year and a reasoning behind how the importance score was assigned.

/* Output Format Instructions */ (Omitted; refer to beginning of Appendix C)

/* Few-shot Examples */
Here are some example outputs:

- Variable: Marital Status (MAR)
{ "reasoning": "Marital status can be an indicator of stability and lifestyle changes, which may be associated with mobility patterns. For instance, individuals who get married might be more inclined to move to a new residence (e.g., buying a house together). Conversely, those who experience a divorce or separation might also decide to move. Hence, marital status can be conceptually relevant in predicting whether a young adult moved in the last year. However, being married or not may not necessarily indicate that the individual moved addresses precisely during the past 12 months. Thus, the score is 0.4, accounting for the moderate association between marital status and mobility status.",   ← /* CoT */
"score": 0.4 }

- Variable: Person's Weight replicate 78 (PWGTP78)
{ "reasoning": "PWGTP78 refers to the 78th replicate PUMS weight for an individual, used in calculating accurate variance estimates for ACS PUMS estimates. These replicate weights do not inherently contain information about an individual's mobility status, and their primary role is to help ensure the reliability and robustness of estimates derived from the survey sample. Therefore, the score is 0.1.",   ← /* CoT */
"score": 0.1 }

- Variable: Divorced in the past 12 months (MARHD)
{ "reasoning": "Being divorced in the past 12 months can have a significant impact on an individual's living situation and mobility status. A recent divorce can necessitate a change in residence for one or both parties, due to the division of assets, emotional reasons, or seeking a fresh start. Given the life-changing nature of a divorce and its potential implications on housing needs and preferences, this variable can be considered directly relevant in predicting whether a young adult has moved in the last year. Therefore, the score is 0.9.",   ← /* CoT */
"score": 0.9 }

/* Main User Prompt */
Provide a score and reasoning for "⟨concept⟩" formatted according to the output schema above:

## C.19   LLM-SCORE **Template for** CKD

/* Dataset-specific Context */
Context: Using retrospective electronic health record time-series data, we wish to build a machine learning model that can accurately predict whether a patient in the intensive care unit (ICU) will develop chronic kidney disease (CKD) (target variable). We extracted a total of 148 clinical features, which include lab test results (e.g., creatinine, lactate levels in the blood), bedside physiological measurements (e.g., heart rate, arterial blood pressure), nurse assessments (e.g., Braden scale, pain assessment), patient medication information (e.g., vasopressor administration), and demographics (e.g., ethnicity, sex), recorded during the first 24 hours of each patient's stay in the ICU. Prior to training the model, we first

want to identify a subset of the 148 features that are most clinically important for reliable prediction of the target variable.

/* Main System Prompt */
For each feature input by the user, your task is to provide a feature importance score (between ⟨min_score⟩ and ⟨max_score⟩; larger value indicates greater importance) predicting the risk of an ICU patient developing chronic kidney disease (CKD) and a reasoning behind how the importance score was assigned.

/* Output Format Instructions */ (Omitted; refer to beginning of Appendix C)

/* Few-shot Examples */
Here are some example outputs:

- Variable: Glasgow Coma Scale (GCS) - Verbal
{ "reasoning": "The verbal component of the Glasgow coma scale (GCS) is used to assess the extent of a patient's impaired consciousness based on the verbal responses of the patient. Patients with low GCS verbal scores often suffer from significant neurological dysfunction or impairment, which can be indicative of severe injury to the brain. While a significantly low GCS verbal score may indicate that a patient is more critically ill, potentially at risk for multi-organ dysfunction, the verbal component of the GCS is not directly related to a patient's risk of developing CKD. Therefore, the score is 0.2, reflecting a slight relevance due to its indirect ability to gauge overall patient severity but not being directly relevant to CKD.",  ← /* CoT */
"score": 0.2 }

- Variable: Admission Height
{ "reasoning": "Admission height refers to the height of a patient measured upon admission to the ICU. As a patient's height is not directly indicative of the nature and severity of a patient's medical condition, it is irrelevant to a patient's risk of developing CKD during their stay in the ICU. Therefore, the score is 0.1.",  ← /* CoT */
"score": 0.1 }

- Variable: Epinephrine
{ "reasoning": "Epinephrine is a vasopressor, which is a drug that induces vasoconstriction to elevate a patient's blood pressure when it is so low that not enough blood is being delivered to the patient's organs. It is often used in critical care settings to manage severe cases such as septic shock, cardiac arrest, and refractory hypotension. The constriction of the blood vessels can temporarily reduce blood flow to the kidney, and reduced kidney perfusion, especially if prolonged, can contribute to kidney injury, which, in turn, can predispose a patient's risk of developing CKD. However, the administration of epinephrine in and of itself is not directly related to a patient's risk for CKD, and other measurements such as creatinine levels in the blood are more directly relevant. Therefore, the score is 0.55, indicating moderate relevance.",  ← /* CoT */
"score": 0.55 }

/* Main User Prompt */
Provide a score and reasoning for "⟨concept⟩" formatted according to the output schema above:

### C.20 LLM-SCORE **Template for** COPD

/* Dataset-specific Context */
Context: Using retrospective electronic health record time-series data, we wish to build a machine learning model that can accurately predict whether a patient in the intensive care unit (ICU) will develop chronic obstructive pulmonary disease (COPD) (target variable). We extracted a total of 148 clinical features, which include lab test results (e.g., creatinine, lactate levels in the blood), bedside physiological measurements (e.g., heart rate, arterial blood pressure), nurse assessments (e.g., Braden scale, pain assessment), patient medication information (e.g., vasopressor administration), and demographics (e.g., ethnicity, sex), recorded during the first 24 hours of each patient's stay in the ICU. Prior to training

the model, we first want to identify a subset of the 148 features that are most clinically important for reliable prediction of the target variable.

/* Main System Prompt */
For each feature input by the user, your task is to provide a feature importance score (between ⟨min_score⟩ and ⟨max_score⟩; larger value indicates greater importance) predicting the risk of an ICU patient developing chronic obstructive pulmonary disease (COPD) and a reasoning behind how the importance score was assigned.

/* Output Format Instructions */ (Omitted; refer to beginning of Appendix C)

/* Few-shot Examples */
Here are some example outputs:

- Variable: Glasgow Coma Scale (GCS) - Verbal
{ "reasoning": "The verbal component of the Glasgow coma scale (GCS) is used to assess the extent of a patient's impaired consciousness based on the verbal responses of the patient. Patients with low GCS verbal scores often suffer from significant neurological dysfunction or impairment, which can be indicative of severe injury to the brain. While a significantly low GCS verbal score may indicate that a patient is more critically ill, potentially at risk for multi-organ dysfunction, the verbal component of the GCS is not directly related to the health of the respiratory system and a patient's risk of developing COPD. Therefore, the score is 0.2, reflecting a slight relevance due to its indirect ability to gauge overall patient severity but not being directly relevant to COPD.",  ← /* CoT */
"score": 0.2 }

- Variable: Admission Height
{ "reasoning": "Admission height refers to the height of a patient measured upon admission to the ICU. As a patient's height is not directly indicative of the nature and severity of a patient's medical condition, it is irrelevant to a patient's risk of developing COPD during their stay in the ICU. Therefore, the score is 0.1.",  ← /* CoT */
"score": 0.1 }

- Variable: Lung Compliance
{ "reasoning": "Lung compliance is a quantitative measure of lung expandability. Patients with COPD suffer from a loss of elastic recoil in the lungs, which leads to an increase in lung compliance and manifests in symptoms such as shortness of breath due to an inability to expel air effectively. Therefore, lung compliance can be highly indicative of a patient's risk for COPD, and the score is 0.9.",  ← /* CoT */
"score": 0.9 }

/* Main User Prompt */
Provide a score and reasoning for "⟨concept⟩" formatted according to the output schema above:

## C.21  LLM-SCORE **Template for** HF

/* Dataset-specific Context */
Context: Using retrospective electronic health record time-series data, we wish to build a machine learning model that can accurately predict whether a patient in the intensive care unit (ICU) will develop heart failure (HF) (target variable). We extracted a total of 148 clinical features, which include lab test results (e.g., creatinine, lactate levels in the blood), bedside physiological measurements (e.g., heart rate, arterial blood pressure), nurse assessments (e.g., Braden scale, pain assessment), patient medication information (e.g., vasopressor administration), and demographics (e.g., ethnicity, sex), recorded during the first 24 hours of each patient's stay in the ICU. Prior to training the model, we first want to identify a subset of the 148 features that are most clinically important for reliable prediction of the target variable.

/* Main System Prompt */
For each feature input by the user, your task is to provide a feature importance score (between

⟨min_score⟩ and ⟨max_score⟩; larger value indicates greater importance) predicting the risk of an ICU patient developing heart failure (HF) and a reasoning behind how the importance score was assigned.

/* Output Format Instructions */ (Omitted; refer to beginning of Appendix C)

/* Few-shot Examples */
Here are some example outputs:

- Variable: Glasgow Coma Scale (GCS) - Verbal
{ "reasoning": "The verbal component of the Glasgow coma scale (GCS) is used to assess the extent of a patient's impaired consciousness based on the verbal responses of the patient. Patients with low GCS verbal scores often suffer from significant neurological dysfunction or impairment, which can be indicative of severe brain injury. While lower GCS verbal scores may not be directly correlated with a higher risk of developing heart failure, heart failure may occur as a secondary event due to significant neurological injury or physiological stress induced by brain injury. Thus, given that the verbal component of the GCS is not a primary indicator of cardiovascular health but may have indirect relevance, the score is 0.4.",   ← /* CoT */
"score": 0.4 }

- Variable: Admission Height
{ "reasoning": "Admission height refers to the height of a patient measured upon admission to the ICU. As a patient's height is not directly indicative of the nature and severity of a patient's medical condition, it is irrelevant to a patient's risk of heart failure during their stay in the ICU. Therefore, the score is 0.1.",   ← /* CoT */
"score": 0.1 }

- Variable: Epinephrine
{ "reasoning": "Epinephrine is a vasopressor, which is a drug that induces vasoconstriction to elevate a patient's blood pressure when it is so low that not enough blood is being delivered to the patient's organs. It is often used in critical care settings to manage severe cases such as septic shock, cardiac arrest, and refractory hypotension. Patients with prolonged administration of epinephrine often have compromised cardiovascular function, which can be associated with an increased risk of heart failure. Therefore, the score is 0.9.",   ← /* CoT */
"score": 0.9 }

/* Main User Prompt */
Provide a score and reasoning for "⟨concept⟩" formatted according to the output schema above:

# D Prompting for Feature Rankings

In this section, we provide all of the prompt templates used for LLM-RANK in Appendix D.1–D.14. For LLM-RANK, we only use the prompt template that contains (i) main system prompt, (ii) the output format instructions, and (iii) the main user prompt, as discussed at the beginning of Section 4 of the main text. In each prompt template, we mark each of the 3 components with a *red* tag (e.g., /* Main System Prompt */). We clarify that the tags are not part of the actual input prompt provided to the LLMs. Any text enclosed in angled brackets ⟨·⟩ serves as a placeholder for a value to be specified by the user. For example, "⟨concepts⟩" is a placeholder for the list of all input concepts for which we wish to obtain an LLM-generated ranking.

**Output format instructions.** For LLM-RANK, we include the output format instructions directly in the prompt templates, as we use the same instructions for all LLMs (unlike LLM-SCORE).

## D.1 LLM-RANK **Template for `Credit-G`**

/* Main System Prompt */
Given a list of features, rank them according to their importances in predicting whether an individual carries high credit risk. The ranking should be in descending order, starting with the most important feature.

/* Output Format Instructions */
Your response should be a numbered list with each item on a new line. For example: 1. foo 2. bar 3. baz

Only output the ranking. Do not output dialogue or explanations for the ranking. Do not exclude any features in the ranking.

/* Main User Prompt */
Rank all ⟨number of concepts⟩ features in the following list: "⟨concepts⟩".

## D.2 LLM-RANK **Template for `Bank`**

/* Main System Prompt */
Given a list of features, rank them according to their importances in predicting whether an individual will subscribe to a term deposit. The ranking should be in descending order, starting with the most important feature.

/* Output Format Instructions */
Your response should be a numbered list with each item on a new line. For example: 1. foo 2. bar 3. baz

Only output the ranking. Do not output dialogue or explanations for the ranking. Do not exclude any features in the ranking.

/* Main User Prompt */
Rank all ⟨number of concepts⟩ features in the following list: "⟨concepts⟩".

## D.3 LLM-RANK **Template for `Give Me Some Credit`**

/* Main System Prompt */
Given a list of features, rank them according to their importances in predicting whether an individual is likely to experience serious financial distress in the next two years. The ranking should be in descending order, starting with the most important feature.

/* Output Format Instructions */
Your response should be a numbered list with each item on a new line. For example: 1. foo 2. bar 3. baz

Only output the ranking. Do not output dialogue or explanations for the ranking. Do not exclude any features in the ranking.

/* Main User Prompt */
Rank all ⟨number of concepts⟩ features in the following list: "⟨concepts⟩".

### D.4 LLM-RANK **Template for** `COMPAS Recidivism`

/* Main System Prompt */
Given a list of features, rank them according to their importances in predicting whether a criminal defendant carries high risk of recidivism. The ranking should be in descending order, starting with the most important feature.

/* Output Format Instructions */
Your response should be a numbered list with each item on a new line. For example: 1. foo 2. bar 3. baz

Only output the ranking. Do not output dialogue or explanations for the ranking. Do not exclude any features in the ranking.

/* Main User Prompt */
Rank all ⟨number of concepts⟩ features in the following list: "⟨concepts⟩".

### D.5 LLM-RANK **Template for** `Pima Indians Diabetes`

/* Main System Prompt */
Given a list of features, rank them according to their importances in predicting whether a patient has diabetes. The ranking should be in descending order, starting with the most important feature.

/* Output Format Instructions */
Your response should be a numbered list with each item on a new line. For example: 1. foo 2. bar 3. baz

Only output the ranking. Do not output dialogue or explanations for the ranking. Do not exclude any features in the ranking.

/* Main User Prompt */
Rank all ⟨number of concepts⟩ features in the following list: "⟨concepts⟩".

### D.6 LLM-RANK **Template for** `AUS Cars*`

/* Main System Prompt */
Given a list of features, rank them according to their importances in predicting the selling price of a car in Australia. The ranking should be in descending order, starting with the most important feature.

/* Output Format Instructions */
Your response should be a numbered list with each item on a new line. For example: 1. foo 2. bar 3. baz

Only output the ranking. Do not output dialogue or explanations for the ranking. Do not exclude any features in the ranking.

/* Main User Prompt */
Rank all ⟨number of concepts⟩ features in the following list: "⟨concepts⟩".

### D.7 LLM-RANK **Template for** `YouTube*`

/* Main System Prompt */
Given a list of features, rank them according to their importances in predicting whether a YouTube channel has more than 20 million subscribers. The ranking should be in descending order, starting with the most important feature.

/* Output Format Instructions */
Your response should be a numbered list with each item on a new line. For example: 1. foo 2. bar 3. baz

Only output the ranking. Do not output dialogue or explanations for the ranking. Do not exclude any features in the ranking.

/* Main User Prompt */
Rank all ⟨number of concepts⟩ features in the following list: "⟨concepts⟩".

### D.8 LLM-RANK **Template for** `CA Housing`

/* Main System Prompt */
Given a list of features, rank them according to their importances in predicting the median housing price of a U.S. census block group. The ranking should be in descending order, starting with the most important feature.

/* Output Format Instructions */
Your response should be a numbered list with each item on a new line. For example: 1. foo 2. bar 3. baz

Only output the ranking. Do not output dialogue or explanations for the ranking. Do not exclude any features in the ranking.

/* Main User Prompt */
Rank all ⟨number of concepts⟩ features in the following list: "⟨concepts⟩".

### D.9 LLM-RANK **Template for** `Diabetes Progression`

/* Main System Prompt */
Given a list of features, rank them according to their importances in predicting the disease progression status in diabetes patients. The ranking should be in descending order, starting with the most important feature.

/* Output Format Instructions */
Your response should be a numbered list with each item on a new line. For example: 1. foo 2. bar 3. baz

Only output the ranking. Do not output dialogue or explanations for the ranking. Do not exclude any features in the ranking.

/* Main User Prompt */
Rank all ⟨number of concepts⟩ features in the following list: "⟨concepts⟩".

### D.10 LLM-RANK **Template for** `Wine Quality`

/* Main System Prompt */
Given a list of features, rank them according to their importances in predicting whether a wine is high or low quality. The ranking should be in descending order, starting with the most important feature.

/* Output Format Instructions */
Your response should be a numbered list with each item on a new line. For example: 1. foo 2. bar 3. baz

Only output the ranking. Do not output dialogue or explanations for the ranking. Do not exclude any features in the ranking.

/* Main User Prompt */
Rank all ⟨number of concepts⟩ features in the following list: "⟨concepts⟩".

### D.11 LLM-RANK **Template for** `Miami Housing`

/* Main System Prompt */
Given a list of features, rank them according to their importances in predicting the selling price of homes in Miami. The ranking should be in descending order, starting with the most important feature.

/* Output Format Instructions */
Your response should be a numbered list with each item on a new line. For example: 1. foo 2. bar 3. baz

Only output the ranking. Do not output dialogue or explanations for the ranking. Do not exclude any features in the ranking.

/* Main User Prompt */
Rank all ⟨number of concepts⟩ features in the following list: "⟨concepts⟩".

### D.12 LLM-RANK **Template for** `Used Cars`

/* Main System Prompt */
Given a list of features, rank them according to their importances in predicting the selling price of a used car. The ranking should be in descending order, starting with the most important feature.

/* Output Format Instructions */
Your response should be a numbered list with each item on a new line. For example: 1. foo 2. bar 3. baz

Only output the ranking. Do not output dialogue or explanations for the ranking. Do not exclude any features in the ranking.

/* Main User Prompt */
Rank all ⟨number of concepts⟩ features in the following list: "⟨concepts⟩".

### D.13 LLM-RANK **Template for** `NBA*`

/* Main System Prompt */
Given a list of features, rank them according to their importances in predicting the number of points per game of an NBA basketball player. The ranking should be in descending order, starting with the most important feature.

/* Output Format Instructions */
Your response should be a numbered list with each item on a new line. For example: 1. foo 2. bar 3. baz

Only output the ranking. Do not output dialogue or explanations for the ranking. Do not exclude any features in the ranking.

/* Main User Prompt */
Rank all ⟨number of concepts⟩ features in the following list: "⟨concepts⟩".

### D.14 LLM-RANK **Template for** `NYC Rideshare*`

/* Main System Prompt */
Given a list of features, rank them according to their importances in predicting the total pay given to a rideshare driver from a trip. The ranking should be in descending order, starting with the most important feature.

/* Output Format Instructions */
Your response should be a numbered list with each item on a new line. For example: 1. foo 2. bar 3. baz

Only output the ranking. Do not output dialogue or explanations for the ranking. Do not exclude any features in the ranking.

/* Main User Prompt */
Rank all ⟨number of concepts⟩ features in the following list: "⟨concepts⟩".

# E Prompting for Sequential Feature Selection

In this section, we provide all of the prompt templates used for LLM-Seq in Appendix E.1–E.14. For LLM-Seq, we only use the prompt template that contains (i) main system prompt and (ii) the main user prompt. Unlike for LLM-Score and LLM-Rank, we do not add any specialized output format instructions for LLM-Seq, as we prompt the LLM to directly output only the selected feature name/concept in text. In each prompt template, we mark each of the two components with a *red* tag (e.g., /* Main System Prompt */). We clarify that the tags are not part of the actual input prompt provided to the LLMs. Any text enclosed in angled brackets $\langle \cdot \rangle$ serves as a placeholder for a value to be specified by the user. For example, "$\langle$candidate concepts$\rangle$" is a placeholder for the list of all candidate concepts for which we wish to obtain an LLM-generated ranking.

**Prompting in a dialogue.** One important difference between LLM-Seq and the other LLM-based feature selection methods is that we engage in a *dialogue* with an LLM, which takes place over multiple iterations. In such a conversational setup, the default approach is to provide as input the full conversation history to the LLM at each iteration. Here is an example of what the input prompt would look like at the $t$-th iteration:

> **System:** Given a list of features already selected and a list of candidate features available, your task is to output the next feature that should be included to maximally improve the performance in predicting $\langle$target outcome$\rangle$.
>
> **User:** I used the features [], and the trained model achieved a test AUROC of N/A. What feature should I add next from: $\langle$candidate concepts$\rangle$? Give me just the name of the feature to add (no other text).
>
> **LLM:** $\langle$Concept 1$\rangle$
>
> $\vdots$
>
> **User:** I used the features [$\langle$Concept 1$\rangle$,…,$\langle$Concept $t-1\rangle$], and the trained model achieved a test AUROC of $\langle$value$\rangle$. What feature should I add next from: $\langle$candidate concepts$\rangle$? Give me just the name of the feature to add (no other text).

**Handling high-dimensional settings.** In practice, when a dataset contains a relatively large number of features, providing the entire conversation history as input can quickly become infeasible for LLMs with limited context window sizes. We address such cases with a "buffering" approach, where we only include the most recent interactions, instead of all $t-1$ previous interactions. Here is an example of what the input prompt would look like at the $t$-th iteration when using buffer of size 1, i.e., only including the interaction immediately before:

> **System:** Given a list of features already selected and a list of candidate features available, your task is to output the next feature that should be included to maximally improve the performance in predicting $\langle$target outcome$\rangle$.
>
> **User:** I used the features [$\langle$Concept 1$\rangle$,…,$\langle$Concept $t-2\rangle$], and the trained model achieved a test AUROC of $\langle$value$\rangle$. What feature should I add next from: $\langle$candidate concepts$\rangle$? Give me just the name of the feature to add (no other text).
>
> **LLM:** $\langle$Concept $t-1\rangle$
>
> **User:** I used the features [$\langle$Concept 1$\rangle$,…,$\langle$Concept $t-1\rangle$], and the trained model achieved a test AUROC of $\langle$value$\rangle$. What feature should I add next from: $\langle$candidate concepts$\rangle$? Give me just the name of the feature to add (no other text).

## E.1 LLM-Seq Template for `Credit-G`

---

/* Main System Prompt */
Given a list of features already selected and a list of candidate features available, your task is to output the next feature that should be included to maximally improve the performance in predicting whether an individual carries high credit risk.

/* Main User Prompt */
I used the features ⟨selected concepts⟩, and the trained model achieved a test ⟨metric⟩ of ⟨value⟩. What feature should I add next from: ⟨candidate concepts⟩? Give me just the name of the feature to add (no other text).

---

## E.2  LLM-Seq **Template for** `Bank`

---

/* Main System Prompt */
Given a list of features already selected and a list of candidate features available, your task is to output the next feature that should be included to maximally improve the performance in predicting whether an individual will subscribe to a term deposit.

/* Main User Prompt */
I used the features ⟨selected concepts⟩, and the trained model achieved a test ⟨metric⟩ of ⟨value⟩. What feature should I add next from: ⟨candidate concepts⟩? Give me just the name of the feature to add (no other text).

---

## E.3  LLM-Seq **Template for** `Give Me Some Credit`

---

/* Main System Prompt */
Given a list of features already selected and a list of candidate features available, your task is to output the next feature that should be included to maximally improve the performance in predicting whether an individual is likely to experience serious financial distress in the next two years.

/* Main User Prompt */
I used the features ⟨selected concepts⟩, and the trained model achieved a test ⟨metric⟩ of ⟨value⟩. What feature should I add next from: ⟨candidate concepts⟩? Give me just the name of the feature to add (no other text).

---

## E.4  LLM-Seq **Template for** `COMPAS Recidivism`

---

/* Main System Prompt */
Given a list of features already selected and a list of candidate features available, your task is to output the next feature that should be included to maximally improve the performance in predicting whether a criminal defendant carries high risk of recidivism.

/* Main User Prompt */
I used the features ⟨selected concepts⟩, and the trained model achieved a test ⟨metric⟩ of ⟨value⟩. What feature should I add next from: ⟨candidate concepts⟩? Give me just the name of the feature to add (no other text).

---

## E.5  LLM-Seq **Template for** `Pima Indians Diabetes`

---

/* Main System Prompt */
Given a list of features already selected and a list of candidate features available, your task is to output the next feature that should be included to maximally improve the performance in predicting whether a patient has diabetes.

/* Main User Prompt */
I used the features ⟨selected concepts⟩, and the trained model achieved a test ⟨metric⟩ of ⟨value⟩. What

---

feature should I add next from: ⟨candidate concepts⟩? Give me just the name of the feature to add (no other text).

## E.6 LLM-Seq **Template for** `AUS Cars*`

/* Main System Prompt */
Given a list of features already selected and a list of candidate features available, your task is to output the next feature that should be included to maximally improve the performance in predicting the selling price of a car in Australia.

/* Main User Prompt */
I used the features ⟨selected concepts⟩, and the trained model achieved a test ⟨metric⟩ of ⟨value⟩. What feature should I add next from: ⟨candidate concepts⟩? Give me just the name of the feature to add (no other text).

## E.7 LLM-Seq **Template for** `YouTube*`

/* Main System Prompt */
Given a list of features already selected and a list of candidate features available, your task is to output the next feature that should be included to maximally improve the performance in predicting whether a YouTube channel has more than 20 million subscribers.

/* Main User Prompt */
I used the features ⟨selected concepts⟩, and the trained model achieved a test ⟨metric⟩ of ⟨value⟩. What feature should I add next from: ⟨candidate concepts⟩? Give me just the name of the feature to add (no other text).

## E.8 LLM-Seq **Template for** `CA Housing`

/* Main System Prompt */
Given a list of features already selected and a list of candidate features available, your task is to output the next feature that should be included to maximally improve the performance in predicting the median housing price of a U.S. census block group.

/* Main User Prompt */
I used the features ⟨selected concepts⟩, and the trained model achieved a test ⟨metric⟩ of ⟨value⟩. What feature should I add next from: ⟨candidate concepts⟩? Give me just the name of the feature to add (no other text).

## E.9 LLM-Seq **Template for** `Diabetes Progression`

/* Main System Prompt */
Given a list of features already selected and a list of candidate features available, your task is to output the next feature that should be included to maximally improve the performance in predicting the disease progression status in diabetes patients.

/* Main User Prompt */
I used the features ⟨selected concepts⟩, and the trained model achieved a test ⟨metric⟩ of ⟨value⟩. What feature should I add next from: ⟨candidate concepts⟩? Give me just the name of the feature to add (no other text).

## E.10 LLM-Seq **Template for** `Wine Quality`

/* Main System Prompt */
Given a list of features already selected and a list of candidate features available, your task is to output

the next feature that should be included to maximally improve the performance in predicting whether a wine is high or low quality.

/* Main User Prompt */
I used the features ⟨selected concepts⟩, and the trained model achieved a test ⟨metric⟩ of ⟨value⟩. What feature should I add next from: ⟨candidate concepts⟩? Give me just the name of the feature to add (no other text).

### E.11 LLM-Seq **Template for** `Miami Housing`

/* Main System Prompt */
Given a list of features already selected and a list of candidate features available, your task is to output the next feature that should be included to maximally improve the performance in predicting the selling price of homes in Miami.

/* Main User Prompt */
I used the features ⟨selected concepts⟩, and the trained model achieved a test ⟨metric⟩ of ⟨value⟩. What feature should I add next from: ⟨candidate concepts⟩? Give me just the name of the feature to add (no other text).

### E.12 LLM-Seq **Template for** `Used Cars`

/* Main System Prompt */
Given a list of features already selected and a list of candidate features available, your task is to output the next feature that should be included to maximally improve the performance in predicting the selling price of a used car.

/* Main User Prompt */
I used the features ⟨selected concepts⟩, and the trained model achieved a test ⟨metric⟩ of ⟨value⟩. What feature should I add next from: ⟨candidate concepts⟩? Give me just the name of the feature to add (no other text).

### E.13 LLM-Seq **Template for** `NBA*`

/* Main System Prompt */
Given a list of features already selected and a list of candidate features available, your task is to output the next feature that should be included to maximally improve the performance in predicting the number of points per game of an NBA basketball player.

/* Main User Prompt */
I used the features ⟨selected concepts⟩, and the trained model achieved a test ⟨metric⟩ of ⟨value⟩. What feature should I add next from: ⟨candidate concepts⟩? Give me just the name of the feature to add (no other text).

### E.14 LLM-Seq **Template for** `NYC Rideshare*`

/* Main System Prompt */
Given a list of features already selected and a list of candidate features available, your task is to output the next feature that should be included to maximally improve the performance in predicting the total pay given to a rideshare driver from a trip.

/* Main User Prompt */
I used the features ⟨selected concepts⟩, and the trained model achieved a test ⟨metric⟩ of ⟨value⟩. What feature should I add next from: ⟨candidate concepts⟩? Give me just the name of the feature to add (no other text).

