# OpenReview forum: "LLM-Select: Feature Selection with Large Language Models"
_TMLR — Accepted by TMLR_

### Review · Reviewer_4H3k · 2024-08-07

**Summary Of Contributions:**

This paper illustrates the use of large language models (LLMs) for feature selection in various prediction tasks, given only the feature names and task description as the input. The experimental results suggest that without using the training data, these models can identify features that achieve predictive performance competitive with commonly used data-driven feature selection methods.

**Audience:**

Yes

**Claims And Evidence:**

Yes

**Requested Changes:**

It is reasonable to assume that the performance of the proposed method depends on (1) the quality and relevance of the variable descriptions, and (2) whether the predictive relationship is an existing or a new one. The authors should conduct more experiments to examine the sensitivity of the proposed method to these factors.

Also, more complex and challenging problems should be considered. For example, it would be interesting to see how the LLM-based method performs in inferring the gene-gene and gene-trait relationships in a gene expression dataset with thousands of genes.

**Strengths And Weaknesses:**

Strengths:

The paper describes three approaches to prompting LLMs for feature selection. The experiments are conducted on small-scale and large-scale datasets, and the performance of LLMs is compared with data-driven feature selection methods.

Weaknesses:

Overall, the paper demonstrates the effectiveness of using LLMs for feature selection via extracting information from feature names. The results, however, are not surprising, since all datasets tested in the paper contain informative and well-understood variable descriptions; the underlying relationships have, of course, been studied in the literature, even though the datasets are published after the release of the LLMs. By contrast, the data-driven methods learn the relationships solely from the current dataset. There is not much point in conducting such comparisons.

---

### Review · Reviewer_Jem7 · 2024-09-05

**Summary Of Contributions:**

This paper reveals that large language models (LLMs) can select the most predictive features for a task, using only input feature names and a task description. Their performance matches standard data science tools, and they demonstrate this ability across different query methods.

**Audience:**

Yes

**Broader Impact Concerns:**

N/A.

**Claims And Evidence:**

No

**Requested Changes:**

Weaknesses

**Strengths And Weaknesses:**

Strengths

1. This paper considers different decoding strategies: greedy decoding and self-consistency decoding.

2. The experiments in the paper use various setups and evaluation metrics, and the result analysis is quite detailed.

Weakness

1. The datasets involved in the experiments are from very common domains (Credit-G, Bank, Give Me Some Credit, COMPAS Recidivism, Pima Indians Diabetes, AUS Cars, YouTube). For datasets from particularly specialized and rare fields, the effectiveness of LLM feature selection is unknown.

2. The baselines used in this paper are very old. LASSO (1996), recursive feature elimination (2002), minimum redundancy maximum relevance selection (2005), filtering by mutual information (1992).

3. This paper lacks more exploration of the underlying principles behind using LLMs for feature selection.

4. Based on the experimental results, the best-performing feature selection methods are still some of the classic approaches. LLM-based feature selection does not show any particularly significant advantages.

---

### Review · Reviewer_npvb · 2024-11-01

**Summary Of Contributions:**

This paper proposes to use LLMs to perform feature selection for the downstream tasks without access the whole training dataset. Specifically, the paper proposes three methods for feature selection using LLMs, i.e., LLM-Score, LLM-Rank, and LLM-Seq, and conducts extensive experiments on small-scale datasets and large-scale datasets, and the experimental results show that using LLMs for feature selection can be comparable to (strong) traditional data-driven feature selection methods, especially when the size of LLMs is large enough.

**Audience:**

Yes

**Claims And Evidence:**

Yes

**Requested Changes:**

1. In the related work section, please include more recent studies from 2024 on LLMs, especially regarding new prompting techniques like ToT [1] and GoT [2], and novel decoding methods. This will help to strengthen the paper.

> [1] Yao, Shunyu, et al. "Tree of thoughts: Deliberate problem solving with large language models." Advances in Neural Information Processing Systems 36 (2024).

> [2] Besta, Maciej, et al. "Graph of thoughts: Solving elaborate problems with large language models." Proceedings of the AAAI Conference on Artificial Intelligence. Vol. 38. No. 16. 2024.

2. It is unclear whether a validation set is included in the LLM-Seq method. The paper should specify if a validation set is used and how it is partitioned from the main dataset and how to get the performance over the validation set.

3. **Unclear mechanism for feature selection in LLM-Seq:** In LLM-Seq method, the paper does not clearly explain how the LLM interprets the performance of the t-1 th model on the validation set as presented in the prompt. A detailed description of how the LLM processes and reacts to these performance metrics (values) is essential to understand the mechanics behind the feature selection process. The paper should clarify the mechanism by how the performance metrics guide the LLM in selecting the next feature, ensuring the method’s transparency and replicability.

4. The LLM-Seq method is a greedy approach that does not seem to guarantee an optimal solution.  The paper could benefit from adding some discussion on it.

5. The paper does not address whether different prompt templates were tested to observe their impact on results. Exploring the effects of various templates is essential for assessing the robustness and adaptability of the proposed LLM based method.

6. **Need more details about the ICL examples:** The paper lacks disclosure regarding the specifics of human-annotations used for In-Context Learning (ICL), i.e., the few-shot examples. It is unclear how many samples are utilized as ICL examples and whether the ICL examples for different test samples from the same dataset are fixed or vary. Additionally, the criteria for selecting ICL examples are not discussed—is it based on similarity or another standard? More detailed information on these aspects would improve the paper.

7. **Need more details about CoT:** The paper does not provide details on how the content used in the Chain of Thought (CoT) approach is obtained. It is unclear whether this content is derived from human-annotations or generated by the LLM itself. Further clarification on the source of CoT content would enhance the understanding of the methods used.

8. The paper mentions averaging over 5 samples with $T=0$ for GPT-4 and GPT-3.5 to address the inherent non-determinism in generation. However, it is unclear how this averaging is implemented since direct averaging of text outputs is not feasible. Are the authors averaging scores rather than reasoning outputs? More detailed methods, such as those for obtaining deterministic outputs found in OpenAI's cookbook [1], could provide a clearer and more efficient approach to managing non-determinism without requiring multiple inferences. Further explanation on this process would significantly improve the clarity and reproducibility of the results.

> [1] https://cookbook.openai.com/

9. **Misleading metrics used in labeled imbalance setting:** The use of AUROC as the sole evaluation metric on datasets with label imbalance issues, such as the Bank and Give Me Some Credit datasets, may lead to misleading interpretations [1,2]. The authors should include additional metrics that better handle class imbalance.

> [1] Muschelli III, John. "ROC and AUC with a binary predictor: a potentially misleading metric." Journal of classification 37.3 (2020).

> [2] Saito, Takaya, and Marc Rehmsmeier. "The precision-recall plot is more informative than the ROC plot when evaluating binary classifiers on imbalanced datasets." PloS one 10.3 (2015).

10. According to Appendix A1, the experiments utilize 5-fold cross-validation with five different seeds (1,2,3,4,5), and the testing set does not vary across these seeds. It is unclear whether this repeated k-fold validation is conducted for each train-test split. The purpose of this approach should be clarified. Is it intended to ensure the robustness of the results, or could it potentially lead to redundant computations?

11. **Concerns Regarding Inconsistencies in Results Across Figures:** I have concerns regarding the results presented in Fig. 2 (a) and Fig. A1 (a), (b), which suggest that the LLM-Score (based on GPT-4.0) performs better than LLM-Rank and LLM-Seq, particularly in classification tasks. However, according to the results in Fig. 3 (a), the performance of LLM-Score does not appear to be superior to that of LLM-Rank and LLM-Seq, especially under the 30% feature setting. The authors need to clarify why there are inconsistencies in these results.

12. In Fig. 4, it is not disclosed what proportion of features is used to obtain the results.

13. **Concerns About One-Shot Setting in Fig. 4 Experiments:** The experiments in Fig. 4 utilize a one-shot setting when adding examples, which is not a common configuration and casts doubt on the credibility of the results for me. Although the authors claim that this choice is due to the reduced number of features, further explanation is needed. Why does a smaller feature set justify the use of a one-shot setting? This justification is crucial for assessing the validity of the experimental design and results. Additionally, for further concerns about few-shot examples, please refer to Question 6.

14. In Fig. 4, why does the "context" condition show better performance than the "context + examples" condition? The authors should provide potential explanations for this unexpected result.

15. In Appendix B.1, why is the test set kept constant during the dataset splitting process? The authors should explain the rationale behind maintaining an unchanged test set.

16. **Inconsistency in Baseline Methods for Different Dataset Scales:** Why is Lasso used as the feature selection baseline for small-scale datasets, while group Lasso (gLasso) is used for large-scale datasets? The authors should provide a rational explanation for this choice. What considerations led to the use of different methods based on the scale of the dataset? Additionally, there is a discrepancy in Fig. B2 of the large-scale dataset experimental results, where Lasso is reported instead of gLasso. This inconsistency is confusing and needs to be clarified.

17. **Concerns About Results in Fig. B2 for the Folktables: Employment Dataset:** I have concerns regarding the results shown in Fig. B2, specifically for the Folktables: Employment dataset. The performance across three different downstream tasks (i.e., LightGBM, MLP, and LR) appears remarkably similar. The authors should provide a reasonable explanation for this uniformity in performance. Is there an underlying factor within the dataset or feature selection method that leads to such consistent results across varied models?

**Strengths And Weaknesses:**

**Strengths:**

1. This paper addresses an interesting topic by employing LLMs for feature selection. The proposed method, which can also help to reduce the cost of collecting data in specific domains such as healthcare, is both practical and forward-thinking.

2. The experiments are well-conceived, covering datasets of various sizes and different time periods to mitigate the concern of data memorization by LLMs. This ensures the robustness and applicability of the results.

3. This paper provides extensive implementation details, which enhances its clarity and replicability.

**Weaknesses:**

Please refer to the Requested Changes part.

---

### Decision · Action_Editor_bZ1b · 2025-03-19

**Recommendation:** Accept as is

**Comment:**

Two reviewers recommended accepting this paper. One did not post an official recommendation and highlighted some weaknesses in their review. On my read, the authors suitably addressed the weaknesses in their rebuttal. I therefore recommend acceptance.

**Audience:**

Certainly some of TMLR's audience is interested in feature selection. One reviewer pointed out that the method doesn't necessarily outperform data-based baselines, but the authors noted that the method is still potentially useful in data-free settings, specifically for the purpose of deciding what features to measure before any data is collected. It may be pertinent to highlight this in the paper (e.g. via boldface).

**Claims And Evidence:**

This paper demonstrates that LLMs can perform reasonable feature selection based only on a description of the features and the problems themselves. Reviewers generally agreed that the experiments supported the method. One reviewer was concerned that the datasets did not reflect specialized scenarios and that there were only relatively old data-based feature selection baselines, but the authors added additional experiments and clarification to address this concern.